# AutoUAD: Hyper-parameter Optimization for Unsupervised Anomaly Detection

**Wei Dai   Jicong Fan**[*]
School of Data Science, The Chinese University of Hong Kong, Shenzhen
`weidai@link.cuhk.edu.cn`   `fanjicong@cuhk.edu.cn`

## Abstract

Unsupervised anomaly detection (UAD) has important applications in diverse fields such as manufacturing industry and medical diagnosis. In the past decades, although numerous insightful and effective UAD methods have been proposed, it remains a huge challenge to tune the hyper-parameters of each method and select the most appropriate method among many candidates for a specific dataset, due to the absence of labeled anomalies in the training phase of UAD methods and the high diversity of real datasets. In this work, we aim to address this challenge, so as to make UAD more practical and reliable. We propose two internal evaluation metrics, *relative-top-median* and *expected-anomaly-gap*, and one semi-internal evaluation metric, *normalized pseudo discrepancy* (NPD), as surrogate functions of the expected model performance on unseen test data. For instance, NPD measures the discrepancy between the anomaly scores of a validation set drawn from the training data and a validation set drawn from an isotropic Gaussian. NPD is simple and hyper-parameter-free and is able to compare different UAD methods, and its effectiveness is theoretically analyzed. We integrate the three metrics with Bayesian optimization to effectively optimize the hyper-parameters of UAD models. Extensive experiments on 38 datasets show the effectiveness of our methods.

## 1 Introduction

Unsupervised anomaly detection (UAD), often referred to as the one-class classification problem, is a critical machine learning task with applications across diverse fields such as cybersecurity, fraud detection, industrial quality control, and medical diagnostics (Aggarwal, 2016; Pang et al., 2021). These domains typically encounter rare or unknown anomalous events, making the lack of labeled anomaly data a key challenge. UAD addresses this by training models exclusively on "normal" data[1], with the primary objective of learning the boundary that defines normality. This allows the model to identify unseen data points deviating from the normality as potential anomalies.

Recent advances in deep learning have led to highly effective deep UAD methods (Aggarwal, 2016; Ruff et al., 2018; Zong et al., 2018; Pidhorskyi et al., 2018; Pang et al., 2019; Goyal et al., 2020; Yan et al., 2021; Qiu et al., 2021; Shenkar & Wolf, 2022; Han et al., 2022; Cai & Fan, 2022; Tur et al., 2023; Zhang et al., 2023; 2024; Fu et al., 2024), often outperforming traditional shallow models like K-nearest-neighbors (KNN) (Zimek et al., 2012; Sun et al., 2022), local outlier factor (LOF) (Breunig et al., 2000), isolation forest (IForest) (Liu et al., 2008), and one-class support vector machine (OCSVM) (Schölkopf et al., 2001). While deep UAD methods offer greater performance and efficiency, they exhibit a longer list of hyper-parameters, such as network depth, hidden dimension, weight decay, training epochs, learning rate, and algorithm-specified hyper-parameters. Meanwhile, they are sensitive to the configuration of their hyper-parameters.

Compared to supervised learning, hyper-parameter tuning for unsupervised learning tasks like UAD is particularly challenging. In supervised tasks, labeled validation sets provide direct feedback during the hyper-parameter search, allowing practitioners to iteratively adjust parameters based on measurable improvements (Hutter et al., 2019). In contrast, the absence of labeled data poses a unique

---

[*]Corresponding author
[1]UAD assumes all or most of the training data are from normal conditions of a system. See Definition 1.

challenge in unsupervised tasks, making it highly difficult to compare across different models and tune their hyper-parameters (Halkidi & Vazirgiannis, 2001; Poulakis, 2020; Fan et al., 2022). In Figure 1, more intuitively, we present the sensitivity of four UAD methods when tuning two of their hyper-parameters. It can be seen that the performance of each method in terms of AUC over its two hyper-parameters has many local maximums and minimums. Moreover, the distributions of each method's performance on different datasets are significantly different.

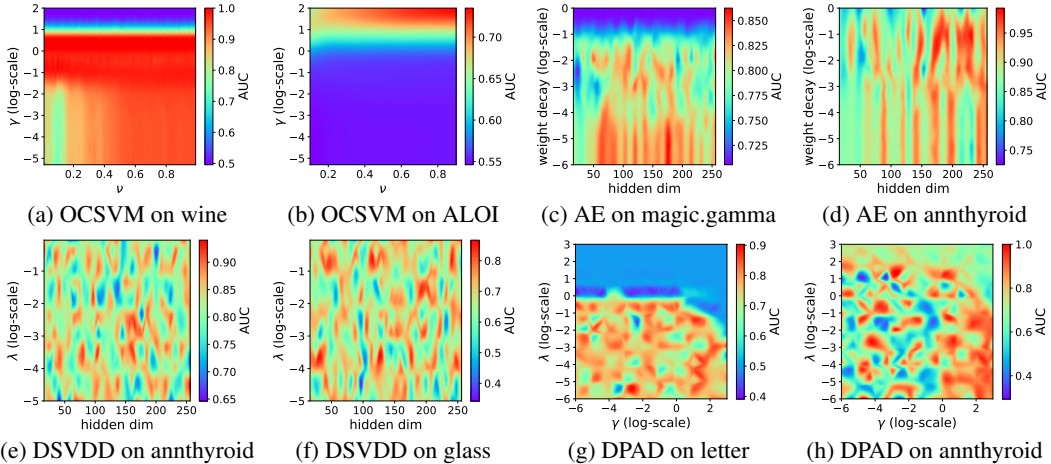

Figure 1: Heatmap visualization of AUC (area under the ROC curve) over two hyper-parameters (described in Appendix H) in four UAD methods including OCSVM (Schölkopf et al., 2001), AE (autoencoder), Deep SVDD (Ruff et al., 2018), and DAPD (Fu et al., 2024) on different datasets.

Although there have been some insightful attempts at hyper-parameter tuning and model selection for anomaly detection, they face significant challenges in unsupervised settings. Recent methods attempt to address this by using historical datasets to train meta-models or assuming a prior knowledge of the anomaly ratio. However, both strategies introduce elements of supervision, contradicting the fundamental principles of unsupervised learning. This reliance on prior information compromises the unsupervised nature of the task. Specifically, meta-learning-based methods (Zhao et al., 2021; 2022; Zhao & Akoglu, 2024; Ding et al., 2024) require access to a collection of historical datasets, which may not always be available or relevant. Similarly, methods (Nguyen et al., 2016) that assume a known anomaly ratio are impractical in the real world, where anomalies are rare and their proportion is typically unknown. These limitations highlight the urgent need for more effective and fully unsupervised model selection techniques that can operate without these assumptions.

To address these challenges, we first propose two new internal evaluation metrics, *relative-topk-median* and *expected-anomaly-gap*, under two proper assumptions in UAD. In the empirical studies, we found that these two metrics do not always work well and are not effective in comparing different UAD methods because they could overfit the training data when implemented on a complex UAD model, and they both have an additional hyper-parameter to determine in advance, as the assumption defined. We then propose a semi-internal evaluation metric, *normalized pseudo discrepancy* (NPD), which measures the discrepancy between the anomaly scores of a validation set drawn from the training data and a validation set drawn from an isotropic Gaussian distribution. It offers more robust and reliable results under simpler assumptions without additional hyper-parameters. Our metrics provide a more nuanced understanding of model performance based on a reliable theoretical guarantee, enabling a better selection of hyper-parameters and models in UAD tasks.

Aiming at model and hyper-parameter selection for UAD algorithms and improving the convenience, accuracy, and efficiency of UAD, our contributions are highlighted as follows.

- We propose two internal evaluation metrics, *relative-top-median* and *expected-anomaly-gap*, and one semi-internal evaluation metric, *normalized pseudo discrepancy*, for automated UAD.
- We implement automated UAD using Bayesian optimization. It automatically and efficiently selects the possibly best hyper-parameters guided by our proposed metrics.
- We provide theoretical guarantees for our NPD metric to ensure feasibility and reliability.

- We conduct extensive empirical experiments on 38 benchmark datasets with four popular UAD algorithms. The results show that NPD consistently outperforms existing model selection heuristics and significantly works well on the complex state-of-art UAD algorithms.

## 2 RELATED WORK

Prior studies have shown that UAD methods are sensitive to hyper-parameter choices (Goldstein & Uchida, 2016; Ding et al., 2022). Recently, hyper-parameter tuning and model selection for unsupervised outlier detection have gained attention. Marques et al. (2015) proposed IREOS to evaluate KNN and LOF, but it requires additional per-sample training, reducing efficiency. Nguyen et al. (2016) compared clustering quality measures for outlier detection, but these metrics need prior knowledge of the outlier ratio, introducing extra hyper-parameters. It contradicts the goal of hyper-parameter optimization. Consensus-based methods (Duan et al., 2020; Lin et al., 2020; Ma et al., 2023) use voting mechanisms to select reliable models, but model pools construction are still sensitive to hyper-parameters. Goix (2016) proposed statistical metrics Mass-Volume (MV) and Excess-Mass (EM), assuming outliers appear in the tail of the score distribution. Nevertheless , these studies focus only on shallow outlier detection methods.

Deep outlier detection methods have three main categories for model selection. The first, *early stopping*, as proposed by (Huang et al., 2024), uses an inlier priority assumption and loss entropy to select models during deep learning training iterations. The second category, *hyper-ensemble* methods, combines multiple models with varying hyper-parameters to enhance detection capability (Ding et al., 2022; Nawaz et al., 2024). However, these methods introduce additional hyper-parameters, such as ensemble weights, and do not guarantee effective outlier detection from the hyper-parameter pool. The third category is based on *meta-learning*, which leverages historical tasks to inform new outlier detection tasks. For example, (Zhao et al., 2021; 2022; Zhao & Akoglu, 2024; Ding et al., 2024) train a meta-model on historical datasets with ground-truth labels to predict hyper-parameters for new unlabeled datasets. However, this approach requires supervision, conflicting with unsupervised learning principles. Moreover, if there's a significant domain gap between historical and current datasets(exampled by the significantly different performance distributions in Figure 1), meta-learning may underperform. Lastly, meta-learning typically focuses on a single detection model, making it challenging to compare various candidate models.

It is worth mentioning that previous hyper-parameter optimization methods focus on the unsupervised outlier detection problem where transductive learning is performed to recognize the outliers in the training set. In contrast, the UAD problem considered in this study is to train a model on a "normal" dataset to detect the newly incoming anomalies in an inductive learning manner. Existing model selection methods for unsupervised outlier detection may fail when encountering UAD because the training set contains no outliers or very few outliers. For example, the inlier priority assumption does not hold for the early stopping method proposed by (Huang et al., 2024). In this paper, we focus on the hyper-parameter tuning and model selection for UAD methods and take advantage of Bayesian optimization (Snoek et al., 2012).

Note that hyperparameter tuning is a challenge in other unsupervised learning tasks as well. (Halkidi & Vazirgiannis, 2001; Poulakis, 2020; Fan et al., 2022) presented some clustering validity metrics to guide the hyper-parameter search using grid search or Bayesian optimization. Particularly, Fan et al. (2022) proposed AutoSC to select models and hyperparameters for spectral clustering. Lin & Fukuyama (2024); Liao et al. (2023) also used Bayesian optimization to tune the hyperparameter in dimensionality reduction methods such as t-SNE.

## 3 AUTOMATED UNSUPERVISED ANOMALY DETECTION (AUTOUAD)

For completeness and clarity, we present the following definitions of UAD and AutoUAD.

**Definition 1 (UAD)** *Consider a dataset $\mathcal{X} = \{\boldsymbol{x}_i \in \mathbb{R}^d : i = 1, 2, \ldots, N\}$ satisfies all of the following assumptions:*

- *$N_0$ samples are randomly drawn from an unknown distribution $\mathcal{D}_0$ deemed as a distribution of normal data and $N_1 := N - N_0$ samples are randomly drawn from another unknown distribution $\mathcal{D}_1$ deemed as an anomalous distribution;*

- *For every $\boldsymbol{x}_i$ in $\mathcal{X}$, whether it is drawn from $\mathcal{D}_0$ or $\mathcal{D}_1$ remains unknown;*
- *$N_0$ and $N_1$ are unknown but $N_0 \gg N_1$ (e.g., $N_0 = 10N_1$);*
- *The overlap (Inman & Bradley Jr, 1989) between $\mathcal{D}_0$ and $\mathcal{D}_1$, defined as $\eta(\mathcal{D}_0, \mathcal{D}_1) = \int_{\mathbb{R}^d} \min\{p_{\boldsymbol{x} \sim \mathcal{D}_0}(\boldsymbol{x}), p_{\boldsymbol{x} \sim \mathcal{D}_1}(\boldsymbol{x})\} d\boldsymbol{x}$, is sufficiently small.*

*Based on such an $\mathcal{X}$, UAD learns a function $f : \mathbb{R}^d \to \mathbb{R}$ that assigns an anomaly score for an arbitrary unseen data (test data) $\boldsymbol{x}$, where the higher $f(\boldsymbol{x})$ is, the more likely $\boldsymbol{x}$ is an anomalous data belonging an unknown anomaly distribution $\mathcal{D}_1'$, where $\eta(\mathcal{D}_1', \mathcal{D}_0)$ is sufficiently small.*

**Remark 3.1** *In some scenarios, we may have $N_1 = 0$, meaning that all samples in the training dataset are normal. The sufficiently small overlaps $\eta(\mathcal{D}_1, \mathcal{D}_0)$ and $\eta(\mathcal{D}_1', \mathcal{D}_0)$ mentioned in the definition are to ensure that the problem is meaningful, i.e., it is possible to distinguish between normal samples and anomalous samples. In addition, when the difference between $\mathcal{D}_1'$ and $\mathcal{D}_1$ is larger, the task is more difficult. Note that each of $\mathcal{D}_0$, $\mathcal{D}_1$, and $\mathcal{D}_1'$ may be a mixture of multiple distributions (sub-populations), which will further increase the difficulty of the task.*

**Definition 2 (AutoUAD)** *Given a dataset $\mathcal{X}$ defined in Definition 1, suppose we have a set of $C$ candidate UAD methods $\mathbb{M} := \{\mathcal{M}_1, \mathcal{M}_2, \ldots, \mathcal{M}_C\}$ and each method $\mathcal{M}_i$ has a set of $H_i$ hyper-parameters $\Theta_i := \{\theta_1^{(i)}, \theta_2^{(i)}, \ldots, \theta_{H_i}^{(i)}\}$, where $\theta_j^{(i)} \in \mathcal{S}_j^{(i)}$ and $\mathcal{S}_j^{(i)}$ denotes a constraint set of the $j$-th hyper-parameter of the $i$-th method. Denote $f_{\mathcal{M}_i}(\boldsymbol{x}|\Theta_i)$ the scoring function of the method $\mathcal{M}_i$ with parameters $\Theta_i$. Let $\tilde{\mathcal{D}}$ be the joint distribution of feature and label of unseen test data and let $(\tilde{\mathcal{X}}, \tilde{\mathcal{Y}})$ be a test set with $\tilde{N}$ samples drawn from $\tilde{\mathcal{D}}$. Denote $\mathcal{E}(\tilde{\mathcal{Y}}, \hat{\tilde{\mathcal{Y}}})$ an evaluation metric of anomaly detection (e.g., AUC or F1-score) for $\tilde{\mathcal{X}}$, where $\hat{\tilde{\mathcal{Y}}}$ denotes the anomaly scores provided by a UAD method. We aim to achieve the following two goals:*

- *Goal 1: For each $\mathcal{M}_i \in \mathbb{M}$, solve*

$$\Theta_i^* = \underset{\Theta_i \in \prod_{j=1}^{H_i} \mathcal{S}_j^{(i)}}{\arg\max} \; \mathbb{E}_{(\tilde{\mathcal{X}}, \tilde{\mathcal{Y}}) \sim \tilde{\mathcal{D}}^{\tilde{N}}} \left[ \mathcal{E}(\tilde{\mathcal{Y}}, \{f_{\mathcal{M}_i}(\boldsymbol{x}|\Theta_i) : \boldsymbol{x} \in \tilde{\mathcal{X}}\}) \right] \tag{1}$$

*where $\prod$ is the Cartesian product and $\mathbb{E}$ is the expectation over the randomness of $\tilde{\mathcal{X}}$.*
- *Goal 2: Let $Q(\mathcal{M}_i(\Theta_i^*)) := \mathbb{E}_{(\tilde{\mathcal{X}}, \tilde{\mathcal{Y}}) \sim \tilde{\mathcal{D}}^{\tilde{N}}}[\mathcal{E}(\tilde{\mathcal{Y}}, \{f_{\mathcal{M}_i}(\boldsymbol{x}|\Theta_i^*) : \boldsymbol{x} \in \tilde{\mathcal{X}}\})]$ and $\mathbb{M}(\Theta^*) := \{\mathcal{M}_1(\Theta_1^*), \mathcal{M}_2(\Theta_2^*), \ldots, \mathcal{M}_C(\Theta_C^*)\}$, solve*

$$\mathcal{M}^*(\Theta^*) = \underset{\mathcal{M}(\Theta^*) \in \mathbb{M}(\Theta^*)}{\arg\max} \; Q(\mathcal{M}(\Theta^*)) \tag{2}$$

*where $\mathcal{M}^*(\Theta^*)$ is the best among all candidate methods with their best hyper-parameters.*

**Remark 3.2** *An example for $\mathbb{M}$ is $\{$KNN, OCSVM, DeepSVDD$\}$. Particularly, for $\mathcal{M}_2$, namely OCSVM, the set of hyper-parameters is $\Theta_2 = \{\nu, \text{'kernel'}, \gamma, \alpha_0\}^2$, where, for example, the search space for $\gamma$ is $\mathcal{S}_3^{(2)} = (0, \infty)$, though we may reduce the search space according to experience.*

AutoUAD defined by Definition 2 is a challenging problem due to the fact that the training data $\mathcal{X}$ is unlabeled and there is no available prior knowledge about the test data $\tilde{\mathcal{X}}$. What we can do is designing a metric $\mathcal{V}$ based on $\mathcal{M}_i$ and $\mathcal{X}$ as a surrogate for $\mathbb{E}[\mathcal{E}(\tilde{\mathcal{Y}}, \{f_{\mathcal{M}_i}(\boldsymbol{x}|\Theta_i) : \boldsymbol{x} \in \tilde{\mathcal{X}}\})]$, i.e.,

$$\mathcal{V}(\mathcal{M}_i, \mathcal{X}) \approx g\left( \mathbb{E}_{(\tilde{\mathcal{X}}, \tilde{\mathcal{Y}}) \sim \tilde{\mathcal{D}}^{\tilde{N}}} \left[ \mathcal{E}(\tilde{\mathcal{Y}}, \{f_{\mathcal{M}_i}(\boldsymbol{x}|\Theta_i) : \boldsymbol{x} \in \tilde{\mathcal{X}}\}) \right] \right) \tag{3}$$

where $g : \mathbb{R} \to \mathbb{R}$ is ideally a monotonically increasing function.

## 3.1 RELATIVE TOP-MEDIAN METRIC

In any real-world dataset, there are typically some data points that are far from the majority of the data distribution, especially when the number of data points becomes large. These data points, though not necessarily true anomalies, often exhibit characteristics that make them appear closer to

---

[2]According to scikit-learn (Pedregosa et al., 2011), $\nu$ is the training error bound, $\gamma, \alpha_0$ are kernel coefficient, and 'kernel' is the choice of kernel function.

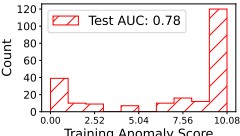 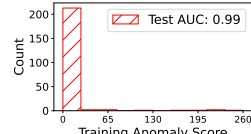 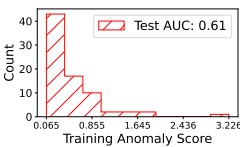 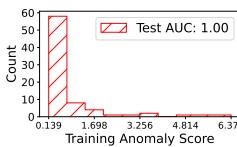

(a) Low-AUC OCSVM model on breastw    (b) High-AUC OCSVM model on breastw    (c) Low-AUC DSVDD model on lymph.    (d) High-AUC DSVDD model on lymph.

Figure 2: Histograms of training anomaly scores of models with low and high testing AUCs. This conforms to Assumption 1. A larger RTM (Definition 3) corresponds a higher AUC.

anomalies than the majority of the data. For example, in a dataset following a Gaussian distribution in $\mathbb{R}^2$ or $\mathbb{R}^3$, most data points cluster near the mean, but a few data points naturally occur in the tail regions of the distribution. While these tail points are part of the normal data, their distance from the majority can make them resemble potential anomalies. They should be assigned relatively high anomaly scores by a good UAD model. Based on this observation, we make the following assumption.

**Assumption 1** *A good UAD model will assign low anomaly scores to the majority of data points and will assign relatively high anomaly scores to the minority of data points.*

Actually, Assumption 1 aligns with the way many UAD models are designed. For example, in OCSVM, the decision boundary is shaped to enclose the majority of normal data points tightly, leaving the points near or beyond the boundary with higher anomaly scores. To further show the feasibility of the assumption, we present an empirical comparison between models with high AUC and low AUC in Figure 2, where "DSVDD" stands for Deep SVDD (Ruff et al., 2018).

Assumption 1 and Figure 2 provide the following insights:

- The majority of data points, close to the median of the distribution, are presumed to represent normal behavior and should be assigned low anomaly scores.
- A small percentage of data points (e.g., the top $\tau\%$) farthest from the majority are more likely to deviate from normality and should receive relatively higher anomaly scores.

Thus we propose the following metric to approximate $g\big(\mathbb{E}[\mathcal{E}(\tilde{\mathcal{Y}}, \{f_{\mathcal{M}}(\boldsymbol{x}|\Theta) : \boldsymbol{x} \in \tilde{\mathcal{X}}\})]\big)$.

**Definition 3 (RTM)** *Given a UAD method $\mathcal{M}$ with a set of hyper-parameters $\Theta$ trained on a dataset $\mathcal{X}$, the output anomaly scores are $\boldsymbol{s} = \{s_1, s_2, \ldots, s_N\}$ where $s_i = f_{\mathcal{M}}(\boldsymbol{x}_i|\Theta)$ and $\boldsymbol{x}_i \in \mathcal{X}$. Denote $s_{(i)}$ for sorted values with $s_{(1)} \leq s_{(2)} \leq \ldots \leq s_{(\tau/100N)} \leq \ldots \leq s_{(N)}$. The relative top-median (RTM) is defined as*

$$\mathcal{V}_{RTM}(\mathcal{M}, \mathcal{X}) = \frac{mean(\{s_i | s_i \geq s_{(\tau/100N)}\}) - median(\boldsymbol{s})}{median(\boldsymbol{s}) + \epsilon}, \tag{4}$$

*where $\epsilon$ (throughout this paper) is a small constant (e.g. $10^{-6}$) to avoid zero denominator.*

We heuristically set $\tau = 5$ in our experiments. During hyper-parameter searching, if $RTM_{5\%}$ is larger, we say the corresponding model is better.

## 3.2 Expected Anomaly Gap Metric

RTM may overlook a case in which all scores are uniformly distributed from small to large value. To solve this, we propose the following *anomaly gap* metric.

**Definition 4 (AG)** *Let $p(s)$ be the unknown and theoretical distribution of the anomaly score $\boldsymbol{s}$ defined in Definition 3 given by a UAD model and let $\xi$ be a value in the domain of $s$. Define $w_0(\xi) = P(s < \xi), w_1(\xi) = P(s \geq \xi), \mu_0(\xi) = E[s|s < \xi], \mu_1(\xi) = E[s|s \geq \xi], \sigma_0^2(\xi) = Var(s|s < \xi)$, and $\sigma_1^2(\xi) = Var(s|s > \xi)$. Based on Assumption 1, there exists $\xi$ such that $P(s \geq \xi)$ is small, and the variance of both $s \geq \xi$ and $s < \xi$ should also be small. The anomaly gap (AG) at $\xi$ is calculated by*

$$AG(\xi; p(s)) = \frac{w_0(\xi)w_1(\xi)(\mu_0(\xi) - \mu_1(\xi))^2}{w_0(\xi)\sigma_0^2(\xi) + w_1\sigma_1^2(\xi) + \epsilon}. \tag{5}$$

Ideally, the choice of $\xi$ depends on how many training samples are around the optimal decision boundary or similar to the true anomaly, i.e. drawn from $\mathcal{D}_1$ defined in Definition 1. It relies on a threshold number, e.g. top-5%. Due to the non-linearity of real-world data and the lack of supervision, it is hard to select the threshold for each dataset. The metric may also be sensitive to an arbitrarily selected threshold. To reduce the sensitivity, we design to use a relaxed threshold $s_{thr}$ indicating a range where data may draw from $\mathcal{D}_1$ and propose the following *expected-anomaly-gap*.

**Definition 5 (EAG)** *Given the output anomaly scores $\boldsymbol{s}$, let $s_{max} = \max(\boldsymbol{s})$ and let $s_{thr}$ be a threshold value such that $s_{thr} < s_{max}$. Suppose $\xi$ follows a uniform distribution with $p(\xi) = 1/(s_{max} - s_{thr})$. The expected anomaly gap (EAG) is defined as*

$$\mathcal{V}_{EAG}(\mathcal{M}, \mathcal{X}) = \mathbb{E}[AG(\xi; p(s))|\xi \geq s_{thr}] = \int_{s_{thr}}^{s_{max}} AG(\xi)p(\xi)\mathrm{d}\xi = \frac{1}{s_{max} - s_{thr}} \int_{s_{thr}}^{s_{max}} AG(\xi)\mathrm{d}\xi. \quad (6)$$

To determine $s_{\text{thr}}$ for EAG, we make the following assumption.

**Assumption 2** *(Low Quality Data Upper Bound) At most 20% data points in the training set are similar to the true anomalies, in which the UAD model gives them higher anomaly scores.*

This is a very mild assumption for UAD. If there is too much low-quality data, the UAD model may overfit them, leading to bad performance. Limiting the proportion prevents the model from being overly influenced by low-quality data, ensuring that the learned normality patterns remain accurate. In many practical scenarios, such as medical diagnosis and mechanical fault detection, noisy or wrongly collected data constitute a small portion. So, it is realistic and applicable to make the 20% upper bound assumption. Following Assumption 2, we set $s_{\text{thr}}$ as $G^{-1}(0.8)$, where $G(s)$ is the cumulative distribution function (CDF) of $s$ and $G^{-1}(0.8)$ is the inverse CDF evaluated at 0.8, representing the 80th percentile. Since we do not know $p(s)$ but we have $\boldsymbol{s}$ sampled from $p(s)$, we finally calculate EAG by the discrete form $\mathcal{V}_{\text{EAG}}(\mathcal{M}, \mathcal{X}) = \frac{1}{0.2N} \sum_{\xi \geq s_{\text{thr}}} AG(\xi; \boldsymbol{s})$. Sequential and vectorized implementations for calculating EAG are shown in Algorithms 1 and 2 of Appendix A.

## 3.3 NORMALIZED PSEUDO DISCREPANCY GUIDED SEARCH

Previous methods for evaluating UAD models have primarily relied on utilizing the training data's anomaly scores $\boldsymbol{s}$ solely. While these approaches are sometimes useful, they suffer from the following limitations. First, internal evaluation metrics such as RTM and EAG may overfit the training set, as they only observe the data available during training; Second, both RTM and EAG introduce an additional hyper-parameter (a relaxed threshold). While the hyper-parameters can be set to reasonable value under proper assumptions, they tend to be sensitive when applied to complex datasets.

We propose a new method called *normalized pseudo discrepancy* to address these limitations. Recall that our goal is to construct a surrogate metric $\mathcal{V}$ to approximate the expected test performance $\mathbb{E}[\mathcal{E}(\tilde{\mathcal{Y}}, \{f_{\mathcal{M}_i}(\boldsymbol{x}|\Theta_i) : \boldsymbol{x} \in \tilde{\mathcal{X}}\})]$. This can be done if we could generate a reasonable proxy for $\tilde{\mathcal{X}}$. NPD generates this proxy via taking advantage of a randomly generated dataset in addition to a subset of the original training data, aiding in the evaluation by incorporating more diverse perspectives and reducing the risk of overfitting, without using any real anomalies.

**Definition 6 (NPD)** *Suppose $\mathcal{M}$ is the black box UAD model and $\Theta$ is the corresponding hyper-parameters. We randomly split the training data into two subsets $\mathcal{X}_{trn}$ and $\mathcal{X}_{val}$ with sizes $N - M$ and $M$ respectively. The UAD model is trained on $\mathcal{X}_{trn}$. Let $\boldsymbol{\mu}_{trn} \in \mathbb{R}^d$ and $\boldsymbol{\sigma}_{trn}^2 \in \mathbb{R}^d$ be the mean vector and the variance vector of $\mathcal{X}_{trn}$. We generate a dataset $\mathcal{X}_{gen}$ consisting of $M$ samples drawn from an isotropic Gaussian $\mathcal{N}(\boldsymbol{\mu}_{trn}, diag(\boldsymbol{\sigma}_{trn}^2))$. Then, we compute the anomaly score vectors for $\mathcal{X}_{val}$ and $\mathcal{X}_{gen}$, i.e. $\boldsymbol{s}_{val} = f_{\mathcal{M}}(\mathcal{X}_{val}|\Theta)$ and $\boldsymbol{s}_{gen} = f_{\mathcal{M}}(\mathcal{X}_{gen}|\Theta)$ respectively. Finally, we calculate the normalized pseudo discrepancy (NPD) as*

$$\mathcal{V}_{NPD}(\mathcal{M}, \mathcal{X}) = \frac{(Mean(\boldsymbol{s}_{gen}) - Mean(\boldsymbol{s}_{val}))^2}{2(Var(\boldsymbol{s}_{gen}) + Var(\boldsymbol{s}_{val})) + \epsilon}. \quad (7)$$

We argue that $\mathcal{X}_{\text{val}} \cup \mathcal{X}_{\text{gen}}$ is a good proxy for $\tilde{\mathcal{X}}$ and NPD is an effective proxy of the expected test performance $\mathbb{E}[\mathcal{E}(\tilde{\mathcal{Y}}, \{f_{\mathcal{M}_i}(\boldsymbol{x}|\Theta_i) : \boldsymbol{x} \in \tilde{\mathcal{X}}\})]$ by answering the following questions.

- *Why use Gaussian rather than other distributions for $\mathcal{X}_{gen}$?* Real-world data are often non-Gaussian but are often close to Gaussian to some extent (Żuławiński et al., 2023; Khokhlov & Hulot, 2017; Weinberg & Cole, 1992; Marko & Weil, 2012). We hope that $\mathcal{X}_{gen}$ are comparable to $\mathcal{X}_{trn}$ while having high diversity. It is known that within all distributions with the same variance, the Gaussian distribution has the largest entropy, meaning the highest diversity. Theorem 1 shows that the uncertainty of $\mathcal{X}_{gen}$ is always higher than that of $\mathcal{X}_{trn}$. In Figure 3, we illustrate the correlation coefficient matrices of $\mathcal{X}_{trn}$ and $\mathcal{X}_{gen}$, and compare the distribution of normalized $\mathcal{X}_{trn}$ with standard Gaussian distribution. Although they have the same mean and variance, $\mathcal{X}_{trn}$ is from skewed distributions, and the features in $\mathcal{X}_{trn}$ are correlated.
- *Can $\mathcal{X}_{gen}$ contain samples very close to real anomalies?* In Figure 3(right), the distributions of $\mathcal{X}_{trn}$ and $\mathcal{X}_{gen}$ are a little similar but substantially different, which indicates that $\mathcal{X}_{gen}$ contains anomalies close to normal data, raising the difficulty of detection. We visualized $\mathcal{X}_{gen}$ in Figure 7 where $\mathcal{X}_{gen}$ can contain samples close to the real anomalies. Although there is the case of not, the $\mathcal{X}_{gen}$ helps the UAD model build a tight decision boundary around normal data.
- *What if some samples in $\mathcal{X}_{gen}$ are very close to normal data?* In Theorem 2, we show that NPD is upper bounded by the score gap between normal data and anomalies. When maximizing NPD, the gap also becomes larger even if $\mathcal{X}_{gen}$ contains some normal data.
- *Will NPD overfit the training data?* No. NPD is calculated from $\mathcal{X}_{val}$ and $\mathcal{X}_{gen}$ that are independent from the model training data $\mathcal{X}_{trn}$.

**Theorem 1** *Denote $\boldsymbol{x}_{trn}$ the variables of $\mathcal{X}_{trn}$ and $\boldsymbol{x}_{gen}$ the variables of $\mathcal{X}_{gen}$, the entropy of $\boldsymbol{x}_{gen}$ is, almost surely, higher than the entropy of $\boldsymbol{x}_{trn}$, i.e., $H(\boldsymbol{x}_{gen}) > H(\boldsymbol{x}_{trn})$.*

**Theorem 2** *Let $\mathcal{X}_{val} = \mathcal{X}_{val}^0 \cup \mathcal{X}_{val}^1$ and $\mathcal{X}_{gen} = \mathcal{X}_{gen}^0 \cup \mathcal{X}_{gen}^1$, where $\mathcal{X}_{val}^0$ and $\mathcal{X}_{gen}^0$ are normal, $\mathcal{X}_{val}^1$ and $\mathcal{X}_{gen}^1$ are anomalous. The anomaly scores are denoted by $\boldsymbol{s}_{val}^0, \boldsymbol{s}_{val}^1, \boldsymbol{s}_{gen}^0, \boldsymbol{s}_{gen}^1$ accordingly. NPD has the following properties:*

*(a) Letting $\Delta = \left(\frac{|\mathcal{X}_{val}^1|}{M}Mean(\boldsymbol{s}_{val}^1) - \frac{|\mathcal{X}_{gen}^0|}{M}Mean(\boldsymbol{s}_{gen}^0)\right)^2$ and $\Delta' = \left(\frac{|\mathcal{X}_{gen}^1|}{M}Mean(\boldsymbol{s}_{gen}^1) - \frac{|\mathcal{X}_{val}^0|}{M}Mean(\boldsymbol{s}_{val}^0)\right)^2$, it holds that*

$$\mathcal{V}_{NPD}(\mathcal{M}, \mathcal{X}) \leq \frac{\Delta + \Delta'}{(Var(\boldsymbol{s}_{val}) + Var(\boldsymbol{s}_{gen})) + \epsilon/2} \tag{8}$$

*(b) NPD is translation-invariant and scale-invariant w.r.t. the scoring function $f_{\mathcal{M}}$ when $\epsilon = 0$.*

Theorem 2 shows that NPD is upper bounded by the score gap between normal data and anomalies. Specifically, $\Delta$ and $\Delta'$ represent the squared gap between the mean anomaly scores of normal and anomalous data across $\mathcal{X}_{val}$ and $\mathcal{X}_{gen}$. As NPD is maximized, the gap between normal and anomalous data also becomes larger, meaning a clearer separation is achieved. Note that in the worst case, NPD is always zero for all UAD models, which however indicates that $\mathcal{X}_{trn}, \mathcal{X}_{val}$, and $\mathcal{X}_{gen}$ are from the same distribution, i.e. $\mathcal{N}(\boldsymbol{\mu}_{trn}, \text{diag}(\boldsymbol{\sigma}_{trn}^2))$. As the distribution of $\mathcal{X}_{trn}$ is already identified as $\mathcal{N}(\boldsymbol{\mu}_{trn}, \text{diag}(\boldsymbol{\sigma}_{trn}^2))$, it is easy to detect anomalies without hyper-parameters.

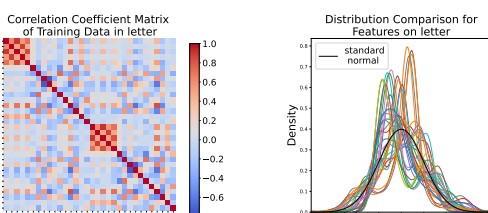

Figure 3: The left is the correlation coefficient matrix of $\mathcal{X}_{trn}$ in the dataset 'letter'. It shows that the distribution of $\mathcal{X}_{trn}$ is far away from an isotropic Gaussian. The right is the distribution comparison between the normalized features and a standard Gaussian. The majority of features are skewed, indicating the $\mathcal{X}_{gen}$ is not similar to the $\mathcal{X}_{trn}$.

**Geometry Interpretation for NPD** From a geometry perspective, the outline of $\mathcal{X}_{gen}$ is a hyper-sphere, denoted as $\mathcal{S}^{d-1}$, in $\mathbb{R}^d$, centered at the mean of $\mathcal{X}_{trn}$. $\mathcal{S}^{d-1}$ is the tightest hyper-sphere enclosing almost all of the points of $\mathcal{X}_{trn}$. In practice, features in normal data are usually correlated such that the volume occupied is small in the hyper-sphere. The rest of the space could be covered by potential anomalies from $\mathcal{D}_1'$. We visualized 4 real-world datasets in Figure 4. Based on the fact that a Gaussian mixture model is a universal approximator of densities, we present a theorem (Theorem 3) in Appendix B to show that the overlap between the distribution of normal training data and the distribution of $\mathcal{X}_{gen}$ could be effectively bounded.

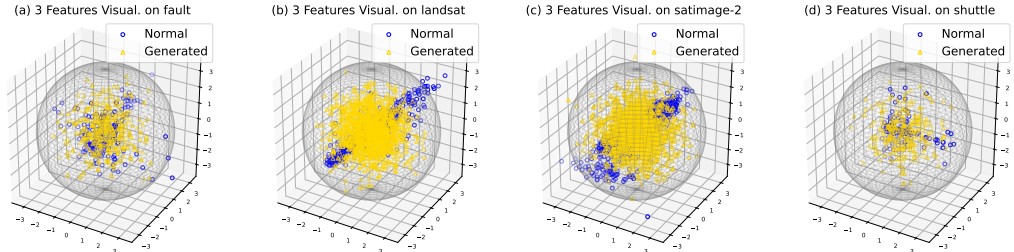

Figure 4: Visualization of 3 randomly selected features of $\mathcal{X}_{\text{trn}}$, $\mathcal{X}_{\text{gen}}$, and the hyper-sphere $\mathcal{S}^2$ supported by $\mathcal{X}_{\text{gen}}$. Notice $\mathcal{X}_{\text{trn}}$ only occupies part of the hyper-sphere. Potential anomalies could appear in the rest of the hyper-sphere space.

**Theoretical Error Rate Bound of NPD** Theorem 4 in Appendix C presents an error bound of FPR and FNR for AutoUAD with NPD, which further supports the effectiveness of our method.

### 3.4 AUTOUAD VIA BAYESIAN OPTIMIZATION

Bayesian optimization (BO) (Jones et al., 1998) is an effective tool for hyperparameter optimization in supervised learning (Snoek et al., 2012; Klein et al., 2017). It optimizes hyperparameters sequentially, using past results to guide future iterations. Compared to grid search, BO is more efficient, particularly for models with many hyperparameters, as grid search faces exponentially growing search spaces and high computational costs. For example, using grid search to solve State 1 of the AutoUAD problem (Definition 2), the number of training models $\mathcal{M}_i$ is $\prod_{j=1}^{H_i} |\mathcal{S}_j^{(i)}|$. Additionally, grid search requires predefined hyperparameter grids, which often involve guesswork. We use BO with the Tree-structured Parzen Estimator (TPE) (Bergstra et al., 2011) for optimization. Details of BO based on $\mathcal{V}(\mathcal{M}, \mathcal{X})$ are in Appendix F, as illustrated in Figure 9.

### 4 EXPERIMENTS AND RESULTS

**Benchmark Datasets and UAD Methods** We here[3] show the effectiveness of AutoUAD with four UAD methods including a shallow method **OCSVM** (Schölkopf et al., 2001) and three deep methods AutoEncoder (**AE**) (Aggarwal, 2016), **DeepSVDD** (Ruff et al., 2018), and **DPAD** (Fu et al., 2024), though there are more UAD methods in the literature. The experiments are conducted on 38 widely-used real-world benchmark datasets collected by ADBench (Han et al., 2022) and DAMI (Campos et al., 2016), excluding those with 50,000 or more samples due to time constraints, though our method is scalable to larger datasets. The dataset information is summarized in Table 3 of Appendix E. Similar to (Shenkar & Wolf, 2022), we randomly split 50% of normal samples for training and used the rest with anomalous data for testing. All data are standardized using the training set's mean and standard deviation. The split of each dataset is the same across different UAD methods. We repeat all experiments with 5 different data splits and report the results with mean and standard deviation. Note that the hyperparameter tuning and model selection are done with the training data only (and the validation data used in our NPD is part of the training data), while the test data is only used to evaluate the final model. Please also see the flowchart in Figure 9.

**Baselines and Evaluation Metric** We compare the proposed three metrics, RTM, EAG, and NPD, with three existing internal evaluation metrics, including ModelCentrality (**MC**) (Lin et al., 2020), **HITS**(Ma et al., 2023), and Mass-Volume (**MV**) / Excess-Mass (**EM**) (Goix, 2016), which uses a similar assumption to RTM and EAG. Besides, we also compare three *no model selection* baselines, **Default** uses the default hyper-parameters adopted by the corresponding UAD algorithm, **Random** picks a set of hyper-parameters randomly from a grid search pool, and **Max** operates as an oracle and is an upper bound of the performance. For the performance metric, we report the area under the ROC curve (AUC) and F1 score of the selected model. The calculation of the F1 score is consistent with (Shenkar & Wolf, 2022; Qiu et al., 2021). The implementation details are in Appendix G.

---

[3]Due to space limitation, the results on more UAD methods are in Appendix J.

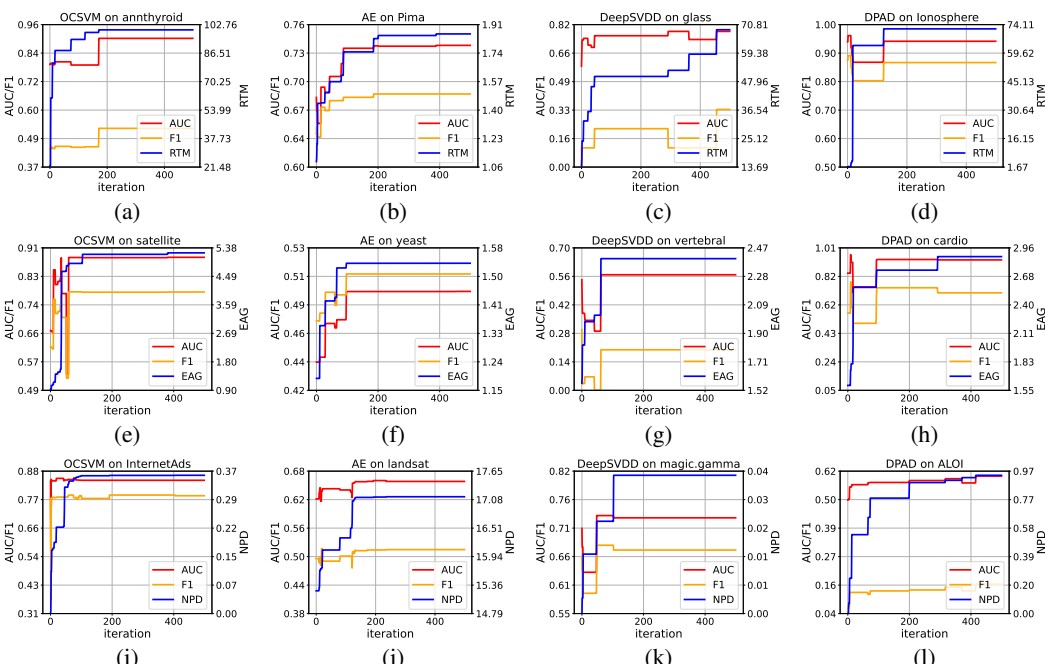

Figure 5: AUC, F1, and the proposed evaluation metrics of OCSVM, AE, DeepSVDD, and DPAD on different datasets in each iteration of BO. The first row (a, b, c, d) are the results of RTM; The second row (e, f, g, h) are the results of EAG; The last row (i, j, k, l) are the results of NPD.

Table 1: Mean AUC and F1 scores through 38 datasets. Default uses the default hyper-parameters. Random is the expected value of the grid search. The p-value of the paired t-test shows the statistical significance compared with Random. The best performance is **bold** and the second best is underline. p-value* is the comparison between NPD and the second best.

| Method | | OCSVM | | AE | | DeepSVDD | | DPAD | |
| --- | --- | --- | --- | --- | --- | --- | --- | --- | --- |
| Metric | | AUC | F1 | AUC | F1 | AUC | F1 | AUC | F1 |
| Max | mean±std | 85.75±15.3 | 63.33±26.1 | 84.86±16.0 | 64.11±26.5 | 80.18±18.1 | 57.63±28.7 | 83.38±14.9 | 61.88±24.5 |
| Default | mean±std | 78.73±19.4 | 54.32±27.6 | 81.47±16.9 | 56.68±25.7 | 73.90±21.7 | 48.93±28.4 | 73.65±18.5 | 48.66±25.1 |
| Random | mean±std | 74.71±16.4 | 48.90±22.8 | 81.88±17.2 | 57.86±25.6 | 74.45±19.9 | 48.73±26.6 | 71.2±14.39 | 46.57±21.1 |
| EM/MV | mean±std | 71.41±22.5 | 45.71±28.7 | 83.08±16.6 | 59.67±25.9 | 74.43±21.7 | 47.63±29.5 | 72.59±18.5 | 46.28±25.9 |
| | p-value | 0.0065 | 0.0437 | 0.0156 | 0.04304 | 0.3301 | 0.3694 | 0.1385 | 0.6731 |
| RTM(Ours) | mean±std | 76.52±20.8 | 53.15±28.9 | 82.15±16.6 | 57.72±27.8 | 73.64±21.4 | 44.87±27.7 | 74.00±20.7 | 44.24±27.4 |
| | p-value | 0.1768 | 0.0415 | 0.7653 | 0.7414 | 0.4833 | 0.0325 | 0.0935 | 0.1684 |
| EAG(Ours) | mean±std | 77.17±19.5 | 51.95±27.5 | 82.35±16.9 | 57.16±26.2 | **75.56**±18.6 | 46.53±27.3 | 71.14±19.3 | 46.16±26.9 |
| | p-value | 0.1333 | 0.1354 | 0.5727 | 0.3866 | 0.1674 | 0.0446 | 0.9189 | 0.7773 |
| NPD(Ours) | mean±std | **84.03**±15.5 | **60.26**±26.2 | **83.58**±16.0 | **59.73**±25.4 | 74.38±20.1 | **48.31**±27.8 | **80.36**±16.8 | **55.67**±23.7 |
| | p-value | **0.0000** | **0.0000** | **0.0212** | **0.0701** | 0.8838 | 0.6024 | **0.0000** | **0.0000** |
| | p-value* | **0.0011** | **0.0014** | 0.181130 | 0.6954 | 0.5909 | 0.1657 | **0.0001** | **0.0006** |

**Intuitive Validation of AutoUAD via BO**    In Figure 5, we present the performance in each iteration of our proposed metrics RTM, EAG, and NPD with four UAD methods. The setting is detailed in Table 5 of Appendix H. We see that in most cases, higher metrics correspond to higher AUC and F1, meaning that they are effective surrogates for the expected model performance on unseen data.

**Comparative Study**    We compare the performance using BO to search for the best model of RTM, EAG, and NPD with Max, Default, Random, and EM/MV in Table 1. The consensus-based metrics are not adaptable for BO because they are designed to select the most reliable model from a set of models, while the performance of each model is not a consideration. The average performance of AUC and F1 through 38 benchmark datasets indicates that our methods have the highest AUC and F1. NPD significantly outperforms other methods when applied to OCSVM, AE, and DPAD. The results for each dataset are in Appendix I. Thus, we achieve Goal 1 of AutoUAD.

To see the effectiveness of our methods for Goal 2 of AutoUAD, taking Figure 6 as an example, we see NPD consistently has a strict positive correlation with AUC and F1, where the Spearman rank

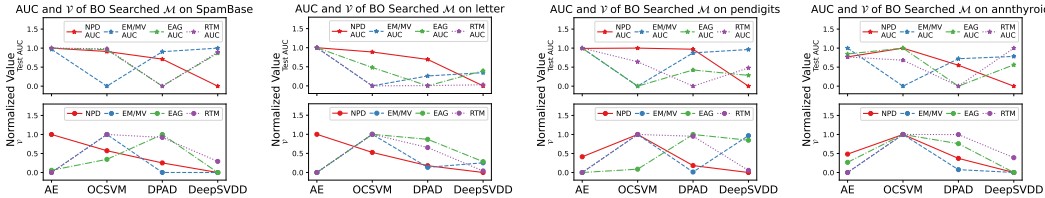

Figure 6: Comparison of EM/MV, RTM, EAG, and NPD on test AUC of selected model in BO hyper-parameter search in 4 datasets. $\mathcal{V}$ is calculated on the training set, and AUC is calculated on the testing set. Both AUC and $\mathcal{V}$ have been normalized. The NPD is positively proportional to the AUC. We can select the best model through different UAD algorithms.

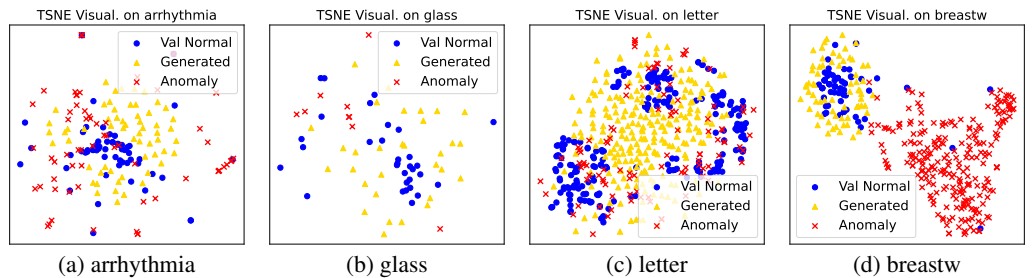

| (a) arrhythmia | (b) glass | (c) letter | (d) breastw |

Figure 7: T-SNE visualization of $\mathcal{X}_{\text{val}}$, $\mathcal{X}_{\text{gen}}$, and true anomalies. The generated Gaussian samples guide the model in learning a decision boundary around normal data.

coefficient is always 1. In contrast, RTM and EAG do not always monotonically increase the AUC and F1 due to overfitting of the training data and the additional hyper-parameter.

**Unsupervised Outlier Model Selection (UOMS) Study**   Instead of using BO to optimize the hyper-parameters, the goal of UOMS is to select the best pair of $\{\mathcal{M}, \theta\}$ among a pool of options (usually a grid search pool). We use the same grid search pool in Table 4 and compare the performance with the consensus-based metrics, MC and HITS, and IForest with the default hyper-parameter. The size of our grid search is much larger, up to 2667 models, compared with previous works Huang et al. (2024); Ma et al. (2023). The results are reported in Table 2. It is seen that our metrics RTM and NPD consistently outperform the baselines indicating our method can achieve goal 2 in AutoUAD among a grid search model pool. MC and HITS obtain a worse performance because the consensus-based methods are not designed to be a surrogate in equation 3. They will be affected by the majority of models. If the majority model has a lower score, the selected model will thereby perform worse.

**Data Visualization**   See the t-SNE (Van der Maaten & Hinton, 2008) results in Figure 7.

Table 2: UOMS comparison through 38 datasets. The notation meanings are the same as Table 1.

## 5  CONCLUSION

This paper studied the problem of hyper-parameter optimization and model comparison for unsupervised anomaly detection. We proposed three surrogate metrics for the expected model performance on unseen data without using any information of anomaly. Particularly, the last metric NPD showed significant improvement over the baselines in the experiments of 38 benchmark datasets. The success of NPD stems from its simplicity and theoretical ground. Nevertheless, the highest NPD did not always correspond to the best model performance. Future work may continuously try to provide more effective surrogate metrics.

| Metrics | | AUC | F1 |
|---|---|---|---|
| Max | mean±std | 89.28±11.8 | 73.33±22.0 |
| Default (IForest) | mean±std | 78.08±18.6 | 49.29±27.7 |
| Random | mean±std | 75.15±16.2 | 49.71±23.2 |
| EM/MV | mean±std | 75.79±19.2 | 51.09±27.70 |
| MC | mean±std | 70.7±18.4 | 44.17±25.7 |
| HITS | mean±std | 73.44±18.8 | 48.13±27.1 |
| RTM (Ours) | mean±std | 79.39±19.6 | 56.84±28.1 |
| | p-value | 0.1308 | 0.1227 |
| EAG (Ours) | mean±std | 76.5±17.4 | 46.25±25.0 |
| | p-value | 0.5670 | 0.3453 |
| NPD (Ours) | mean±std | **83.49**±16.4 | **59.30**±26.2 |
| | p-value | **0.0000** | **0.0000** |
| | p-value* | **0.0014** | **0.0003** |

ACKNOWLEDGMENTS

This work was supported by the General Program of Guangdong Basic and Applied Basic Research under Grant No.2024A1515011771, the National Natural Science Foundation of China under Grant No.62376236, the Guangdong Provincial Key Laboratory of Mathematical Foundations for Artificial Intelligence (2023B1212010001), Shenzhen Science and Technology Program ZDSYS20230626091302006, and Shenzhen Stability Science Program 2023.

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

## A  EAG ALGORITHM

The sequential and vectorized algorithms for calculating EAG are shown in Algorithm 1 and Algorithm 2, where $\rho$ in the algorithm is set to 0.2. The time complexity is $O(N)$ with prefix sum computation of mean and variance. So, EAG is scalable to large dataset.

---

**Algorithm 1** Expected Anomaly Gap

---

**Input:** maximum percentile $\rho$, anomaly score vector $s$;
1: $N \leftarrow s.\text{size}; \Delta \leftarrow 0$;
2: Top $\rho N$ sort $s$ in descending order;
3: **for** $k \in \{1, ..., \rho N\}$ **do**
4:    $\mu_0 \leftarrow \text{mean}(s[: k]); \mu_1 \leftarrow \text{mean}(s[k :])$;
5:    $\sigma_0 \leftarrow \text{variance}(s[: k]); \sigma_1 \leftarrow \text{variance}(s[k :])$;
6:    $\Delta \leftarrow \Delta + k(N - k)(\mu_0 - \mu_1)^2/N(k\sigma_0^2 + (N - k)\sigma_1^2 + \epsilon)$;        $\triangleright$ equation 5
7: **end for**
8: $\Delta \leftarrow \Delta/\rho N$;
**Output:** EAG value $\Delta$ through different $k$;

---

---

**Algorithm 2** Expected Anomaly Gap Vectorized

---

**Input:** Percentile $\rho$, anomaly score vector $\boldsymbol{s}$;
1: $N \leftarrow \boldsymbol{s}.\text{size}; \Delta \leftarrow 0;$
2: Top $\rho N$ sort $\boldsymbol{s}$ in descending order;
3: $\boldsymbol{\Sigma} \leftarrow \texttt{PrefixSum}(\boldsymbol{s}); sum \leftarrow \boldsymbol{\Sigma}[-1]$
4: $\boldsymbol{n} \leftarrow [1, ..., N]; \boldsymbol{w_0} \leftarrow \boldsymbol{n}/N; \boldsymbol{w_1} \leftarrow 1 - \boldsymbol{w_1};$
5: $\boldsymbol{\mu_0} \leftarrow \boldsymbol{\Sigma}/\boldsymbol{n}; \boldsymbol{\mu_1} \leftarrow (sum - \boldsymbol{\Sigma})/\boldsymbol{n}.\text{reverse};$
6: $\boldsymbol{\Sigma}^2 \leftarrow \texttt{PrefixSum}(\boldsymbol{s} \odot \boldsymbol{s});$
7: $\boldsymbol{\sigma_0^2} \leftarrow (\boldsymbol{\Sigma}^2 - \boldsymbol{\Sigma} \odot \boldsymbol{\Sigma}/\boldsymbol{n})/(\boldsymbol{n} - 1 + \epsilon);$         ▷ Sample variance
8: $\boldsymbol{\sigma_1^2} \leftarrow (\boldsymbol{\Sigma}^2[-1] - \boldsymbol{\Sigma}^2 - (sum - \boldsymbol{\Sigma}) \odot (sum - \boldsymbol{\Sigma})/\boldsymbol{n}.\text{reverse})/(\boldsymbol{n}.\text{reverse} - 1 + \epsilon);$
9: $\sigma^2_{\text{inter}} \leftarrow \boldsymbol{w_0}\boldsymbol{w_1}(\boldsymbol{\mu_0} - \boldsymbol{\mu_1}) \odot (\boldsymbol{\mu_0} - \boldsymbol{\mu_1});$
10: $\sigma^2_{\text{intra}} \leftarrow \boldsymbol{w_0}\boldsymbol{\sigma_0^2} + \boldsymbol{w_1}\boldsymbol{\sigma_1^2};$
11: $\Delta \leftarrow \sigma^2_{\text{inter}}/\sigma^2_{\text{intra}};$
**Output:** $\texttt{mean}(\Delta[1 : \rho N]);$

---

# B  BOUND OF OVERLAP BETWEEN DISTRIBUTIONS OF $\mathcal{X}_{\text{TRN}}$ AND $\mathcal{X}_{\text{GEN}}$

Here we provide the bound of the overlap between the distribution of normal training data and the distribution of $\mathcal{X}_{\text{gen}}$. The theorem is proved in Appendix D.

**Theorem 3** *Let $p_{trn}$ be the density function of the training data and $p_{mix} = \sum_{i=1}^{\top} w_i \mathcal{N}(\boldsymbol{\mu}_i, \boldsymbol{\Sigma}_i)$ be a Gaussian mixture model to approximate $p_{trn}$ such that $|\log(p_{trn}(\boldsymbol{x})) - \log(p_{mix}(\boldsymbol{x}))| \leq \varepsilon$ holds for any $\boldsymbol{x} \in \mathbb{R}^d$. Denote $p_{gen}$ the density function $\mathcal{X}_{gen}$. Suppose $\underline{\lambda} \leq \lambda_{\min}(\boldsymbol{\Sigma}_i) \leq \lambda_{\max}(\boldsymbol{\Sigma}_i) \leq \bar{\lambda}$, $\underline{\phi} \leq \|\boldsymbol{\mu}_i\| \leq \bar{\phi}, \underline{\beta} \leq \|\boldsymbol{\Sigma}_i\|_* \leq \bar{\beta}$ for all $i \in [T]$. Then the distance between $p_{trn}$ and $p_{gen}$, measured by the KL-divergence, satisfies the following inequalities:*
*(a) $D_{KL}(p_{gen}\|p_{trn}) \leq \frac{1}{2T}\left(d\log\bar{\lambda} + \underline{\lambda}^{-1}\bar{\phi}^2 + d\underline{\lambda}^{-1}\right) - \frac{d}{2} + \varepsilon;$*
*(b) $D_{KL}(p_{trn}\|p_{gen}) \geq \exp(-\varepsilon)(-d\log\bar{\lambda} - d - 2(T-1)\underline{\lambda}^{-1}\bar{\phi} - (T-1)\underline{\lambda}^{-1}\bar{\beta} + T\underline{\phi} + T\underline{\beta}) - \varepsilon\exp(-\varepsilon).$*

# C  BOUND OF ERROR RATE OF NPD

In this section, we analyze the generalization error of our AutoUAD with the evaluation metric NPD using tool of the Rademacher complexity, which is defined as follows.

**Definition 7 (Rademacher Complexity)** *Let $H$ be a set of real-valued functions defined over a set $X$. Given a sample $S \in X^m$ and $\sigma = (\sigma_1, \ldots, \sigma_n)$, where $\sigma_i$s are independent uniform random variables taking values in $\{-1, +1\}$. The empirical Rademacher complexity of $H$ is defined as follows:*

$$\widehat{\mathfrak{R}}_S(H) = \frac{2}{m}\underset{\sigma}{\mathrm{E}}\left[\sup_{h \in H}\left|\sum_{i=1}^{m}\sigma_i h(x_i)\right| \mid S = (x_1, \ldots, x_m)\right].$$

To calculate the false positive rate (FPR) and false negative rate (FNR), we need to determine a threshold for the anomaly scores given by a model $\mathcal{M}$. For convenience, we give the following definitions.

**Definition 8** *Let $\tau_{\mathcal{M}}(z) = \mathbb{1}(z > c)$ be a threshold function, where $c > 0$ is the threshold. Let $F_{\mathcal{M}} = \tau_{\mathcal{M}} \circ f_{\mathcal{M}} : \mathbb{R}^d \to \{0, 1\}$ and $\mathcal{F}_{\mathcal{M}}$ be the class of $F_{\mathcal{M}}$ defined by $\mathcal{M}$ with hyperparameters $\Theta$. The FPR and FNR on the unseen testing data are then defined as $FPR = \mathbb{E}_{\mathbf{x} \sim \mathcal{D}_0}[F_{\mathcal{M}}(\mathbf{x})]$ and $FNR = \mathbb{E}_{\mathbf{x} \sim \mathcal{D}_1'}[1 - F_{\mathcal{M}}(\mathbf{x})]$ respectively.*

Without loss of generality and for convenience, we assume that in the problem defined by Definition 1, $N_1 = 0$, and in NPD, $M = N/2$. The following theorem (proved in Section D.4) provides a bound for the FPR and FNR on the unseen testing data.

**Theorem 4** *Based on Definition 8, letting $\Delta = \min\{\max_{\mathbf{x} \in \mathcal{X}_{gen}} f_{\mathcal{M}}(\mathbf{x}) - \min_{\mathbf{x} \in \mathcal{X}_{gen}} f_{\mathcal{M}}(\mathbf{x}), \max_{\mathbf{x} \in \mathcal{X}_{val}} f_{\mathcal{M}}(\mathbf{x}) - \min_{\mathbf{x} \in \mathcal{X}_{val}} f_{\mathcal{M}}(\mathbf{x})\}, \varsigma = \max_{\mathbf{x} \in \mathcal{X}_{gen}} f_{\mathcal{M}}(\mathbf{x}),$ and*

$\kappa = 1.2 + 2(c^{-1} - \frac{1}{5\varsigma}) \sum_{\mathbf{x} \in \mathcal{X}_{gen}} f_{\mathcal{M}}(\mathbf{x})/N$, *then over the randomness of $\mathcal{X}_{val}$ and $\mathcal{X}_{gen}$, the following inequality holds with probability at least $1 - 2\delta$:*

$$FPR + FNR \leq \kappa - \frac{2\Delta}{c\sqrt{N}} \sqrt{\mathcal{V}_{NPD}(\mathcal{M}, \mathcal{X})} + \frac{\sqrt{2}}{2} \sqrt{D_{KL}(\mathcal{D}_{gen} \| \mathcal{D}_1')}$$

$$+ \widehat{\mathfrak{R}}_{\mathcal{X}_{val}}(\mathcal{F}_{\mathcal{M}}) + \widehat{\mathfrak{R}}_{\mathcal{X}_{gen}}(\mathcal{F}_{\mathcal{M}}) + 6\sqrt{\frac{\log \frac{2}{\delta}}{N}} \tag{9}$$

In the theorem, the empirical Rademacher complexities $\widehat{\mathfrak{R}}_{\mathcal{X}_{val}}(\mathcal{F}_{\mathcal{M}})$ and $\widehat{\mathfrak{R}}_{\mathcal{X}_{gen}}(\mathcal{F}_{\mathcal{M}})$ can be explicitly bounded for any $\mathcal{M}$ (e.g., OC-SVM and AE) with any hyperparameters $\Theta$ and the corresponding technique is fairly standard in the literature (Bartlett & Mendelson, 2002; Bartlett et al., 2017). Due to this, together with the fact that our work AutoUAD is a framework not specialized to a single $\mathcal{M}$, we will not show the $\mathcal{M}$-specific computation of $\widehat{\mathfrak{R}}_{\mathcal{X}_{val}}(\mathcal{F}_{\mathcal{M}})$ and $\widehat{\mathfrak{R}}_{\mathcal{X}_{gen}}(\mathcal{F}_{\mathcal{M}})$. Note that $D_{KL}(\mathcal{D}_{gen} \| \mathcal{D}_1')$ can be further bounded by the similar approach used in Theorem 3. Our AutoUAD finds the model with the largest $\mathcal{V}_{NPD}$ and hence has the potential to reduce the false positive rate and false negative rate. In addition, a smaller $D_{KL}(\mathcal{D}_{gen} \| \mathcal{D}_1')$ or complexity of $\mathcal{M}$ (measured as $\widehat{\mathfrak{R}}_{\mathcal{X}_{val}}(\mathcal{F}_{\mathcal{M}})$ and $\widehat{\mathfrak{R}}_{\mathcal{X}_{gen}}(\mathcal{F}_{\mathcal{M}})$) may also lead to a lower error rate.

## D PROOF FOR THEOREMS

### D.1 PROOF FOR THEOREM 1

*Proof:* Let $g(x)$ be the density function of a Gaussian distribution with mean $\mu$ and variance $\sigma^2$ and let $p(x)$ be the density function of an arbitrary distribution with the same variance $\sigma^2$. Without loss of generality, we assume that $f(x)$ has the same mean of $\mu$ as $g(x)$, because differential entropy is translation invariant. Denote the entropy of $g$ and $f$ as $H(g)$ and $H(f)$ respectively. The Kullback-Leibler divergence between $g$ and $f$ satisfies

$$0 \leq D_{KL}(f \| g) = \int_{-\infty}^{\infty} f(x) \log \left( \frac{f(x)}{g(x)} \right) dx = -H(f) - \int_{-\infty}^{\infty} f(x) \log(g(x)) dx \tag{10}$$

We have

$$H(g) = - \int g(x) \log g(x) \mathrm{d}x$$
$$= -\mathbb{E} \left[ \log \mathcal{N} \left( \mu, \sigma^2 \right) \right]$$
$$= -\mathbb{E} \left[ \log \left[ (2\pi\sigma^2)^{-1/2} \exp \left( -\frac{1}{2\sigma^2}(x - \mu)^2 \right) \right] \right] \tag{11}$$
$$= \frac{1}{2} \log \left( 2\pi\sigma^2 \right) + \frac{1}{2\sigma^2} \mathbb{E} \left[ (x - \mu)^2 \right]$$
$$= \frac{1}{2} \log \left( 2\pi\sigma^2 \right) + \frac{1}{2}$$

Note that according to the basic definitions, $\int_{-\infty}^{\infty} f(x) dx = 1$ and $\int_{-\infty}^{\infty} f(x)(x - \mu)^2 dx = \sigma^2$. Then, for the second term of RHS of (10), we have

$$\int_{-\infty}^{\infty} f(x) \log(g(x)) dx = \int_{-\infty}^{\infty} f(x) \log \left( \frac{1}{\sqrt{2\pi\sigma^2}} e^{-\frac{(x-\mu)^2}{2\sigma^2}} \right) dx$$
$$= \int_{-\infty}^{\infty} f(x) \log \frac{1}{\sqrt{2\pi\sigma^2}} dx + \int_{-\infty}^{\infty} f(x) \left( -\frac{(x-\mu)^2}{2\sigma^2} \right) dx$$
$$= -\frac{1}{2} \log \left( 2\pi\sigma^2 \right) - \frac{\sigma^2}{2\sigma^2} \tag{12}$$
$$= -\frac{1}{2} \left( \log \left( 2\pi\sigma^2 \right) + 1 \right)$$
$$= -H(g)$$

Combining the results of (10) and (12), we obtain

$$H(g) - H(f) \geq 0. \tag{13}$$

This inequality indicates that with a given variance, the entropy of the Gaussian distribution is the largest amongst all distributions.

For $\mathcal{X}_{\text{trn}}$, we denote the density function of each variable as $f_i$, $i = 1, \ldots, d$. For $\mathcal{X}_{\text{gen}}$, we denote the density function of each variable as $g_i$, $i = 1, \ldots, d$. Since $g_1, g_2, \ldots, g_d$ or the corresponding variables more formally are independent, we have

$$H(g_1, g_2, \ldots, g_d) = \sum_{i=1}^{d} H(g_i). \tag{14}$$

Since $g_1, g_2, \ldots, g_d$ or the corresponding variables more formally are not independent almost surely, we have

$$H(f_1, f_2, \ldots, f_d) < \sum_{i=1}^{d} H(f_i). \tag{15}$$

Appying 13 to each $i$ and combing them with (14) and (15), we arrive at

$$H(g_1, g_2, \ldots, g_d) > H(f_1, f_2, \ldots, f_d). \tag{16}$$

We finish the proof by changing the format of entropy using variables rather than density functions. Q.E.D.

## D.2 Proof for Theorem 2

*Proof:*    For (a), we have the following derivation:

$$
\begin{aligned}
\mathcal{V}_{\text{NPD}}(\mathcal{M}, \mathcal{X}) =& \frac{(\text{Mean}(s_{\text{val}}) - \text{Mean}(s_{\text{gen}}))^2}{2(\text{Var}(s_{\text{val}}) + \text{Var}(s_{\text{gen}})) + \epsilon} \\
=& \frac{\left(\frac{|\mathcal{X}_{\text{val}}^0|}{M}\text{Mean}(s_{\text{val}}^0) + \frac{|\mathcal{X}_{\text{val}}^1|}{M}\text{Mean}(s_{\text{val}}^1) - \frac{|\mathcal{X}_{\text{gen}}^0|}{M}\text{Mean}(s_{\text{gen}}^0) - \frac{|\mathcal{X}_{\text{gen}}^1|}{M}\text{Mean}(s_{\text{gen}}^1)\right)^2}{2(\text{Var}(s_{\text{val}}) + \text{Var}(s_{\text{gen}})) + \epsilon} \\
\leq& \frac{2\left(\frac{|\mathcal{X}_{\text{val}}^1|}{M}\text{Mean}(s_{\text{val}}^1) - \frac{|\mathcal{X}_{\text{gen}}^0|}{M}\text{Mean}(s_{\text{gen}}^0)\right)^2 + 2\left(\frac{|\mathcal{X}_{\text{gen}}^1|}{M}\text{Mean}(s_{\text{gen}}^1) - \frac{|\mathcal{X}_{\text{val}}^0|}{M}\text{Mean}(s_{\text{val}}^0)\right)^2}{2(\text{Var}(s_{\text{val}}) + \text{Var}(s_{\text{gen}})) + \epsilon} \\
=& \frac{\left(\frac{|\mathcal{X}_{\text{val}}^1|}{M}\text{Mean}(s_{\text{val}}^1) - \frac{|\mathcal{X}_{\text{gen}}^0|}{M}\text{Mean}(s_{\text{gen}}^0)\right)^2 + \left(\frac{|\mathcal{X}_{\text{gen}}^0|}{M}\text{Mean}(s_{\text{gen}}^1) - \frac{|\mathcal{X}_{\text{val}}^0|}{M}\text{Mean}(s_{\text{val}}^0)\right)^2}{(\text{Var}(s_{\text{val}}) + \text{Var}(s_{\text{gen}})) + \epsilon/2}
\end{aligned}
\tag{17}
$$

For (b), let $c_0$ and $c_1$ be two arbitrary real numbers and let the transformed scoring function be $f' = c_1 f + c_0$. Then denote the anomaly score given by $f'$ as $s'$. We have the following derivation:

$$
\begin{aligned}
\mathcal{V}_{\text{NPD}}(f') =& \frac{(\text{Mean}(s'_{\text{val}}) - \text{Mean}(s'_{\text{gen}}))^2}{2(\text{Var}(s'_{\text{val}}) + \text{Var}(s'_{\text{gen}})) + \epsilon} \\
=& \frac{((c_1 \times \text{Mean}(s_{\text{val}}) + c_0) - (c_1 \times \text{Mean}(s_{\text{gen}}) + c_0))^2}{2(c_1^2 \times \text{Var}(s'_{\text{val}}) + c_1^2 \times \text{Var}(s_{\text{gen}})) + \epsilon} \\
=& \frac{c_1^2 \times (\text{Mean}(s_{\text{val}}) - \text{Mean}(s_{\text{gen}}))^2}{2c_1^2 \times (\text{Var}(s'_{\text{val}}) + \text{Var}(s_{\text{gen}})) + \epsilon}
\end{aligned}
\tag{18}
$$

When $\epsilon = 0$, we have $\mathcal{V}_{\text{NPD}}(f') = \mathcal{V}_{\text{NPD}}(f)$. This finished the proof.    Q.E.D.

## D.3 Proof for Theorem 3

*Proof:*    (a) The KL-divergence between $p_{\text{gen}}$ and $p_{\text{trn}}$

$$D_{KL}(p_{\text{gen}} \| p_{\text{trn}}) = \int_{\boldsymbol{x}} p_{\text{gen}}(\boldsymbol{x}) \log \frac{p_{\text{gen}}(\boldsymbol{x})}{p_{\text{trn}}(\boldsymbol{x})}, \tag{19}$$

where $p_{\text{gen}}(\boldsymbol{x}) = \frac{1}{(2\pi)^{d/2}|\boldsymbol{\Sigma}_{\text{gen}}|^{1/2}} \exp\left(-\frac{1}{2}(\boldsymbol{x}-\boldsymbol{\mu}_{\text{gen}})^\top \boldsymbol{\Sigma}^{-1}(\boldsymbol{x}-\boldsymbol{\mu}_{\text{gen}})\right)$. In addition, $|\log(p_{\text{trn}}(\boldsymbol{x})) - \log(p_{\text{mix}}(\boldsymbol{x}))| \leq \varepsilon$ holds for any $\boldsymbol{x}$, where $p_{mix}(\boldsymbol{x}) = \sum_{i=1}^\top w_i \frac{1}{(2\pi)^{d/2}|\boldsymbol{\Sigma}_i|^{1/2}} \exp\left(-\frac{1}{2}(\boldsymbol{x}-\boldsymbol{\mu}_i)^\top \boldsymbol{\Sigma}_i^{-1}(\boldsymbol{x}-\boldsymbol{\mu}_i)\right)$.

We have

$$
\begin{aligned}
D_{KL}(p_{\text{gen}}\|p_{\text{trn}}) =& \mathbb{E}_{p_{\text{gen}}}[\log(p_{\text{gen}}) - \log(p_{\text{trn}})] \\
\leq& \mathbb{E}_{p_{\text{gen}}}[\log(p_{\text{gen}}) - \log(p_{\text{mix}}) + \varepsilon)] \\
=& \varepsilon + D_{KL}(p_{\text{gen}}\|p_{\text{mix}})
\end{aligned}
\tag{20}
$$

Using Jensen's inequality, we obtain

$$
\begin{aligned}
D_{KL}(p_{\text{gen}}\|p_{\text{mix}}) =& \mathbb{E}_{p_{\text{gen}}}[\log(p_{\text{gen}}) - \log\sum_{i=1}^T w_i \mathcal{N}(\boldsymbol{\mu}_i, \boldsymbol{\Sigma}_i)] \\
\leq& \mathbb{E}_{p_{\text{gen}}}[\log(p_{\text{gen}}) - \sum_{i=1}^T w_i \log(\mathcal{N}(\boldsymbol{\mu}_i, \boldsymbol{\Sigma}_i))] \\
=& \sum_{i=1}^T \mathbb{E}_{p_{\text{gen}}}[\log(p_{\text{gen}}^{1/T} - \log(\mathcal{N}(\boldsymbol{\mu}_i, \boldsymbol{\Sigma}_i)^{w_i})]
\end{aligned}
\tag{21}
$$

For each $i$, it is not hard to show that

$$
\begin{aligned}
& \mathbb{E}_{p_{\text{gen}}}\left[\log(p_{\text{gen}}) - \log(\mathcal{N}(\boldsymbol{\mu}_i, \boldsymbol{\Sigma}_i)^{w_i})\right] \\
=& \mathbb{E}_{p_{\text{gen}}}\left[\frac{1}{2}\log\frac{|\boldsymbol{\Sigma}_i|^{w_i}}{|\boldsymbol{\Sigma}_0|^{1/T}} - \frac{1}{2T}(\mathbf{x}-\boldsymbol{\mu}_0)^\top \boldsymbol{\Sigma}_0^{-1}(\mathbf{x}-\boldsymbol{\mu}_0) + \frac{w_i}{2}(\mathbf{x}-\boldsymbol{\mu}_i)^\top \boldsymbol{\Sigma}_i^{-1}(\mathbf{x}-\boldsymbol{\mu}_i)\right] \\
=& \frac{1}{2}\log\frac{|\boldsymbol{\Sigma}_i|^{w_i}}{|\boldsymbol{\Sigma}_0|^{1/T}} - \frac{1}{2T}\mathbb{E}_{p_{\text{gen}}}\left[(\mathbf{x}-\boldsymbol{\mu}_0)^\top \boldsymbol{\Sigma}_0^{-1}(\mathbf{x}-\boldsymbol{\mu}_0)\right] + \frac{w_i}{2}\mathbb{E}_{p_{\text{gen}}}\left[(\mathbf{x}-\boldsymbol{\mu}_i)^\top \boldsymbol{\Sigma}_i^{-1}(\mathbf{x}-\boldsymbol{\mu}_i)\right] \\
=& \frac{1}{2}\log\frac{|\boldsymbol{\Sigma}_i|^{w_i}}{|\boldsymbol{\Sigma}_0|^{1/T}} - \frac{1}{2T}\text{tr}(\mathbb{E}_{p_{\text{gen}}}\left[(\mathbf{x}-\boldsymbol{\mu}_0)^\top(\mathbf{x}-\boldsymbol{\mu}_0)\right]\boldsymbol{\Sigma}_0^{-1}) + \frac{w_i}{2}(\boldsymbol{\mu}_0-\boldsymbol{\mu}_i)^\top \boldsymbol{\Sigma}_i^{-1}(\boldsymbol{\mu}_0-\boldsymbol{\mu}_i) \\
& + \frac{w_i}{2}\text{tr}\left\{\boldsymbol{\Sigma}_i^{-1}\boldsymbol{\Sigma}_0\right\} \\
=& \frac{1}{2}\left[\log\frac{|\boldsymbol{\Sigma}_i|^{w_i}}{|\boldsymbol{\Sigma}_0|^{1/T}} - \frac{d}{T} + w_i(\boldsymbol{\mu}_0-\boldsymbol{\mu}_i)^\top \boldsymbol{\Sigma}_i^{-1}(\boldsymbol{\mu}_0-\boldsymbol{\mu}_i) + w_i\text{tr}(\boldsymbol{\Sigma}_i^{-1}\boldsymbol{\Sigma}_0)\right]
\end{aligned}
\tag{22}
$$

Since $\boldsymbol{\mu}_0 = \mathbf{0}$, $\boldsymbol{\Sigma}_0 = \mathbf{I}_d$, it follows from (22) that

$$
\begin{aligned}
& \mathbb{E}_{p_{\text{gen}}}\left[\log(p_{\text{gen}}) - \log(\mathcal{N}(\boldsymbol{\mu}_i, \boldsymbol{\Sigma}_i)^{w_i})\right] \\
=& \frac{1}{2}\left[w_i\log|\boldsymbol{\Sigma}_i| - \frac{d}{T} + w_i\boldsymbol{\mu}_i^\top \boldsymbol{\Sigma}_i^{-1}\boldsymbol{\mu}_i + w_i\text{tr}(\boldsymbol{\Sigma}_i^{-1})\right]
\end{aligned}
\tag{23}
$$

Combining (21) and (23), and using $\lambda_{\min}(\boldsymbol{\Sigma}_i) \geq \underline{\lambda}$ and $\lambda_{\min}(\boldsymbol{\Sigma}_i) \leq \bar{\lambda}$, we have

$$
\begin{aligned}
D_{KL}(p_{\text{gen}}\|p_{\text{mix}}) \leq& -\frac{d}{2} + \frac{1}{2T}\sum_{i=1}^T\left[w_i\log|\boldsymbol{\Sigma}_i| + w_i\boldsymbol{\mu}^\top \boldsymbol{\Sigma}_i^{-1}\boldsymbol{\mu}_i + w_i\text{tr}(\boldsymbol{\Sigma}_i^{-1})\right] \\
\leq& -\frac{d}{2} + \frac{1}{2T}\sum_{i=1}^T w_i\left(d\log\bar{\lambda} + \underline{\lambda}^{-1}\|\boldsymbol{\mu}_i\|^2 + d\underline{\lambda}^{-1}\right) \\
\leq& -\frac{d}{2} + \frac{1}{2T}\sum_{i=1}^T w_i\left(d\log\bar{\lambda} + \underline{\lambda}^{-1}\max_j\|\boldsymbol{\mu}_j\|^2 + d\underline{\lambda}^{-1}\right) \\
=& -\frac{d}{2} + \frac{1}{2T}\left(d\log\bar{\lambda} + \underline{\lambda}^{-1}\max_j\|\boldsymbol{\mu}_j\|^2 + d\underline{\lambda}^{-1}\right) \\
\leq& -\frac{d}{2} + \frac{1}{2T}\left(d\log\bar{\lambda} + \underline{\lambda}^{-1}\bar{\phi}^2 + d\underline{\lambda}^{-1}\right)
\end{aligned}
\tag{24}
$$

Combining (24) and (20), we complete the proof.

(b) First, $|\log(p_{\text{trn}}(\boldsymbol{x})) - \log(p_{\text{mix}}(\boldsymbol{x}))| \leq \varepsilon$ implies that

$$\frac{p_{\text{trn}}(\boldsymbol{x})}{p_{\text{mix}}(\boldsymbol{x})} \leq \exp(\varepsilon) \quad \text{and} \quad \frac{p_{\text{mix}}(\boldsymbol{x})}{p_{\text{trn}}(\boldsymbol{x})} \leq \exp(\varepsilon). \tag{25}$$

Now we obtain

$$\begin{aligned}
D_{KL}(p_{\text{trn}}\|p_{\text{gen}}) &= \mathbb{E}_{p_{\text{trn}}}[\log(p_{\text{trn}}) - \log(p_{\text{gen}})] \\
&\geq \exp(-\varepsilon)\mathbb{E}_{p_{\text{mix}}}[\log(p_{\text{mix}}) - \log(p_{\text{gen}}) - \varepsilon] \\
&= \exp(-\varepsilon)D_{KL}(p_{\text{mix}}\|p_{\text{gen}}) - \varepsilon\exp(-\varepsilon)
\end{aligned} \tag{26}$$

For convenience, let $q_i := \mathcal{N}(\boldsymbol{\mu}_i, \boldsymbol{\Sigma}_i)$, then $p_{\text{mix}} = \sum_{i=1}^{\top} w_i q_i$. Using Jensen's inequality, we have

$$\begin{aligned}
D_{KL}(p_{\text{mix}}\|p_{\text{gen}}) &= \mathbb{E}_{p_{\text{mix}}}[\log(\sum_{i=1}^{T} w_i q_i) - \log(p_{\text{gen}})] \\
&\geq \mathbb{E}_{p_{\text{mix}}}[\sum_{i=1}^{T} w_i \log(q_i) - \log(p_{\text{gen}})] \\
&= \sum_{i=1}^{T} \mathbb{E}_{p_{\text{mix}}}[\log(q_i^{w_i}) - \log(p_{\text{gen}}^{1/T})]
\end{aligned} \tag{27}$$

For each $i$, we derive that

$$\begin{aligned}
&\mathbb{E}_{p_{\text{mix}}}[\log(q_i^{w_i}) - \log(p_{\text{gen}}^{1/T})] \\
={}&\mathbb{E}_{p_{\text{mix}}}\left[\frac{1}{2}\log\frac{|\boldsymbol{\Sigma}_0|^{1/T}}{|\boldsymbol{\Sigma}_i|^{w_i}} - \frac{w_i}{2}(\mathbf{x}-\boldsymbol{\mu}_i)^{\top}\boldsymbol{\Sigma}_i^{-1}(\mathbf{x}-\boldsymbol{\mu}_i) + \frac{1}{2T}(\mathbf{x}-\boldsymbol{\mu}_0)^{\top}\boldsymbol{\Sigma}_0^{-1}(\mathbf{x}-\boldsymbol{\mu}_0)\right] \\
={}&\frac{1}{2}\log\frac{|\boldsymbol{\Sigma}_0|^{1/T}}{|\boldsymbol{\Sigma}_i|^{w_i}} - \frac{w_i}{2}\mathbb{E}_{p_{\text{mix}}}\left[(\mathbf{x}-\boldsymbol{\mu}_i)^{\top}\boldsymbol{\Sigma}_i^{-1}(\mathbf{x}-\boldsymbol{\mu}_i)\right] + \frac{1}{2T}\mathbb{E}_{p_{\text{mix}}}\left[(\mathbf{x}-\boldsymbol{\mu}_0)^{\top}\boldsymbol{\Sigma}_0^{-1}(\mathbf{x}-\boldsymbol{\mu}_0)\right] \\
={}&\frac{1}{2}\log\frac{|\boldsymbol{\Sigma}_0|^{1/T}}{|\boldsymbol{\Sigma}_i|^{w_i}} - \frac{w_i}{2}w_i\text{tr}(\mathbb{E}_{q_i}\left[(\mathbf{x}-\boldsymbol{\mu}_i)^{\top}(\mathbf{x}-\boldsymbol{\mu}_i)\right]\boldsymbol{\Sigma}_i^{-1}) \\
&- \frac{w_i}{2}\sum_{j\neq i}w_j\mathbb{E}_{q_j}\left[(\mathbf{x}-\boldsymbol{\mu}_i)^{\top}\boldsymbol{\Sigma}_i^{-1}(\mathbf{x}-\boldsymbol{\mu}_i)\right] \\
&+ \frac{1}{2T}\sum_{j=1}^{T}w_j(\boldsymbol{\mu}_j-\boldsymbol{\mu}_0)^{\top}\boldsymbol{\Sigma}_0^{-1}(\boldsymbol{\mu}_j-\boldsymbol{\mu}_0) + \frac{1}{2T}\sum_{j=1}^{T}\text{tr}\{\boldsymbol{\Sigma}_0^{-1}\boldsymbol{\Sigma}_j\} \\
={}&\frac{1}{2}\left[\log\frac{|\boldsymbol{\Sigma}_0|^{1/T}}{|\boldsymbol{\Sigma}_i|^{w_i}} - w_i^2 d - w_i\sum_{j\neq i}(\boldsymbol{\mu}_j-\boldsymbol{\mu}_i)^{\top}\boldsymbol{\Sigma}_i^{-1}(\boldsymbol{\mu}_j-\boldsymbol{\mu}_i) - \frac{1}{T}\sum_{j\neq i}\text{tr}(\boldsymbol{\Sigma}_i^{-1}\boldsymbol{\Sigma}_j)\right. \\
&\left.+ \frac{1}{T}\sum_{j=1}^{T}(\boldsymbol{\mu}_j-\boldsymbol{\mu}_0)^{\top}\boldsymbol{\Sigma}_0^{-1}(\boldsymbol{\mu}_j-\boldsymbol{\mu}_0) + \frac{1}{T}\sum_{j=1}^{T}\text{tr}(\boldsymbol{\Sigma}_0^{-1}\boldsymbol{\Sigma}_j)\right] \\
={}&\frac{1}{2}\left[-w_i\log|\boldsymbol{\Sigma}_i| - w_i^2 d - w_i c_1 - \frac{1}{T}c_2 + \frac{1}{T}c_3 + \frac{1}{T}c_4\right],
\end{aligned} \tag{28}$$

where $c_1 = \sum_{j\neq i}(\boldsymbol{\mu}_j-\boldsymbol{\mu}_i)^{\top}\boldsymbol{\Sigma}_i^{-1}(\boldsymbol{\mu}_j-\boldsymbol{\mu}_i) \leq (T-1)\underline{\lambda}^{-1}\max_{ij}\|\boldsymbol{\mu}_j-\boldsymbol{\mu}_i\|$, $c_2 = \sum_{j\neq i}\text{tr}(\boldsymbol{\Sigma}_i^{-1}\boldsymbol{\Sigma}_j) \leq (T-1)\underline{\lambda}^{-1}\bar{\beta}$, $c_3 = \sum_{j=1}^{T}(\boldsymbol{\mu}_j-\boldsymbol{\mu}_0)^{\top}\boldsymbol{\Sigma}_0^{-1}(\boldsymbol{\mu}_j-\boldsymbol{\mu}_0) \geq T\min_j\|\boldsymbol{\mu}_j\|$, and $c_4 = \sum_{j=1}^{T}\text{tr}(\boldsymbol{\Sigma}_0^{-1}\boldsymbol{\Sigma}_j) \geq T\underline{\beta}$. By the way, $\log|\boldsymbol{\Sigma}_i| \leq d\log\bar{\lambda}$. It follows that

$$\begin{aligned}
D_{KL}(p_{\text{mix}}\|p_{\text{gen}}) &\geq \sum_{i=1}^{T}\left(-w_i d\log\bar{\lambda} - w_i^2 d - 2w_i(T-1)\underline{\lambda}^{-1}\bar{\phi} - (T-1)/T\underline{\lambda}^{-1}\bar{\beta} + \underline{\phi} + \underline{\beta}\right) \\
&\geq -d\log\bar{\lambda} - d - 2(T-1)\underline{\lambda}^{-1}\bar{\phi} - (T-1)\underline{\lambda}^{-1}\bar{\beta} + T\underline{\phi} + T\underline{\beta}
\end{aligned} \tag{29}$$

Finally, we have

$$D_{KL}(p_{\text{trn}}\|p_{\text{gen}}) \geq \exp(-\varepsilon)(-d\log\bar{\lambda} - d - 2(T-1)\underline{\lambda}^{-1}\bar{\phi} - (T-1)\underline{\lambda}^{-1}\bar{\beta} + T\underline{\phi} + T\underline{\beta}) - \varepsilon\exp(-\varepsilon).$$
(30)

Q.E.D.

### D.4 PROOF FOR THEOREM 4

The following is a standard Rademacher complexity bounds (Bartlett & Mendelson, 2002).

**Theorem 5 (Rademacher Bound)** *Let $H$ be a class of functions mapping $Z = X \times Y$ to $[0,1]$ and $S = (z_1, \ldots, z_m)$ a finite sample drawn i.i.d. according to a distribution $Q$. Then, for any $\delta > 0$, with probability at least $1 - \delta$ over samples $S$ of size $m$, the following inequality holds for all $h \in H$ :*

$$R(h) \leq \widehat{R}(h) + \widehat{\mathfrak{R}}_S(H) + 3\sqrt{\frac{\log\frac{2}{\delta}}{2m}}.$$

*Proof:* Since $\mathcal{X}_{\text{val}}$ are sampled from $\mathcal{D}_0$, using the standard Rademacher complexity bound given by Theorem 5, we can obtain

$$\mathbb{E}_{\mathbf{x}\sim\mathcal{D}_0}[F_{\mathcal{M}}(\mathbf{x})] \leq \frac{2}{N}\sum_{\mathbf{x}\in\mathcal{X}_{\text{val}}} F_{\mathcal{M}}(\mathbf{x}) + \widehat{\mathfrak{R}}_{\mathcal{X}_{\text{val}}}(\mathcal{F}_{\mathcal{M}}) + 3\sqrt{\frac{\log\frac{2}{\delta}}{N}},$$
(31)

where $|\mathcal{X}_{\text{val}}| = N/2$ and the inequality holds with probability at least $1 - \delta$.

For convenience, we let $g(\mathbf{x}) := 1 - F_{\mathcal{M}}(\mathbf{x})$ and let $p_1(\mathbf{x}), p_2(\mathbf{x})$ be the density functions of $\mathcal{D}_1', \mathcal{D}_{\text{gen}}$ respectively. We have

$$\begin{aligned}
\mathbb{E}_{\mathbf{x}\sim\mathcal{D}_1'}[g(\mathbf{x})] =& \mathbb{E}_{\mathbf{x}\sim\mathcal{D}_{\text{gen}}}[g(\mathbf{x})] + \mathbb{E}_{\mathbf{x}\sim\mathcal{D}_1'}[g(\mathbf{x})] - \mathbb{E}_{\mathbf{x}\sim\mathcal{D}_{\text{gen}}}[g(\mathbf{x})] \\
\leq& \mathbb{E}_{\mathbf{x}\sim\mathcal{D}_{\text{gen}}}[g(\mathbf{x})] + \left|\int g(\mathbf{x})p_1(\mathbf{x})d\mathbf{x} - \int g(\mathbf{x})p_2(\mathbf{x})d\mathbf{x}\right| \\
\leq& \mathbb{E}_{\mathbf{x}\sim\mathcal{D}_{\text{gen}}}[g(\mathbf{x})] + \int g(\mathbf{x})|p_1(\mathbf{x}) - p_2(\mathbf{x})|d\mathbf{x} \\
\leq& \mathbb{E}_{\mathbf{x}\sim\mathcal{D}_{\text{gen}}}[g(\mathbf{x})] + \int |p_1(\mathbf{x}) - p_2(\mathbf{x})|d\mathbf{x} \\
\leq& \mathbb{E}_{\mathbf{x}\sim\mathcal{D}_{\text{gen}}}[g(\mathbf{x})] + \frac{\sqrt{2}}{2}\sqrt{D_{\text{KL}}(\mathcal{D}_{\text{gen}}\|\mathcal{D}_1')},
\end{aligned}$$
(32)

where the last inequality used the fact that the total variation distance can be upper-bouned by using the KL-divergence.

Using Theorem 5 for $\mathbb{E}_{\mathbf{x}\sim\mathcal{D}_{\text{gen}}}[g(\mathbf{x})]$, we have

$$\mathbb{E}_{\mathbf{x}\sim\mathcal{D}_{\text{gen}}}[1 - F_{\mathcal{M}}(\mathbf{x})] \leq \frac{2}{N}\sum_{\mathbf{x}\in\mathcal{X}_{\text{gen}}}(1 - F_{\mathcal{M}}(\mathbf{x})) + \widehat{\mathfrak{R}}_{\mathcal{X}_{\text{gen}}}(\mathcal{F}_{\mathcal{M}}) + 3\sqrt{\frac{\log\frac{2}{\delta}}{N}},$$
(33)

where $|\mathcal{X}_{\text{gen}}| = N/2$ and the inequality holds with probability at least $1 - \delta$.

Combining (31) and (33), we have

$$\begin{aligned}
\text{FPR} + \text{FNR} \leq& \frac{2}{N}\sum_{\mathbf{x}\in\mathcal{X}_{\text{val}}} F_{\mathcal{M}}(\mathbf{x}) + \frac{2}{N}\sum_{\mathbf{x}\in\mathcal{X}_{\text{gen}}}(1 - F_{\mathcal{M}}(\mathbf{x})) + \frac{\sqrt{2}}{2}\sqrt{D_{\text{KL}}(\mathcal{D}_{\text{gen}}\|\mathcal{D}_1')} \\
& + \widehat{\mathfrak{R}}_{\mathcal{X}_{\text{val}}}(\mathcal{F}_{\mathcal{M}}) + \widehat{\mathfrak{R}}_{\mathcal{X}_{\text{gen}}}(\mathcal{F}_{\mathcal{M}}) + 6\sqrt{\frac{\log\frac{2}{\delta}}{N}}
\end{aligned}$$
(34)

which holds with probability at least $1 - 2\delta$.

As $\tau_{\mathcal{M}}$ is a threshold function, we have

$$F_{\mathcal{M}}(\mathbf{x}) = \frac{\text{sgn}(f(\mathbf{x}) - c) + 1}{2}, \tag{35}$$

where $c$ is a threshold defined by the model $\mathcal{M}$ with hyperparameters $\Theta$. It follows that

$$F_{\mathcal{M}}(\mathbf{x}) \leq c^{-1} f_{\mathcal{M}}(\mathbf{x}) \tag{36}$$

and

$$1 - F_{\mathcal{M}}(\mathbf{x}) \leq 1 + \vartheta - \frac{\vartheta}{\varsigma} f_{\mathcal{M}}(\mathbf{x}) \tag{37}$$

where $\vartheta$ could be any value and $\varsigma = \max_{\mathbf{x} \in \mathcal{X}_{\text{gen}}} f_{\mathcal{M}}(\mathbf{x})$.

Now we can derive that

$$
\begin{aligned}
&\frac{2}{N} \sum_{\mathbf{x} \in \mathcal{X}_{\text{val}}} F_{\mathcal{M}}(\mathbf{x}) + \frac{2}{N} \sum_{\mathbf{x} \in \mathcal{X}_{\text{gen}}} (1 - F_{\mathcal{M}}(\mathbf{x})) \\
\leq& \frac{2}{N} \sum_{\mathbf{x} \in \mathcal{X}_{\text{val}}} c^{-1} f_{\mathcal{M}}(\mathbf{x}) + \frac{2}{N} \sum_{\mathbf{x} \in \mathcal{X}_{\text{gen}}} (1 + \vartheta - \frac{\vartheta}{\varsigma} f_{\mathcal{M}}(\mathbf{x})) \\
\leq& c^{-1} \left( \frac{2}{N} \sum_{\mathbf{x} \in \mathcal{X}_{\text{val}}} f_{\mathcal{M}}(\mathbf{x}) - \frac{2}{N} \sum_{\mathbf{x} \in \mathcal{X}_{\text{gen}}} f_{\mathcal{M}}(\mathbf{x}) \right) \\
&+ \frac{2}{N} \sum_{\mathbf{x} \in \mathcal{X}_{\text{gen}}} \left( 1 + \vartheta + (c^{-1} - \frac{\vartheta}{\varsigma}) f_{\mathcal{M}}(\mathbf{x}) \right)
\end{aligned} \tag{38}
$$

Recall that $\frac{2}{N} \sum_{\mathbf{x} \in \mathcal{X}_{\text{val}}} f_{\mathcal{M}}(\mathbf{x}) = \text{mean}(\mathbf{s}_{\text{val}})$ and $\frac{2}{N} \sum_{\mathbf{x} \in \mathcal{X}_{\text{gen}}} f_{\mathcal{M}}(\mathbf{x}) = \text{mean}(\mathbf{s}_{\text{gen}})$ and $\text{mean}(\mathbf{s}_{\text{val}}) \leq \text{mean}(\mathbf{s}_{\text{gen}})$, it follows from (39) that

$$
\begin{aligned}
&\frac{2}{N} \sum_{\mathbf{x} \in \mathcal{X}_{\text{val}}} F_{\mathcal{M}}(\mathbf{x}) + \frac{2}{N} \sum_{\mathbf{x} \in \mathcal{X}_{\text{gen}}} (1 - F_{\mathcal{M}}(\mathbf{x})) \\
\leq& -c^{-1} \sqrt{(\text{mean}(\mathbf{s}_{\text{gen}}) - \text{mean}(\mathbf{s}_{\text{var}}))^2} \\
&+ \frac{2}{N} \sum_{\mathbf{x} \in \mathcal{X}_{\text{gen}}} \left( 1 + \vartheta + (c^{-1} - \frac{\vartheta}{\varsigma}) f_{\mathcal{M}}(\mathbf{x}) \right) \\
\leq& -c^{-1} \sqrt{\varrho \mathcal{V}_{\text{NPD}}(\mathcal{M}, \mathcal{X})} + \kappa
\end{aligned} \tag{39}
$$

where $\varrho = 2(\text{Var}(\mathbf{s}_{\text{gen}}) + \text{Var}(\mathbf{s}_{\text{val}})) + \epsilon$ and $\kappa = \frac{2}{N} \sum_{\mathbf{x} \in \mathcal{X}_{\text{gen}}} \left( 1 + \vartheta + (c^{-1} - \frac{\vartheta}{\varsigma}) f_{\mathcal{M}}(\mathbf{x}) \right) = 1 + \vartheta + (c^{-1} - \frac{\vartheta}{\varsigma}) \text{mean}(\mathbf{s}_{\text{gen}})$.

Based on the von-Szokefalvi-Nagy's inequality, we have

$$
\begin{aligned}
\varrho \geq& \frac{2((\max(\mathbf{s}_{\text{gen}}) - \min(\mathbf{s}_{\text{gen}}))^2 + (\max(\mathbf{s}_{\text{val}}) - \min(\mathbf{s}_{\text{val}}))^2)}{N} + \epsilon \\
\geq& \frac{4\Delta^2}{N} + \epsilon \geq \frac{4\Delta^2}{N}
\end{aligned} \tag{40}
$$

where $\Delta = \min\{\max(\mathbf{s}_{\text{gen}}) - \min(\mathbf{s}_{\text{gen}}), \max(\mathbf{s}_{\text{val}}) - \min(\mathbf{s}_{\text{val}})\}$.

Combining the above result with (39) and (34) and letting $\vartheta = 0.2$ we obtain

$$
\begin{aligned}
\text{FPR} + \text{FNR} \leq& \kappa - \frac{2\Delta}{c\sqrt{N}} \sqrt{\mathcal{V}_{\text{NPD}}(\mathcal{M}, \mathcal{X})} + \frac{\sqrt{2}}{2} \sqrt{D_{\text{KL}}(\mathcal{D}_{\text{gen}} || \mathcal{D}'_1)} \\
&+ \widehat{\mathfrak{R}}_{\mathcal{X}_{\text{val}}}(\mathcal{F}_{\mathcal{M}}) + \widehat{\mathfrak{R}}_{\mathcal{X}_{\text{gen}}}(\mathcal{F}_{\mathcal{M}}) + 6\sqrt{\frac{\log \frac{2}{\delta}}{N}}
\end{aligned} \tag{41}
$$

This completed the proof.

$$\text{Q.E.D.}$$

Table 3: 38 real-world tabular datasets tested. * represent the dataset is from DAMICampos et al. (2016)

| Dataset | # Sample | Dim. | % Anomaly |
|---|---|---|---|
| ALOI | 49534 | 27 | 3.04 |
| anthyroid | 7200 | 6 | 7.42 |
| arrhythmia* | 452 | 274 | 14.60 |
| breastw | 683 | 9 | 34.99 |
| cardio | 1831 | 21 | 9.61 |
| Cardiotocography | 2114 | 21 | 22.04 |
| fault | 1941 | 27 | 34.67 |
| glass | 214 | 9 | 42.21 |
| Hepatitis | 80 | 17 | 16.25 |
| InternetAds | 1966 | 1555 | 18.72 |
| Ionosphere | 351 | 32 | 35.90 |
| landsat | 6435 | 36 | 20.71 |
| letter | 1600 | 32 | 15.10 |
| Lymphography | 148 | 18 | 4.05 |
| magic | 19020 | 10 | 35.16 |
| mammography | 11183 | 6 | 2.32 |
| mnist | 7603 | 100 | 9.21 |
| musk | 3062 | 166 | 3.17 |
| optdigits | 5216 | 64 | 2.88 |
| PageBlocks | 5393 | 10 | 9.46 |
| pendigits | 6870 | 16 | 2.27 |
| Pima | 768 | 8 | 34.90 |
| satellite | 6435 | 36 | 31.64 |
| satimage-2 | 5803 | 36 | 1.22 |
| shuttle* | 1013 | 9 | 1.28 |
| SpamBase | 4207 | 57 | 39.91 |
| speech | 3686 | 400 | 1.65 |
| Stamps | 340 | 9 | 9.12 |
| thyroid | 3772 | 6 | 2.47 |
| vertebral | 240 | 6 | 12.50 |
| vowels | 1456 | 12 | 3.43 |
| Waveform | 3443 | 21 | 2.90 |
| WBC | 223 | 9 | 4.48 |
| WDBC | 367 | 30 | 2.72 |
| Wilt | 4819 | 5 | 5.33 |
| wine | 129 | 13 | 7.75 |
| WPBC | 198 | 33 | 23.74 |
| yeast | 1484 | 8 | 34.16 |

## E   DATASETS SUMMARY

We conducted our experiment under the UAD dataset setting with 38 benchmark datasets commonly used in UAD research. The dataset information is shown in Table 3. Similar to (Shenkar & Wolf, 2022), we randomly split 50% of normal samples for training and used the rest with anomalous data for testing. All data are standardized using the training set's mean and standard deviation. The split of each dataset is the same across different UAD methods. We repeat all experiments with 5 different data splits and report the results with mean and standard deviation.

## F   AUTOUAD VIA BAYESIAN OPTIZATION

Bayesian optimization (BO) (Jones et al., 1998) has emerged as a powerful tool for hyper-parameter optimization in supervised learning (Snoek et al., 2012; Klein et al., 2017). BO optimizes hyper-parameters sequentially by leveraging historical search results to guide the next iteration. Com-

pared with grid search, BO offers significant advantages, especially for models with many hyper-parameters.

Given a black-box function $h : \mathcal{Z} \rightarrow \mathbb{R}$, BO aims to find an optimal point $z^* \in \mathcal{Z}$ that minimizes $h$ globally, and typically proceeds through three steps. First, BO identifies the most promising point $z_{t+1} \in \arg\max_z \alpha_{p(h)}(z)$ using numerical optimization, where $\alpha_{p(h)}$ is an acquisition function (e.g., Expected Improvement) that depends on a prior $p(h)$ (e.g., Gaussian processes (Williams & Rasmussen, 2006)). Next, BO evaluates the potentially expensive and noisy function $y_{t+1} \sim h(z_{t+1}) + \mathcal{N}(0, \sigma^2)$ and updates the observation set $D_t = \{(z_1, z_1), \ldots, (z_t, y_t)\}$ with the new sample $(z_{t+1}, y_{t+1})$. Finally, it updates both the prior $p(h)$ and the acquisition function $\alpha_{p(h)}$ using the updated dataset $D_{t+1}$, allowing the process to iterate toward the optimal solution.

We sequentially maximize the proposed evaluation metrics via BO instead of defining a hyper-parameter grid in a grid search. Suppose we have a set of different models for a UAD method $\mathcal{M}$, i.e., $\mathcal{F}_{\mathcal{M}} = \{f_{\mathcal{M};\Theta_1}, f_{\mathcal{M};\Theta_2}, \ldots, f_{\mathcal{M};\Theta_H}\}$, where $\Theta_i$ is the hyper-parameters in $f_{\mathcal{M}_i}$. Let

$$h_i(\Theta_i) := -\Delta(f_{\mathcal{M}}(\mathcal{X}|\Theta_i)), \ \ i = 1, 2, \ldots, H,$$

where $\Delta$ can RTM, EAG, or NPD, and $\mathcal{X}$ denotes the dataset (either training set or validation set). Then we use BO to find

$$\Theta_i^* = \arg\min_{\Theta_i \in \mathcal{S}_i} h_i(\Theta_i),$$

where $\mathcal{S}_i$ denotes the set of constraints. We can find the best model for each UAD method. Finally, we get the best model with its best hyper-parameters using equation 2. we use Tree-structured Parzen Estimator (TPE) (Bergstra et al., 2011). Then, the Expected Improvement (EI) acquisition function is

$$EI_{y^*}(\Theta) = \int_{-\infty}^{y^*} (y^* - y) p(y|\Theta) dy = \int_{-\infty}^{y^*} (y^* - y) \frac{p(\Theta|y)p(y)}{p(\Theta)} dy,$$

where $y = h(\theta)$, and $y^* = h_{\min}$ is the best function value known. Let $\gamma = p(y < y^*)$ and $p(\Theta) = \int_{\mathbb{R}} p(\Theta|y)p(y) dy = \gamma \ell(\Theta) + (1 - \gamma)\varphi(\Theta)$. We have

$$EI_{y^*}(\Theta) = \frac{\gamma y^*(\Theta) - \gamma \ell(\Theta) - \int_{-\infty}^{y^*} p(y) dy}{\gamma \ell(\Theta) + (1-\gamma)\varphi(\Theta)} \propto \left(\gamma + \frac{\varphi(\Theta)}{\ell(\Theta)}(1 - \gamma)\right)^{-1},$$

where $\ell(\Theta)$ is the density formed by using the observations $\{\Theta^{(i)}\}$ such that the corresponding loss $h(\Theta^{(i)})$ was less than $y^*$, and $\varphi(\Theta)$ is the density formed by using the remaining observations.

## G  IMPLEMENTATION DETAILS

**Implementation**  All experiments are implemented by Pytorch (Paszke et al., 2017) on NVIDIA RTX 3090 and AMD Ryzen Threadripper 3990X platform. For OCSVM, AE, and DeepSVDD, we utilize popular PyOD implementation (Zhao et al., 2019). For DPAD, we use the code provided by the authors. We utilize optuna (Akiba et al., 2019) to implement the Bayesian optimization with Tree-structured Parzen Estimator (TPE) sampler (Bergstra et al., 2011). For each run, we perform 500 searches. We consider the core hyper-parameters of each UAD algorithm, they are listed in Table 5. For MV/EM, MC, and HITS baseline, we utilize implementation in (Ma et al., 2023) [4]. For Max baseline, the highest performance among all observations is reported. For Random baseline, the grid search pool is listed in Table 4. For NPD, we set the size of validation data as $M = 0.3N$. For all $\epsilon$, we set it as $1 \times 10^{-9}$.

## H  HYPER-PARAMETERS OF UAD ALGORITHMS

We first describe each hyper-parameter of 4 UAD algorithms studied in this paper. For deep learning-based algorithms, we only consider the core hyper-parameters, such as hidden dimension, regularization, etc, because deep UAD algorithms are usually not sensitive to optimization's hyper-parameters (learning rate, batch size, etc.).

---

[4]https://github.com/yzhao062/uoms

Table 4: UAD Algorithms studied for hyper-parameter sensitivity and grid search pool construction. We search for two core hyper-parameters for each algorithm while keeping the other hyper-parameters as default. 2667 UAD models are constructed in the pool.

| UAD Algorithms | Hyper-parameters Searched [Grid Values] | # models |
|---|---|---|
| OCSVM | $\nu : [0.01, 0.02, ..., 1], \gamma : [100, 50, 10, 5, 1, 0.5, 0.1, 0.01, ..., 10^{-6}]$ | 1500 |
| AutoEncoder | weight decay: $[1, 0.5, 0.1, 0.01, ..., 10^{-6}]$, hidden dim.: $[16, 24, 32, ..., 256]]$ | 403 |
| DeepSVDD | $\lambda : [0.9, 0.5, 0.1, 0.01, ..., 10^{-6}]$, hidden dim.: $[16, 24, 32, ..., 256]$ | 403 |
| DPAD | $\lambda : [1000, 500, 100, 50, ..., 10^{-6}], \gamma : [1000, 500, 100, 50, ..., 10^{-6}]$ | 361 |

Table 5: Hyper-parameter Searched in Bayesian Optimization.

| UAD Algorithms | Hyper-parameters Searched (Value Range) |
|---|---|
| OCSVM | kernel, $\nu(0, 1), \gamma(1e - 6, 100), \alpha_0(0, 1000)$ |
| AutoEncoder | weight decay$(1e - 6, 0.1)$, hidden dim. $1(16, 256)$, hidden dim $2(16, 256)$ |
| DeepSVDD | $\lambda(1e - 6, 1)$, hidden dim. $1(16, 256)$, hidden dim. $2(16, 256)$ |
| DPAD | $\lambda(1e - 6, 1000), k(3, 1000), \gamma(1e - 6, 100)$ hidden dim. $1(16, 256)$, hidden dim $2(16, 256)$ |

- OCSVM: We repeat the description in Scikit-Learn(Pedregosa et al., 2011). $\nu$: An upper bound on the fraction of training errors and a lower bound on the fraction of support vectors. $\gamma$: Kernel coefficient for rbf, poly, and sigmoid. $\alpha_0$: Independent term in kernel function. It is only significant in poly and sigmoid. *Kernel*: kernel type to be used in the algorithm.

- AE: *Hidden dim. 1*: the width of the first and the last hidden layer of the neural network. *Hidden dim. 2*: the width of the second and the penultimate hidden layer of the neural network, which is known as the size of the hidden representation. *Weight decay*: the regularization term to control model complexity.

- DeepSVDD: We do not use an AE for pre-training to save time. $c$ is computed based on the network initialization first forward pass. *Hidden dim. 1*: the width of the first and the last hidden layer of the neural network. *Hidden dim. 2*: the width of the second and the penultimate hidden layer of the neural network, which is known as the size of the hidden representation. $\lambda$: weight decay regularizer on the network parameters.

- DPAD: *Hidden dim. 1*: the width of the first and the last hidden layer of the neural network. *Hidden dim. 2*: the width of the second and the penultimate hidden layer of the neural network, which is known as the size of the hidden representation. $\lambda$: weight decay regularizer on the network parameters. $\gamma$: hyper-parameter in dense projection distance calculation. $k$: k nearest neighbors used in finding anomaly score.

Unless specified we train deep UAD methods with 256 batch size, Adam optimizer, 0.001 learning rate, and 200 epochs. For AE we train 100 epochs. For DPAD we use a larger batch size of $4096$.

We list the hyper-parameters searched for grid search and UOMS model pool construction in Table 4. We list the hyper-parameters considered in BO in Table 5.

## I   DETAILED RESULTS FOR EACH DATASET

Table 1 in Section 4 reports the average AUC and F1 results through 38 datasets. We list the results of each dataset in Table 6 and Table 7 for AUC and F1, respectively.

## J   BO RESULTS ON MORE UAD ALGORITHMS

We perform AutoUAD via BO on five additional conventional deep UAD algorithms, including PLAD(Cai & Fan, 2022), NeuTraLAD(Qiu et al., 2021), HRN(Hu et al., 2020), DROCC(Goyal et al., 2020), and SCAD(Shenkar & Wolf, 2022). Due to time constraints, we test them on 14 datasets with smaller sizes (number of samples less than 1000), including arrhythmia, breastw, glass, Hepatitis, Ionosphere, Lymphography Pima, shuttle, Stamps, vertebral, WBC, WDBC, wine, and WPBC. The results compared with Max, EM/MV, RTM, EAG, and NPD are reported in Table 8. It is seen our metrics work well with complex deep UAD methods.

Table 6: AUC (%) score of EM/MV, EAG, NPD, and RTM on 38 benchmark datasets with 4 UAD methods. Average results are reported through 5 different random data splits.

| ad methods | AE | | | | DPAD | | | | DeepSVDD | | | | OCSVM | | | |
|---|---|---|---|---|---|---|---|---|---|---|---|---|---|---|---|---|
| delta method | EM/MV | EAG | NPD | RTM | EM/MV | EAG | NPD | RTM | EM/MV | EAG | NPD | RTM | EM/MV | EAG | NPD | RTM |
| ALOI | 56.08±0.4 | 56.08±0.1 | 55.86±0.5 | 55.78±0.4 | 51.40±2.3 | 52.30±0.7 | 58.69±1.0 | 55.18±1.0 | 54.58±1.1 | 54.85±0.8 | 54.88±1.1 | 54.90±0.3 | 55.25±0.3 | 54.72±0.1 | 61.43±1.2 | 54.91±0.1 |
| Cardiotocography | 73.01±2.6 | 78.35±1.6 | 73.38±2.3 | 70.37±3.4 | 58.93±3.2 | 63.90±14.1 | 70.20±4.4 | 69.02±5.7 | 85.05±4.2 | 79.55±2.9 | 79.09±8.5 | 63.97±7.9 | 78.90±11.4 | 63.47±11.7 | 80.53±1.9 | 84.40±0.5 |
| Hepatitis | 78.74±4.3 | 79.58±4.1 | 80.47±2.3 | 79.81±3.8 | 67.92±17.6 | 67.53±13.5 | 69.23±3.8 | 81.28±3.6 | 78.09±4.6 | 77.86±7.7 | 70.21±11.2 | 78.97±7.2 | 73.36±13.9 | 65.95±13.2 | 78.93±3.2 | 79.54±4.2 |
| InternetAds | 88.59±1.2 | 88.99±0.8 | 87.94±0.5 | 88.91±0.7 | 74.02±16.6 | 83.84±8.3 | 79.17±3.1 | 83.81±6.4 | 88.12±1.0 | 87.64±0.7 | 87.53±0.7 | 88.03±0.9 | 65.96±19.7 | 57.32±16.4 | 85.40±0.5 | 86.72±0.7 |
| Ionosphere | 95.64±0.6 | 96.12±0.9 | 96.30±0.6 | 96.35±0.4 | 91.67±6.3 | 81.18±5.3 | 95.89±1.3 | 84.30±7.7 | 83.84±7.6 | 81.33±5.8 | 87.92±4.8 | 81.70±2.4 | 85.99±9.9 | 76.53±16.7 | 96.99±0.8 | 91.11±2.9 |
| Lymphography | 98.95±1.0 | 99.09±0.8 | 98.52±1.4 | 99.14±0.8 | 99.00±0.4 | 98.76±0.8 | 97.95±1.4 | 98.95±0.8 | 98.95±0.8 | 98.43±1.1 | 98.95±0.6 | 98.95±0.9 | 90.14±18.5 | 99.00±0.9 | 98.71±0.8 | 99.14±0.8 |
| PageBlocks | 94.97±1.1 | 92.72±1.1 | 95.98±1.3 | 93.20±0.4 | 77.73±13.4 | 50.85±9.7 | 95.22±1.3 | 86.43±4.0 | 91.33±1.4 | 90.58±2.3 | 94.01±1.6 | 90.32±2.0 | 86.19±20.3 | 94.33±2.4 | 94.95±0.3 | 93.46±0.4 |
| Pima | 68.69±1.0 | 68.81±2.7 | 70.59±1.1 | 72.80±1.9 | 66.36±5.5 | 64.27±4.6 | 66.72±2.9 | 61.93±5.1 | 70.91±1.9 | 68.72±2.9 | 68.97±1.6 | 69.03±9.5 | 65.72±8.3 | 65.01±5.4 | 72.15±1.4 | 68.63±0.9 |
| SpamBase | 80.90±1.5 | 80.94±1.0 | 82.95±2.1 | 80.64±1.2 | 48.90±18.7 | 58.00±18.4 | 65.85±20.3 | 71.73±11.3 | 78.86±2.7 | 77.10±2.8 | 77.99±2.8 | 77.02±2.1 | 70.66±23.7 | 78.89±1.4 | 83.27±1.0 | 80.06±1.0 |
| Stamps | 93.72±2.7 | 93.18±1.6 | 94.48±1.4 | 93.59±2.2 | 81.88±16.5 | 90.58±5.2 | 89.16±5.2 | 92.89±3.4 | 92.10±3.2 | 90.77±4.5 | 86.95±9.3 | 92.34±1.9 | 90.83±7.9 | 88.48±13.4 | 95.14±1.3 | 92.94±2.3 |
| WBC | 97.70±0.9 | 98.89±0.5 | 97.96±1.2 | 99.04±0.6 | 96.40±2.5 | 98.87±0.6 | 97.19±0.9 | 98.78±1.2 | 99.28±0.2 | 99.34±0.2 | 97.96±1.6 | 99.26±0.3 | 98.47±1.0 | 99.15±0.2 | 98.44±0.6 | 99.25±0.1 |
| WDBC | 98.63±0.5 | 98.01±1.0 | 98.65±0.4 | 96.32±2.5 | 83.28±19.3 | 92.24±13.5 | 95.66±2.4 | 85.65±17.6 | 98.57±1.7 | 95.01±3.6 | 98.15±1.8 | 95.18±2.0 | 77.46±43.2 | 67.76±22.2 | 99.08±0.5 | 99.18±0.2 |
| WPBC | 52.23±3.6 | 49.61±2.8 | 52.56±5.4 | 50.72±2.1 | 48.89±2.5 | 48.42±4.3 | 48.41±6.4 | 50.27±2.5 | 48.74±2.5 | 52.94±4.5 | 50.20±5.6 | 49.54±5.5 | 45.73±5.6 | 52.13±2.0 | 50.37±4.0 | 48.70±2.5 |
| Waveform | 66.43±2.8 | 69.12±3.6 | 64.95±2.1 | 66.75±2.9 | 65.12±9.6 | 60.83±3.1 | 65.82±5.2 | 57.94±5.4 | 68.36±1.7 | 66.73±6.3 | 59.61±10.4 | 64.58±7.6 | 51.25±17.8 | 67.20±5.2 | 77.53±0.6 | 54.18±0.6 |
| Wilt | 71.99±9.8 | 74.60±9.6 | 79.76±3.2 | 69.85±4.9 | 56.11±10.1 | 60.77±5.7 | 76.77±4.0 | 39.04±6.2 | 35.91±7.7 | 42.30±8.6 | 36.33±6.1 | 29.01±5.6 | 48.57±20.1 | 72.15±2.0 | 74.95±2.7 | 33.42±0.5 |
| annthyroid | 89.27±3.3 | 85.37±1.5 | 91.25±2.1 | 87.25±5.1 | 68.29±19.9 | 58.49±17.5 | 89.44±1.5 | 78.21±7.6 | 63.65±4.0 | 70.50±9.2 | 77.64±6.3 | 85.04±3.1 | 69.04±10.9 | 63.35±12.3 | 75.81±1.4 | 75.77±1.7 |
| arrhythmia | 75.84±1.6 | 76.23±1.9 | 74.95±1.6 | 76.13±1.4 | 67.23±9.6 | 71.06±1.3 | 70.63±1.9 | 71.84±3.5 | 73.49±2.8 | 72.25±1.5 | 74.65±2.4 | 74.94±2.1 | 69.04±10.9 | 63.35±12.3 | 75.81±1.4 | 75.77±1.7 |
| breastw | 98.76±0.9 | 98.86±0.6 | 98.74±0.6 | 99.06±0.7 | 83.86±26.3 | 97.52±2.2 | 97.54±1.2 | 98.56±0.9 | 98.86±0.9 | 98.69±0.7 | 98.69±1.0 | 98.96±0.6 | 98.84±0.8 | 95.80±6.7 | 99.64±0.7 | 98.82±0.8 |
| cardio | 94.75±1.8 | 95.80±1.3 | 93.21±2.5 | 93.92±0.8 | 76.00±17.8 | 89.57±6.7 | 89.23±2.2 | 73.74±6.6 | 95.72±0.6 | 94.35±3.7 | 93.92±5.0 | 89.10±4.3 | 87.77±11.1 | 89.12±16.6 | 96.15±0.6 | 96.80±0.4 |
| fault | 74.08±1.7 | 65.63±2.7 | 76.16±1.3 | 72.00±6.0 | 67.85±10.5 | 59.09±11.7 | 78.12±0.5 | 64.59±3.8 | 58.73±4.2 | 50.45±4.6 | 58.74±5.5 | 47.08±4.2 | 66.05±8.3 | 62.75±1.8 | 71.13±1.6 | 56.08±1.5 |
| glass | 84.14±5.4 | 84.42±4.3 | 84.45±5.7 | 81.87±2.6 | 80.34±13.0 | 77.67±12.3 | 86.19±10.2 | 78.30±7.4 | 66.51±5.1 | 73.97±8.7 | 61.11±12.1 | 75.69±5.5 | 64.62±24.3 | 81.33±6.4 | 87.16±1.9 | 60.33±16.5 |
| landsat | 58.86±4.8 | 53.58±3.9 | 64.01±1.2 | 58.25±2.0 | 64.44±8.5 | 52.86±6.6 | 70.70±2.7 | 62.20±4.5 | 44.47±3.2 | 46.91±2.9 | 43.20±5.9 | 42.50±5.6 | 54.34±17.0 | 45.61±0.4 | 59.83±1.3 | 39.33±0.8 |
| letter | 86.26±3.3 | 79.76±4.7 | 91.32±0.7 | 86.54±1.9 | 66.74±15.1 | 54.41±13.7 | 80.76±2.0 | 60.33±5.0 | 53.06±1.4 | 52.23±5.8 | 52.73±3.7 | 52.40±5.9 | 58.64±15.4 | 85.92±13.6 | 86.50±1.6 | 54.70±7.1 |
| magic.gamma | 82.71±3.5 | 78.40±2.6 | 85.23±0.5 | 77.97±2.3 | 68.33±5.8 | 58.96±11.2 | 81.09±0.7 | 82.00±1.5 | 68.11±2.6 | 68.24±4.1 | 70.73±4.6 | 64.68±2.0 | 73.18±8.5 | 74.88±6.8 | 83.76±0.4 | 75.35±7.1 |
| mammography | 89.92±1.3 | 88.34±2.8 | 89.62±0.6 | 87.73±5.2 | 75.21±6.0 | 71.96±21.5 | 83.60±1.8 | 80.44±6.9 | 86.62±5.0 | 87.41±3.0 | 85.20±4.6 | 87.26±3.9 | 86.55±3.8 | 88.48±1.2 | 83.46±1.7 | 88.22±0.6 |
| mnist | 93.41±1.5 | 94.28±0.4 | 93.45±0.8 | 92.38±0.5 | 78.68±5.0 | 75.20±18.6 | 74.86±13.8 | 72.71±2.6 | 84.01±7.3 | 82.59±6.5 | 85.20±4.6 | 87.26±3.9 | 91.00±0.3 | 92.33±0.5 | 90.32±0.4 |  |
| musk | 100.00±0.0 | 100.00±0.0 | 100.00±0.0 | 100.00±0.0 | 93.39±9.6 | 90.47±6.8 | 98.40±3.6 | 99.11±1.9 | 97.88±2.8 | 96.65±3.4 | 94.05±7.8 | 90.55±6.0 | 62.62±21.6 | 90.22±21.5 | 100.00±0.0 | 100.00±0.0 |
| optdigits | 88.88±4.4 | 91.71±1.9 | 77.08±10.3 | 80.21±1.9 | 59.23±10.2 | 48.75±8.1 | 70.57±14.6 | 50.24±11.7 | 46.08±23.6 | 68.29±6.8 | 59.48±17.1 | 60.70±12.1 | 63.75±18.7 | 78.33±25.2 | 88.06±0.9 | 54.08±0.9 |
| pendigits | 99.45±0.4 | 97.12±2.8 | 99.57±0.1 | 99.67±0.2 | 82.85±12.7 | 95.03±2.9 | 98.57±0.9 | 68.35±16.3 | 83.19±17.6 | 90.49±3.2 | 81.26±19.3 | 93.30±2.5 | 80.34±37.9 | 95.39±2.5 | 99.94±0.0 | 94.43±0.2 |
| satellite | 80.37±1.8 | 80.01±0.2 | 81.88±1.5 | 80.02±0.3 | 77.08±7.7 | 70.96±9.8 | 84.14±1.7 | 72.70±5.4 | 65.58±2.6 | 68.22±1.7 | 67.55±2.0 | 66.10±10.1 | 70.95±7.0 | 72.78±20.7 | 82.31±1.0 | 73.41±10.4 |
| satimage-2 | 99.68±0.1 | 99.65±0.1 | 99.59±0.2 | 99.86±0.0 | 97.61±1.0 | 97.50±1.4 | 98.70±0.8 | 82.69±12.1 | 96.94±0.8 | 95.67±3.3 | 97.92±0.6 | 87.00±8.4 | 79.59±27.0 | 99.87±0.0 | 99.83±0.1 | 98.74±1.0 |
| shuttle | 99.49±0.3 | 99.55±0.3 | 99.29±0.5 | 99.50±0.3 | 98.29±1.2 | 86.39±25.0 | 96.50±6.4 | 97.45±3.0 | 97.14±1.7 | 93.70±3.9 | 96.75±3.0 | 93.99±1.3 | 95.54±4.0 | 97.99±1.7 | 99.20±0.2 | 97.37±1.5 |
| speech | 48.03±1.1 | 47.24±0.6 | 50.57±0.4 | 47.34±0.6 | 49.96±3.2 | 51.26±2.4 | 50.20±5.3 | 53.31±3.8 | 47.11±5.1 | 52.28±4.6 | 57.15±4.5 | 50.76±3.1 | 48.20±2.1 | 50.05±0.0 | 51.71±1.2 | 53.15±9.4 |
| thyroid | 98.43±1.1 | 98.19±0.9 | 98.60±0.1 | 98.45±0.6 | 76.02±19.1 | 61.65±22.0 | 96.87±1.0 | 90.20±9.4 | 97.62±1.1 | 95.20±1.4 | 96.91±2.3 | 96.71±0.7 | 98.05±0.5 | 98.31±0.2 | 98.48±0.1 | 98.28±0.1 |
| vertebral | 52.85±10.1 | 48.39±3.1 | 57.58±5.6 | 48.14±6.9 | 47.85±7.5 | 51.89±6.7 | 44.82±6.3 | 45.60±7.6 | 41.70±3.9 | 52.44±5.7 | 74.87±2.3 | 74.11±1.8 | 44.91±3.8 | 47.83±3.5 | 55.54±3.9 | 43.54±2.4 |
| vowels | 97.09±1.4 | 94.31±4.6 | 98.80±0.3 | 94.60±0.4 | 72.84±21.0 | 69.69±10.2 | 95.87±2.3 | 77.12±5.6 | 64.05±6.6 | 63.46±13.3 | 59.95±8.3 | 65.69±16.4 | 70.92±16.7 | 91.65±9.0 | 97.97±0.6 | 67.96±2.7 |
| wine | 97.42±1.7 | 95.46±1.2 | 95.05±3.5 | 94.47±2.2 | 85.19±20.3 | 84.39±16.6 | 94.81±5.2 | 75.49±34.1 | 80.74±12.6 | 93.12±6.5 | 87.59±9.6 | 93.73±4.7 | 65.02±39.8 | 85.85±20.4 | 95.69±3.4 | 91.90±3.0 |
| yeast | 45.64±1.3 | 47.58±1.4 | 45.17±1.7 | 47.72±1.3 | 49.49±1.0 | 52.56±3.4 | 47.51±2.2 | 47.47±2.6 | 43.13±2.7 | 44.38±3.6 | 41.56±1.4 | 41.86±1.1 | 45.47±1.0 | 46.41±1.9 | 48.39±1.1 | 43.97±0.4 |

Table 7: F1 (%) score of EM/MV, EAG, NPD, and RTM on 38 benchmark datasets with 4 UAD methods. Average results are reported through 5 different random data splits.

| ad methods | AE | | | | DPAD | | | | DeepSVDD | | | | OCSVM | | | |
|---|---|---|---|---|---|---|---|---|---|---|---|---|---|---|---|---|
| delta method | EM/MV | EAG | NPD | RTM | EM/MV | EAG | NPD | RTM | EM/MV | EAG | NPD | RTM | EM/MV | EAG | NPD | RTM |
| ALOI | 10.34±1.3 | 9.86±0.6 | 11.32±0.5 | 9.53±0.7 | 8.17±3.4 | 8.16±0.8 | 14.85±0.9 | 9.90±1.1 | 8.21±0.4 | 8.39±0.7 | 8.06±0.6 | 8.29±0.5 | 8.26±0.3 | 8.19±0.1 | 19.55±0.3 | 8.14±0.0 |
| Cardiotocography | 57.55±2.7 | 61.55±1.1 | 57.25±2.3 | 55.41±3.5 | 43.73±4.4 | 50.69±15.3 | 54.81±4.4 | 54.89±5.5 | 68.58±5.5 | 63.13±2.9 | 62.79±8.8 | 46.23±8.0 | 61.03±11.6 | 45.02±11.7 | 64.38±2.0 | 66.27±0.8 |
| Hepatitis | 55.39±6.4 | 55.39±3.4 | 53.85±5.4 | 56.92±8.8 | 47.69±18.4 | 47.69±16.7 | 44.61±6.4 | 56.92±6.9 | 53.85±7.7 | 58.46±8.8 | 50.77±11.7 | 52.31±13.8 | 47.69±11.4 | 47.69±16.7 | 46.15±7.7 | 53.85±7.7 |
| InternetAds | 81.74±2.2 | 81.79±2.1 | 80.22±0.9 | 81.96±2.2 | 59.95±23.5 | 72.72±12.8 | 62.23±4.2 | 72.28±11.9 | 81.03±2.5 | 80.54±2.8 | 79.24±1.2 | 81.36±2.3 | 52.66±25.9 | 41.90±21.0 | 78.15±2.2 | 81.52±1.9 |
| Ionosphere | 89.84±1.1 | 89.52±0.4 | 90.60±0.6 | 90.06±0.1 | 85.56±5.8 | 74.76±5.8 | 90.48±1.1 | 77.46±7.4 | 77.46±7.9 | 75.08±6.5 | 78.73±6.4 | 75.72±1.4 | 78.10±9.6 | 72.38±11.5 | 90.48±1.9 | 83.33±4.6 |
| Lymphography | 86.67±13.9 | 90.00±9.1 | 76.67±14.9 | 90.00±9.1 | 83.33±0.0 | 83.33±11.8 | 73.33±9.1 | 80.00±7.5 | 80.00±7.5 | 76.67±14.9 | 83.33±0.0 | 83.33±11.8 | 66.67±39.1 | 86.67±13.9 | 80.00±13.9 | 90.00±9.1 |
| PageBlocks | 71.02±4.9 | 63.02±3.9 | 75.92±4.7 | 63.80±1.4 | 54.43±18.0 | 16.55±8.9 | 78.12±2.0 | 65.57±3.9 | 58.67±5.5 | 71.53±3.6 | 58.16±4.6 | 65.37±6.9 | 59.45±24.8 | 71.49±7.1 | 78.43±2.1 | 65.29±2.1 |
| Pima | 64.78±0.7 | 66.05±2.7 | 67.54±1.9 | 69.03±2.2 | 63.58±4.5 | 62.99±2.0 | 63.36±2.4 | 60.67±3.1 | 67.98±1.7 | 66.34±2.5 | 67.01±1.1 | 65.37±6.9 | 63.96±6.3 | 62.91±5.4 | 67.24±0.8 | 66.19±0.9 |
| SpamBase | 77.18±1.6 | 77.40±1.0 | 79.22±2.0 | 77.19±1.0 | 57.46±10.4 | 63.91±11.2 | 68.28±12.0 | 70.76±2.7 | 75.77±1.7 | 74.72±2.5 | 74.87±2.3 | 74.11±1.8 | 70.65±16.6 | 76.84±1.4 | 79.91±1.4 | 76.72±1.2 |
| Stamps | 65.81±9.8 | 65.81±4.9 | 70.32±5.3 | 67.09±8.4 | 54.84±22.2 | 63.87±13.2 | 57.42±10.3 | 60.65±16.8 | 58.06±12.7 | 56.77±16.7 | 51.61±16.0 | 61.93±7.7 | 64.52±8.2 | 58.06±22.0 | 70.32±5.3 | 61.29±9.1 |
| WBC | 72.00±8.4 | 78.00±4.5 | 76.00±5.5 | 80.00±5.5 | 66.00±8.9 | 82.00±8.4 | 74.00±5.5 | 82.00±4.5 | 86.00±5.5 | 70.00±8.9 | 82.00±11.4 | 80.00±7.1 | 74.00±15.2 | 86.00±5.5 | 76.00±5.5 | 80.00±5.5 |
| WDBC | 70.00±7.1 | 62.00±8.4 | 70.00±7.1 | 54.00±15.2 | 44.00±38.5 | 58.00±17.9 | 64.00±11.4 | 46.00±18.2 | 74.00±16.7 | 58.00±19.2 | 64.00±27.0 | 38.00±14.8 | 48.00±32.7 | 24.00±33.6 | 80.00±7.1 | 82.00±4.5 |
| WPBC | 38.30±3.0 | 33.61±1.0 | 37.45±8.6 | 32.34±3.5 | 37.45±2.4 | 37.87±6.5 | 34.04±9.2 | 37.45±4.9 | 35.17±4.3 | 39.58±6.1 | 38.30±6.6 | 35.74±1.8 | 33.19±4.9 | 41.28±2.4 | 35.74±4.6 | 31.49±4.1 |
| Waveform | 12.20±2.6 | 12.40±1.8 | 12.00±1.6 | 11.20±3.6 | 12.20±2.5 | 10.60±1.8 | 15.20±5.8 | 12.80±6.0 | 10.80±0.8 | 9.00±2.9 | 7.20±2.5 | 11.00±2.4 | 9.40±11.8 | 13.00±8.4 | 29.00±1.4 | 8.40±1.3 |
| Wilt | 13.23±12.9 | 11.67±9.8 | 21.87±5.0 | 5.68±5.1 | 11.20±10.3 | 19.38±8.2 | 26.61±2.6 | 0.47±0.7 | 1.56±2.4 | 7.16±5.2 | 3.19±2.8 | 0.23±0.0 | 4.13±3.4 | 8.48±3.7 | 7.16±2.8 | 2.10±0.3 |
| annthyroid | 57.00±4.8 | 47.30±2.9 | 62.13±3.9 | 56.52±7.0 | 33.26±23.1 | 22.92±21.5 | 60.11±1.4 | 47.56±7.1 | 48.39±4.4 | 37.00±8.2 | 46.25±6.0 | 48.20±8.9 | 48.01±22.2 | 64.42±2.1 | 64.50±2.1 | 50.19±1.8 |
| arrhythmia | 51.82±1.3 | 52.42±1.4 | 51.21±3.9 | 52.42±1.4 | 48.18±7.8 | 50.91±4.0 | 49.39±3.1 | 50.00±3.7 | 50.91±3.6 | 49.70±2.9 | 49.40±4.1 | 51.21±2.3 | 43.33±13.1 | 41.21±13.9 | 50.91±4.2 | 52.73±1.7 |
| breastw | 96.15±1.1 | 96.23±1.3 | 95.40±0.9 | 96.46±1.3 | 83.01±22.1 | 94.81±2.6 | 92.97±2.5 | 95.73±1.5 | 96.15±1.1 | 95.82±0.8 | 96.40±1.5 | 96.65±1.1 | 96.40±1.3 | 92.13±9.3 | 96.23±0.8 | 96.23±1.5 |
| cardio | 73.52±6.1 | 73.52±6.0 | 70.00±5.5 | 72.05±4.6 | 52.27±22.6 | 69.77±6.1 | 65.91±4.1 | 50.68±8.6 | 78.98±2.2 | 74.32±8.8 | 74.09±10.9 | 61.48±4.8 | 58.41±24.3 | 68.07±27.0 | 75.46±3.9 | 80.34±1.4 |
| fault | 67.99±1.6 | 62.86±2.2 | 70.25±1.2 | 67.79±3.9 | 63.77±7.9 | 57.98±7.8 | 71.59±0.7 | 61.64±3.3 | 57.86±3.3 | 51.47±3.0 | 57.15±4.5 | 50.76±3.1 | 62.85±6.1 | 60.42±0.9 | 68.59±1.5 | 56.91±0.5 |
| glass | 31.11±5.0 | 24.44±9.3 | 31.11±5.0 | 26.66±9.9 | 22.22±7.9 | 31.11±12.2 | 26.66±6.1 | 20.00±5.0 | 15.55±6.1 | 13.33±5.0 | 17.78±9.9 | 20.00±9.3 | 24.44±9.3 | 22.22±7.9 | 20.00±9.3 | 15.55±6.1 |
| landsat | 46.41±3.7 | 46.06±2.7 | 49.33±1.1 | 44.31±3.2 | 47.89±8.8 | 38.80±8.0 | 54.13±2.6 | 46.57±2.8 | 31.66±1.6 | 39.03±3.7 | 30.86±3.1 | 33.94±6.6 | 41.13±14.4 | 38.18±0.2 | 43.74±1.1 | 28.21±0.3 |
| letter | 44.40±7.2 | 35.40±5.9 | 58.40±2.3 | 44.40±4.3 | 27.40±14.4 | 16.60±14.6 | 40.60±3.6 | 24.00±7.1 | 15.40±4.2 | 11.60±3.4 | 16.80±4.1 | 9.40±1.1 | 18.20±11.9 | 47.80±17.2 | 48.00±2.0 | 20.00±5.6 |
| magic.gamma | 75.55±3.0 | 72.14±2.5 | 77.93±0.6 | 71.30±1.9 | 63.47±4.7 | 58.14±6.8 | 74.33±0.7 | 63.82±0.7 | 60.23±4.8 | 64.24±3.8 | 65.51±3.8 | 60.68±2.2 | 68.01±6.6 | 69.36±5.6 | 77.05±0.4 | 69.62±6.1 |
| mammography | 44.62±5.8 | 39.00±12.7 | 43.46±5.5 | 39.85±15.3 | 24.00±8.6 | 25.00±12.8 | 34.31±6.2 | 31.16±8.6 | 39.69±12.4 | 42.23±10.1 | 32.62±10.0 | 45.39±7.0 | 44.38±2.3 | 40.92±3.8 | 44.62±1.4 | 45.46±0.9 |
| mnist | 71.51±2.6 | 73.49±0.9 | 71.66±1.3 | 70.43±1.4 | 52.83±6.5 | 48.37±23.6 | 47.23±15.4 | 46.31±2.5 | 56.57±10.8 | 58.74±5.9 | 57.52±8.6 | 58.24±3.9 | 28.29±22.0 | 67.80±1.5 | 71.66±0.8 | 66.40±1.1 |
| musk | 100.00±0.0 | 100.00±0.0 | 100.00±0.0 | 100.00±0.0 | 67.01±22.3 | 49.48±28.5 | 96.29±8.3 | 92.37±13.1 | 74.64±19.0 | 60.62±19.9 | 66.60±22.4 | 39.59±19.9 | 25.16±41.9 | 78.35±40.3 | 100.00±0.0 | 100.00±0.0 |
| optdigits | 14.40±10.9 | 20.27±9.4 | 5.33±8.2 | 1.33±0.7 | 11.20±10.3 | 3.20±6.1 | 19.47±8.1 | 6.67±6.5 | 0.93±2.1 | 1.87±2.6 | 4.93±10.0 | 0.13±0.3 | 10.27±23.0 | 30.93±23.8 | 11.73±1.9 | 0.00±0.0 |
| pendigits | 82.31±9.2 | 60.38±24.9 | 84.23±2.9 | 86.03±3.2 | 40.90±18.4 | 53.08±14.5 | 74.62±8.9 | 24.74±16.1 | 38.20±33.9 | 18.08±20.6 | 33.08±18.6 | 36.28±13.2 | 54.23±35.3 | 44.36±0.8 | 95.51±1.4 | 45.13±2.0 |
| satellite | 73.00±1.4 | 73.60±0.3 | 73.86±1.1 | 73.60±0.3 | 68.63±6.9 | 73.83±3.0 | 74.99±1.5 | 65.03±4.8 | 63.42±2.1 | 64.63±1.0 | 62.69±1.8 | 59.74±8.9 | 65.35±3.5 | 65.54±16.4 | 72.64±0.8 | 66.70±7.5 |
| satimage-2 | 85.07±7.9 | 81.41±5.9 | 80.00±5.5 | 93.24±1.8 | 74.93±7.0 | 78.59±11.5 | 72.11±9.7 | 44.79±21.8 | 81.13±7.4 | 58.31±24.4 | 79.16±9.9 | 12.68±12.2 | 55.49±49.4 | 94.93±0.8 | 92.11±0.8 | 89.30±4.7 |
| shuttle | 75.38±16.7 | 78.46±16.7 | 69.23±15.4 | 73.85±10.5 | 52.31±16.7 | 47.69±32.4 | 46.15±9.6 | 61.54±28.3 | 43.08±28.1 | 23.08±7.7 | 44.62±21.3 | 26.15±6.9 | 35.38±15.0 | 46.15±10.9 | 63.08±10.0 | 40.00±8.4 |
| speech | 5.58±0.9 | 4.92±0.0 | 7.54±1.5 | 4.59±0.7 | 2.30±2.7 | 2.30±2.7 | 4.59±3.2 | 2.95±1.8 | 2.30±1.9 | 5.39±1.9 | 2.62±1.5 |  | 3.61±2.1 | 3.94±0.9 | 6.56±0.0 | 6.56±3.8 |
| thyroid | 74.41±11.2 | 72.69±2.5 | 73.98±4.0 | 74.84±5.3 | 32.69±30.8 | 18.06±21.0 | 63.66±5.1 | 59.57±18.2 | 70.97±7.5 | 48.17±7.7 | 64.52±14.3 | 62.80±3.9 | 69.68±7.4 | 71.18±4.4 | 70.97±2.3 | 72.90±1.8 |
| vertebral | 22.00±11.9 | 14.67±1.8 | 25.33±8.4 | 14.00±9.3 | 17.33±10.4 | 23.33±11.5 | 20.67±8.9 | 16.66±8.5 | 11.33±3.0 | 11.00±11.6 | 43.33±8.7 | 33.33±5.8 | 16.67±2.4 | 15.33±4.5 | 20.67±7.6 | 14.67±1.8 |
| vowels | 75.60±4.8 | 60.00±18.0 | 80.80±3.9 | 56.40±4.8 | 28.40±20.5 | 29.20±14.2 | 61.60±7.9 | 41.20±7.9 | 14.00±6.2 | 8.80±9.9 | 13.20±7.9 | 8.80±8.6 | 31.60±24.3 | 48.80±32.7 | 76.00±6.9 | 25.20±1.8 |
| wine | 76.00±11.4 | 66.00±11.4 | 62.00±17.9 | 66.00±11.4 | 56.00±30.5 | 58.00±19.2 | 74.00±15.2 | 54.00±35.1 | 42.00±26.8 | 64.00±13.4 | 52.00±26.8 | 66.00±20.7 | 40.00±24.5 | 62.00±25.9 | 68.00±17.9 | 60.00±10.0 |
| yeast | 47.77±1.2 | 49.35±1.6 | 46.98±1.3 | 49.63±0.8 | 51.40±1.1 | 52.94±3.2 | 48.84±2.0 | 48.60±3.3 | 45.56±2.6 | 46.47±3.1 | 44.58±1.7 | 44.61±1.3 | 46.90±0.7 | 47.53±1.3 | 49.82±0.6 | 46.11±0.6 |

Table 8: Comparative study of Bayesian optimization through 14 datasets on 5 additional UAD algorithms. Mean AUC and F1 scores are reported. The best performance is **bold**.

| Method | PLAD | | NeuTraLAD | | HRN | | DROCC | | SCAD | |
| Metric | AUC | F1 | AUC | F1 | AUC | F1 | AUC | F1 | AUC | F1 |
|---|---|---|---|---|---|---|---|---|---|---|
| Max | 77.08+-19.3 | 57.89+-24.5 | 76.24+-16.0 | 52.04+-22.2 | 83.53+-17.2 | 68.27+-18.1 | 49.26+-26.5 | 36.11+-24.8 | 86.36+-15.9 | 63.15+-18.8 |
| EM/MV | 41.20+-25.7 | 23.57+-18.4 | 26.86+-33.7 | 15.9+-22.9 | 78.17+-20.5 | 53.87+-28.6 | 27.60+-24.7 | 16.86+-16.4 | 77.55+-19.4 | 54.96+-27.1 |
| RTM(Ours) | 32.48+-19.8 | 17.53+-16.5 | 60.33+-17.0 | 33.45+-28.7 | **80.44**+-17.1 | **59.48**+-26.4 | 35.41+-25.2 | 23.52+-21.0 | 74.92+-17.3 | 48.06+-24.3 |
| EAG(Ours) | **59.22**+-21.4 | **38.64**+-24.0 | 68.44+-25.2 | **45.49**+-28.6 | 79.24+-18.9 | 55.31+-32.9 | **40.89**+-32.3 | **25.50**+-25.1 | 77.13+-18.8 | 49.74+-26.7 |
| NPD(Ours) | 47.23+-24.3 | 30.31+-23.2 | **70.64**+-18.4 | 39.48+-28.2 | 64.81+-27.5 | 46.07+-33.3 | 26.97+-23.1 | 14.46+-16.0 | **82.62**+-17.2 | **58.11**+-21.3 |

The core hyper-parameters considered for AutoUAD via BO are listed below.

- PLAD: : coefficient to control the perturbation. *Hidden dim. 1*: the width of the first and the last hidden layer of the neural network. *Hidden dim. 2*: the width of the second and the penultimate hidden layer of the neural network, which is known as the size of the hidden representation. *Weight decay*: the regularization term to control model complexity.

- NeuTraLAD: $k$: number of transformations. $\tau$: temperature for contrastive loss. *Hidden dim.* : the size of the hidden representation. *Encoder dim.*: Hidden dimension of encoder module. *Transform dim.*: Hidden dimension of transformation module.

- HRN: $n$: order of the H-regularization. $\lambda$: coefficient of the H-regularization. *Hidden dim.* : the size of the hidden representation. *Weight decay*: the regularization term to control model complexity.

- DROCC: $\gamma$: parameter to vary projection. $r$: radius of hypersphere to sample points from. $lamda$: weight is given to the adversarial loss. *Hidden dim.* : the size of the hidden representation.

- SCAD: $k$: kernel size for sliding window. $\tau$: temperature for contrastive loss. *Hidden dim.* : the size of the hidden representation. *Weight decay*: the regularization term to control model complexity.

## K EFFICIENCY OF PROPOSED METRICS

We test the running time efficiency of datasets of different sizes. Notice that MC and HITS have theoretical computation time complex with $O(LN^2)$ and $O(tLN)$, respectively, where $t$ is the max iteration in HITS, and $L$ is the size of UAD model pool, i.e. $L = |\{(\mathcal{M}_i, \Theta_j)|\mathcal{M}_i \in \mathbb{M}, \Theta_j \in \mathcal{S}^{(i)}\}|$. The running time results for selection through 200 candidate UAD models are shown in Figure 8. It is seen that NPD is the most efficient method.

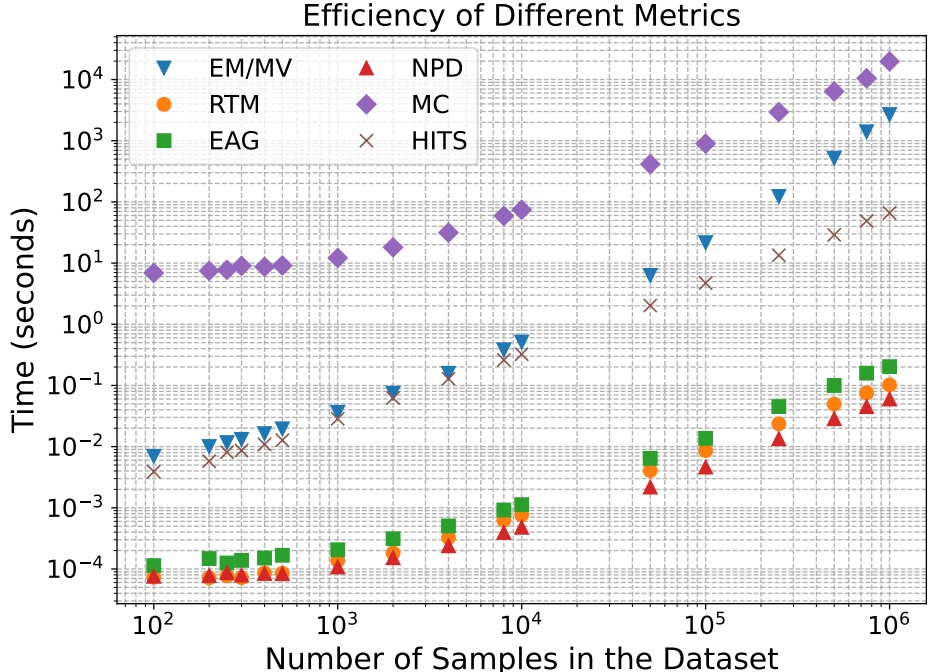

Figure 8: Running time efficiency comparison.

## L    FLOWCHART OF AUTOUAD

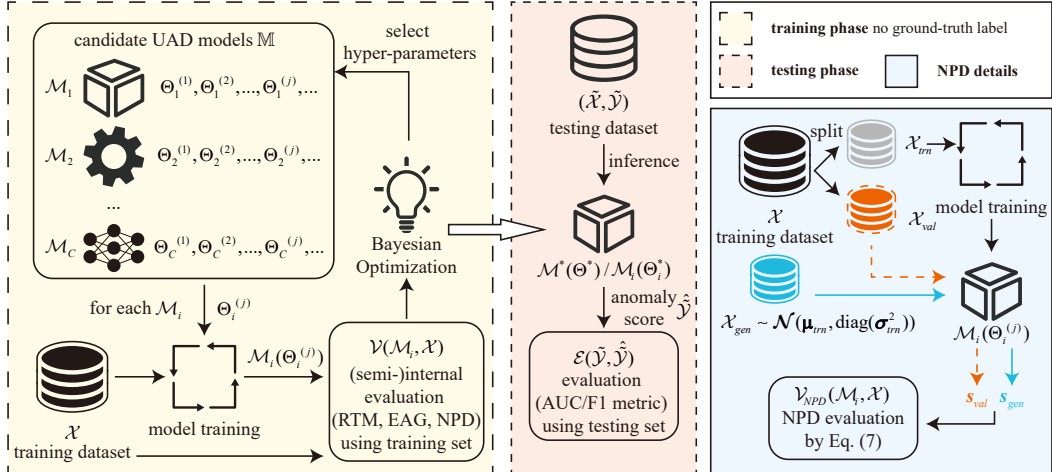

Figure 9: Working flowchart of AutoUAD. **Training Phase:** Candidate UAD models $\mathbb{M}$ contain multiple UAD algorithm $\{\mathcal{M}_1, \mathcal{M}_2, ..., \mathcal{M}_C\}$, and each algorithm $\mathcal{M}_i$ has multiple candidate hyper-parameters $\{\Theta_i^{(1)}, \Theta_i^{(2)}, ..., \Theta_i^{(j)}, ...\}$, $\Theta_i^{(j)} \in \prod_{k=1}^{H_i} \mathcal{S}_k^{(i)}$. For each $\mathcal{M}_i$, the Bayesian optimizer will select hyper-parameters to train the UAD model. After training converges, the model is evaluated by (semi-)internal metric (RTM, EAG, or NPD) using the training set where no ground-truth label is required. The evaluation output $\mathcal{V}(\mathcal{M}_i, \mathcal{X})$ provides feedback to the Bayesian optimizer to select new hyper-parameters for the next round. **Testing Phase:** After training, we obtain the best model $\mathcal{M}_i(\Theta_i^*)$ for each UAD algorithm. By selecting the algorithm with highest $\mathcal{V}$, we obtain $\mathcal{M}^*(\Theta^*)$. During testing, the selected model is evaluated by $\mathcal{E}$ (AUC/F1) using the testing set with the ground-truth label to show the effectiveness of our methods. **NPD Details:** The training dataset is randomly split into $\mathcal{X}_{trn}$ and $\mathcal{X}_{val}$ before training phase. An extra dataset $\mathcal{X}_{gen}$ is generated from an isotropic Gaussian $\mathcal{N}(\boldsymbol{\mu}_{trn}, \text{diag}(\boldsymbol{\sigma}_{trn}^2))$, where $\boldsymbol{\mu}_{trn}$ and $\boldsymbol{\sigma}_{trn}^2$ are the mean and variance vectors of $\mathcal{X}_{trn}$. $\mathcal{X}_{trn}$ is used to train the UAD model. After the model training, anomaly scores $\boldsymbol{s}_{val}$ and $\boldsymbol{s}_{gen}$ are computed from $\mathcal{X}_{val}$ and $\mathcal{X}_{gen}$, respectively. Then, NPD is calculated taking $\boldsymbol{s}_{val}$ and $\boldsymbol{s}_{gen}$ as input using equation 7. We argue that $\mathcal{X}_{gen}$ can contain samples close to real anomalies so that NPD can show the significance of evaluating a good UAD model. It is justified in Theorem 3.

## M    SENSITIVITY OF $\tau$ IN RTM

Due to time constraints, we perform sensitivity analysis varying $\tau$ in $[50, 30, 20, 10, 5, 3, 1]$ tested on 37 datasets (ALOI is dropped due to its size) using DPAD and OCSVM. An average result is summarized in Figure 10. It is seen that the performance varies much as the change of $\tau$, especially in DPAD results, $\tau = 3$ and $\tau = 5$ show very different results, making the choice $\tau$ difficult and imperial. It also reveals the internal evaluation metric is still sensitive to the additional hyper-parameter.

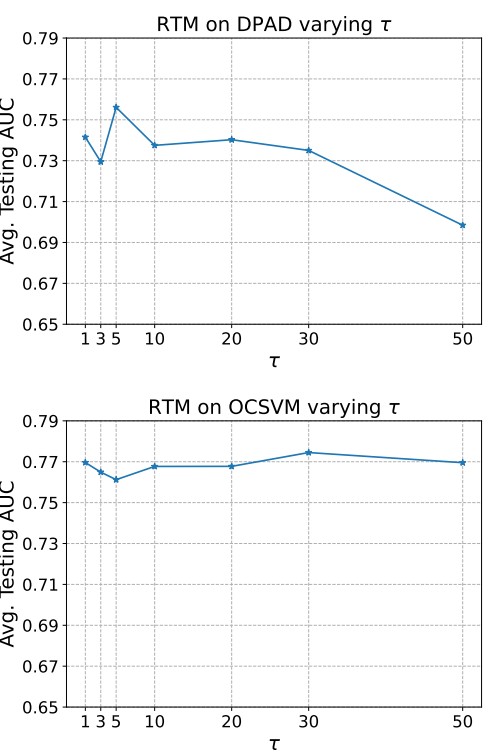

Figure 10: Average testing AUC across 37 datasets varying $\tau$ in RTM.

