# OpenReview forum: "AutoUAD: Hyper-parameter Optimization for Unsupervised Anomaly Detection"
_ICLR.cc/2025/Conference — ICLR 2025 Poster_

### Official Review · Reviewer_FcaY · 2024-10-25

**Soundness:** 2
**Presentation:** 2
**Contribution:** 2
**Rating:** 6
**Confidence:** 4

**Summary:**

This paper introduces different metrics for the optimization of hyperparameters in unsupervised anomaly detection. The authors formulate the problems of hyperparameter tuning and model selection as a maximization of the expected evaluation metric for a random dataset containing anomalies. They then propose three metrics that approximate this unknown expected value (up to some monotonically increasing function).
The key contributions of the paper are (i) the introduction of three evaluation metrics for hyperparameter optimization, (ii) theoretical analysis of the third metric and (iii) empirical experiments comparing the optimization based on the proposed evaluation metrics with other model selection methods.

**Strengths:**

The paper has a red line and the structure from internal to external evaluation metrics is comprehensible. The problem of hyperparameter optimization is well motivated, and extensive empirical experiments are provided.

The proposed metrics are, generally, clearly defined, and the approach is innovative.

This paper could be a helpful contribution to unsupervised hyperparameter tuning.

**Weaknesses:**

The proposed metrics seem to work only under strong assumptions, no theoretical guarantees in terms of the expected false positive/negative rate are provided and the results of experiments seem to be overly optimistic with results being reported on the same data that was used for model selection and hyperparameter tuning.

Given the current shortcomings (see below), I recommend to reject the paper. I would be willing to increase the score, given that the below concerns are addressed.

---
UPDATE: Based on the additional results and explanations provided by the authors, I suggest to accept the paper.

**Questions:**

Comments/Questions:
- Hyperparameter tuning is a challenge in other unsupervised learning tasks as well. Which relevant approaches are there, that are worth being added to the "Related Work" section?
- The internal evaluation metrics depend on hyperparameters themselves, which does not seem to be an advantage. How robust are they w.r.t. different choices of the hyperparameters?
- Definition 2: Are the true labels supposed to be in $\{0, 1\}$? Generally, the anomaly scores are not restricted to the interval $[0, 1]$. What should be passed to the evaluation metric $\mathcal{E}?$ The anomaly scores, or binary predictions based on them?
- L205: "While these tail points are part of the normal data, their distance from the majority
can make them resemble potential anomalies. They should be recognized by a good UAD model." -> Should a good UAD model recognize these points as anomalies or normal data? From my understanding, a *good* UAD model should not only perform well for data close to the majority, but also for data points at the boundary of "normal" behavior
- What is the intuition of using the mean and median to define RTM? It seems more natural to use the $mean(\{s_i|s_i < s_{top\tau\%}\})$ instead of the median (or the median in both cases).
- Can the RTM be interpreted so that an error of $\tau\%$ is accepted? Similar to the significance level of a statistical test?
- L249: What is the intuition behind the statement: "RTM may overlook a case in which all scores are uniformly distributed..."?
- L255: "Based on Assumption 1, there exists (?) and ..." -> What does exist?
- L257: What is $\xi$ in equation (4)? There seems to be something missing.
- What is the intuition of the definition of $AG(\xi, p(s))$? Why are the probabilities $w_0(k)$ and $w_1(k)$ used as weights? This seems to prefer values of $k$ such that $s_k$ is close to the median of the sample $s$.
- Definition 5: $AG(\xi)$ is not defined, only $AG(\xi; p(s))$ - and the latter not correctly
- Definition 5: Why is the conditional expectation needed? $\xi \ge s_{thr}$ by definition, so the conditional expectation reduces the the expectation, right?
- L288: If overfitting w.r.t. the training data is an issue, couldn't this simply be fixed by using a train-test-split or cross validation?
- Definition 6: Here the underlying assumption is that the "normal" data is approximately normal distributed with distribution $\mathcal{N}(\mu, diag(\sigma))$. This seems to be a very strong assumption, especially in applications with real data.
- Definition 6: Why are the coordinates assumed to be uncorrelated (as modeled with the diagonal matrix)? Why are covariances between the coordinates not considered? This assumption is **very** restrictive and even in the examples not met (see, e.g., Fig. 3)
- L372: Geometric interpretation: Why is the outline generally a hypersphere? That is, why is this the case if the variance across different coordinates varies?
- Figure 6: What does the y-axis represent? The correlation of $\nu$ with AUC? In this case, high values would be good, so the results for NPD/NDP seem ambiguous.
- Are the final results, e.g. in Table 1, calculated based on the validation data, i.e., the same data that was used for model selection and hyperparameter tuning? If so, the results are probably overly optimistic for the model selection methods.
- Some considered datasets contain unrealistic many anomalies. Can "anomalies" that occur more than 10-20\% of the time be considered as real anomalies?

Additional Feedback
- Definition 1: "normal distribution" (and the term "normal" in general) is ambiguous: clearly the distribution of normal data points is meant and not the normal distribution $\mathcal{N}(\mu, \sigma)$, but the terms "normal" (as typical) and "normal" (as Gaussian) should be distinguished better
- Definition 1: $p_0$ and $p_1$ are not defined - probably the density functions of $D_0$ and $D_1$?
- Definition 2 (L. 170-171): The joint distribution of $(\tilde{X}, \tilde{Y})$ could be specified instead of the vague "Let $\tilde{X}$ be a random unseen dataset containing both normal samples and anomalous samples"
- Definition 2: $\mathbb{E}$ should denote the expectation over the joint distribution of $(\tilde{X}, \tilde{Y})$
- Definition 3: The notation could be simplified by using the usual notation $s_{(i)}$ for sorted values (with $s_{(1)} \le \dots \le s_{(n)}$ and $s_{(\frac{\tau}{100} N)}$ instead of $s_{top\tau\%}$
- Definition 3: $\varepsilon$ could be avoided by the case distinction $med(s)=0$ and $med(s)\neq 0$
- Definition 4: "Let $p(s)$ be the distribution of the anomaly score $s$" -> Emphasize that $p(s)$ is the unknown, theoretical distribution, not the empirical distribution based on the observations in $s$
- Definition 5: From Def. 4 it is not clear what the role of $\xi$ is. If $\xi=k$, only integer-valued are allowed. In this case, the distribution is simply the discrete uniform distribution on indices $k$ with $s_k > s_{thr}$, right?
- Typo in Definition 6: "varaince vector" -> "variance vector"
- The abbreviation NPD/NDP is not used consistently (mostly NPD, in experiments NDP)

---

> ### Author Response · Authors · 2024-11-19
> **Rebuttal: Part I**
>
> Dear Reviewer FcaY,
>
> We sincerely thank your valuable comments. There seem to be **two big misunderstandings** regarding the weakness if we understand your comments correctly.
>
> **On assumptions** \
> You commented that "The proposed metrics seem to work only under strong assumptions". In our paper, we made two assumptions:
>    * Assumption 1: A good UAD model will assign the majority of data points with low anomaly scores, and assign the minority of data points with relatively high anomaly scores.
>    * Assumption 2: (Low-Quality Data Upper Bound) At most $20\%$ data points in the training set are similar to the true anomalies, in which the UAD model gives them higher anomaly scores.
>
> Assumption 1 is commonly used in unsupervised anomaly detection. It is really a very weak assumption. Assumption 2 is also a weak assumption in unsupervised anomaly detection, where we usually assume that most of the training data are normal. (Note that here we use "At most $20$\%", not "$20$\%".)
>
> It is also worth mentioning that Assumption 1 and Assumption 2 are for our first two metrics RTM and EAG respectively. Our third metric NPD does not rely on any assumption. We generated $\mathcal{X} _{gen}$ from an isotropic Gaussian with the same mean and variance vectors as $\mathcal{X}$ but we never assume that real data are close to $\mathcal{X} _{gen}$. We only use $\mathcal{X} _{gen}$ to form a hypersphere roughly enclosing $\mathcal{X}$, as shown in Figure 4 and Figure 7. More importantly, the effectiveness of our methods is demonstrated by 38 datasets from diverse fields, which also indicates that our methods do not rely on strong assumptions.
>
> **On reported results**\
> You commented that "the results of experiments seem to be overly optimistic with results being reported on the same data that was used for model selection and hyperparameter tuning". We'd like to clarify that in the experiments for every dataset, there is a training set and a testing set, and the training set is unlabeled.  We performed model selection and hyperparameter tuning on the training set and obtained the final optimal model. The results we reported in the tables and figures are obtained by applying the final optimal model to the testing set.

---

> ### Author Response · Authors · 2024-11-19
> **Rebuttal: Part II**
>
> **Bound of expected FPR and FNR**
>
> Regarding the expected false positive/negative rate, in this revision, we provide a theoretical guarantee (Theorem 4) in Appendix B. Since the training data of UAD is not labeled at all, it is difficult to directly provide a theoretical guarantee for the performance (e.g. FPR and FNR) on the testing data and we have to make additional assumptions. For convenience, we show our new result in the following context.
>
>
> To calculate the false positive rate (FPR) and false negative rate (FNR), we need to determine a threshold for the anomaly scores given by a model $\mathcal{M}$. For convenience, we give the following definitions.
>
> **Definition 8:**
> Let $\tau _{\mathcal{M}}(z)={1}(z>c)$ be a threshold function, where $c>0$ is the threshold. Let $F _{\mathcal{M}}=\tau _{\mathcal{M}}\circ f _{\mathcal{M}}:\mathbb{R}^d\rightarrow\{0,1\}$ and $\mathcal{F} _{\mathcal{M}}$ be the class of $F _{\mathcal{M}}$ defined by $\mathcal{M}$ with hyperparameters $\Theta$.
> The FPR and FNR on the unseen testing data are then defined as $\text{FPR}=\mathbb{E} _{\mathbf{x}\sim\mathcal{D} _0}[F _{\mathcal{M}}(\mathbf{x})]$ and $\text{FNR}=\mathbb{E} _{\mathbf{x}\sim\mathcal{D} _1'}[1-F _{\mathcal{M}}(\mathbf{x})]$ respectively.
>
> Without loss of generality and for convenience, we assume that in the problem defined by Definition 1, $N_1=0$, and in NPD, $M=N/2$. The following theorem (proved in Section C.4) provides a bound for the FPR and FNR on the unseen testing data.
>
> **Theorem 4:**
> Based on Definition 8, letting $\Delta=\min\lbrace{\max _{\mathbf{x}\in\mathcal{X} _{\text{gen}}}f _{\mathcal{M}}(\mathbf{x})-\min _{\mathbf{x}\in\mathcal{X} _{\text{gen}}}f _{\mathcal{M}}(\mathbf{x}),\max _{\mathbf{x}\in\mathcal{X} _{\text{val}}}f _{\mathcal{M}}(\mathbf{x})-\min _{\mathbf{x}\in\mathcal{X} _{\text{val}}}f _{\mathcal{M}}(\mathbf{x})\rbrace}$, $\varsigma=\max _{\mathbf{x}\in\mathcal{X} _{\text{gen}}}f _{\mathcal{M}}(\mathbf{x})$, and $\kappa=1.2+2(c^{-1}-\frac{1}{5\varsigma})\sum _{\mathbf{x}\in\mathcal{X} _{\text{gen}}}f _{\mathcal{M}}(\mathbf{x})/N$,
> then over the randomness of $\mathcal{X} _{\text{val}}$ and $\mathcal{X} _{\text{gen}}$, the following inequality holds with probability at least $1-2\delta$:
>
> $$
>            \text{FPR}+\text{FNR}\leq \kappa-\frac{2\Delta}{c\sqrt{N}}\sqrt{\mathcal{V} _{\text{NPD}}(\mathcal{M},\mathcal{X})}+\frac{\sqrt{2}}{2}\sqrt{D _{\text{KL}}(\mathcal{D} _{\text{gen}}||\mathcal{D} _1')}
>            +\widehat{\mathfrak{R}} _{\mathcal{X} _{\text{val}}}(\mathcal{F} _{\mathcal{M}})+\widehat{\mathfrak{R}} _{\mathcal{X} _{\text{gen}}}(\mathcal{F} _{\mathcal{M}})+6\sqrt{\frac{\log \frac{2}{\delta}}{N}}
> $$
>
> In the theorem, the empirical Rademacher complexities $\widehat{\mathfrak{R}} _{\mathcal{X} _{\text{val}}}(\mathcal{F} _{\mathcal{M}})$ and $\widehat{\mathfrak{R}} _{\mathcal{X} _{\text{gen}}}(\mathcal{F} _{\mathcal{M}})$ can be explicitly bounded for any $\mathcal{M}$ (e.g., OC-SVM and AE) with any hyperparameters $\Theta$ and the corresponding technique is fairly standard in the literature [1][2]. Due to this, together with the fact that our work AutoUAD is a framework not specialized to a single $\mathcal{M}$, we will not show the $\mathcal{M}$-specific computation of $\widehat{\mathfrak{R}} _{\mathcal{X} _{\text{val}}}(\mathcal{F} _{\mathcal{M}})$ and $\widehat{\mathfrak{R}} _{\mathcal{X} _{\text{gen}}}(\mathcal{F} _{\mathcal{M}})$.
> Note that $D _{\text{KL}}(\mathcal{D} _{\text{gen}}||\mathcal{D} _1')$ can be further bounded by the similar approach used in Theorem 3.
> Our AutoUAD finds the model with the largest $\mathcal{V} _{\text{NPD}}$ and hence has the potential to reduce the false positive rate and false negative rate. In addition, a smaller $D _{\text{KL}}(\mathcal{D} _{\text{gen}}||\mathcal{D} _1')$ or complexity of $\mathcal{M}$ (measured as $\widehat{\mathfrak{R}} _{\mathcal{X} _{\text{val}}}(\mathcal{F} _{\mathcal{M}})$ and $\widehat{\mathfrak{R}} _{\mathcal{X} _{\text{gen}}}(\mathcal{F} _{\mathcal{M}})$)
> may also lead to a lower error rate.
>
> [1] Bartlett and Mendelson. Rademacher and Gaussian complexities: Risk bounds and structural results. JMLR 2002.
>
> [2] Bartlett et al. Spectrally-normalized margin bounds for neural networks. NeurIPS 2017.

---

> ### Author Response · Authors · 2024-11-19
> **Rebuttal: Part III**
>
> **Q1:** Thank you for your suggestion. Clustering and dimensionality reduction are another important unsupervised learning task. We add the following paragraph in the Related Work section:
>
> Note that hyperparameter tuning is a challenge in other unsupervised learning tasks as well. (Halkidi \& Vazirgiannis, 2001; Poulakis, 2020; Fan et al., 2022) presented some clustering validity metrics
> to guide the hyper-parameter search using grid search or Bayesian optimization. (Lin \& Fukuyama,2024; Liao et al., 2023) also used Bayesian optimization to tune the hyperparameter in dimensionality reduction methods such as t-SNE.
>
> [1] Halkidi and Vazirgiannis. Clustering validity assessment: Finding the optimal partitioning of a data set. ICDM 2001.
>
> [2] Giannis Poulakis. Unsupervised automl: a study on automated machine learning in the context of clustering. 2020.
>
> [3] Fan et al. A simple approach to automated spectral clustering. NeurIPS 2022.
>
> [4] Liao et al. Efficient and robust bayesian selection of hyperparameter. arXiv:2306.00357, 2023.
>
> **Q2:** Take RTM as an example, $\tau$ is a hyper-parameter. Due to time constraints, we perform sensitivity analysis for RTM with varying $\tau$ in $[50, 30, 20, 10, 5, 3, 1]$ tested on 37 datasets (ALOI is dropped due to its size) using DPAD and OCSVM. An average result is summarized in Figure 10 of Appendix L and also shown in the following table https://anonymous.4open.science/r/AutoUAD-FDFC/AutoAD_sensitivity.pdf.  It is seen that compared to OCSVM, the performance of DPAD is more sensitive to $\tau$.
>
> ### Table: Average test AUC RTM on DPAD and OCSVM across 37 datasets (ALOI is dropped due to its size).
>
> | **$\tau$**   | **50**           | **30**           | **20**           | **10**           | **5**            | **3**            | **1**            |
> |--------------|------------------|------------------|------------------|------------------|------------------|------------------|------------------|
> | **DPAD**     | $69.84 \pm 20.3$ | $73.5 \pm 19.2$  | $74.03 \pm 18.5$ | $73.75 \pm 16.0$ | $75.61 \pm 15.1$ | $72.95 \pm 18.9$ | $74.15 \pm 19.2$ |
> | **OCSVM**    | $76.95 \pm 20.3$ | $77.44 \pm 20.8$ | $76.77 \pm 20.8$ | $76.77 \pm 20.7$ | $76.12 \pm 20.9$ | $76.49 \pm 20.8$ | $76.96 \pm 20.1$ |
>
> **Q3:** The ground-truth labels is in $\{0, 1\}$. The input of the metric $\mathcal{E}$ depends on the choice. For example, AUC is calculated based on the anomaly score while F1 needs the binary prediction. A threshold is often required in UAD research to turn the anomaly score into a binary prediction. The calculation of the F1 score is consistent with previous research [5, 6].
>
> [5] Shenkar and Wolf. Anomaly detection for tabular data with internal contrastive learning. ICLR 2022.
>
> [6] Qiu et al. Neural transformation learning for deep anomaly detection beyond images. ICML 2021.
>
> **Q4:** This is an insightful question. You are right. Consider the unsupervised outlier detection on the contaminated dataset, i.e. $N_1>0$ in our Definition 1. A good model should assign higher anomaly scores to those anomalies. When $N_1 = 0$, some data point always exists around the decision boundary, e.g. support vectors in OCSVM and should have higher anomaly scores.  Whether a good UAD model recognizes these samples as anomalies or normal data requires further assumption or prior knowledge. In our statement, we mean a good UAD model should assign relatively high anomaly scores to these samples.
>
> We find the current word is controversial. We revise it as "... They should be assigned relatively high anomaly score by a good UAD model."
>
> **Q5:** The intuition of using median(s) instead of mean(s) is that mean is more likely biased to the large values in {$\{s _i |  s _i < s _{\text{top}\tau}\}$}. During our preliminary exploration, using relative top-$\tau\%$ mean minus mean often underperforms compared with RTM. Using the median in both cases may lose the information of large anomaly scores and make the gap small. In one word, we hope the statistical value of {$\{s _i |  s _i < s _{\text{top}\tau}\}$} reasonably small and that of {$\{s _i |  s _i \geq s _{\text{top}\tau}\}$} large.
>
> **Q6:** Yes. It can be interpreted as an error percentile. We have taken your suggestion in F5. Thank you so much.
>
> **Q7:** Consider anomaly scores inferred on $N$ training sample in the following form $\{1, 2, 3, ..., N\}$. In this case, RTM will give a large value. However, this case does not satisfy Assumption 1. To solve this problem, we then propose the EAG metric which utilizes the variance of {$\{s _i |  s _i < s _{\text{top}\tau}\}$}.

---

> ### Author Response · Authors · 2024-11-19
> **Rebuttal: Part IV**
>
> **Q8, 9, 11, F7, and F8:**  We apologize for the typo and inconsistency in the main text. We notice this is a serious typo and it makes the reading of the whole section difficult. We revise Definition 4 by changing $s_k$ into $\xi$, making AG a function related to $\xi$ (a value from the distribution of $s$) with a parameter $p(s)$. Furthermore, we emphasize that $p(s)$ is an unknown and theoretical distribution of the anomaly score $\boldsymbol{s}$ to reduce the ambiguity.
>
> **Q10:** The intuition of AG is similar to RTM. We hope a good UAD model assigns relatively high anomaly scores to a small portion of training samples. AG utilizes the variance information of the two groups of anomaly scores split by a threshold to avoid the case explained in Q7. In our new revision, $w_0(k)$ and $w_1(k)$ are replaced by $w_0(\xi)$ and $w_1(\xi)$, respectively. $\xi$ is a threshold value in the domain of $s$. In a discrete form, $w_0(\xi)$ and $w_1(\xi)$ are the |{$s_i| s_i < \xi $ }| $/ N$ and |{$  s_i | s_i \ge \xi $ }| $/ N$, respectively. They are naturally the weight.
>
> **Q12:** Using AG solely to evaluate the UAD model depends on the choice of a sensitive hyper-parameter, i.e., the threshold ($\xi$ in the new version). By taking the expectation, we hope to reduce the sensitivity. When choosing a lower threshold, the output AG seems meaningless to evaluate a UAD model. So, we design to use the conditional expectation to reduce the bias from choosing lower thresholds. Surely it will reduce the expectation, but this is what we want. Otherwise, the distinction between the "good" and "bad" UAD model will be small in terms of EAG.
>
> **Q13:** No. Train-val-test split cross-validation is non-trivial in the unsupervised learning setting. In the supervised learning tasks, a **labeled** validation set can be used to prevent overfitting. In contrast, the validation set in the unsupervised task is still **unlabeled**. No ground-truth information can be revealed using a validation set. We tried to simply apply RTM and EAG on a **unlabeled** validation set (using 30\% of $\mathcal{X}$, consistent with NPD in the experiment), yet the performance of the selected model remains almost unchanged or underperforming. We show the average testing AUC results through 10 datasets (Shuttle, Arrhythmia, Fault, Glass, Hepatitis, InternetAds, Ionosphere, Landsat, Letter, and Lymphography) using RTM below.
>
> ### Table: Average testing AUC through 10 datasets.
>
> | **Model**    | **RTM (Original)**   | **RTM (Selected on Validation)** |
> |--------------|----------------------|-----------------------------------|
> | **OCSVM**    | $72.61 \pm 21.8$     | $72.98 \pm 21.5$                 |
> | **DPAD**     | $76.56 \pm 17.1$     | $73.4 \pm 16.9$                  |
>
> Another more important limitation of RTM and EAG is they have an additional hyper-parameter. Therefore, we propose NPD.
>
> **Q14 and W1: strong assumption:** In Definition 6, we do not assume the "normal" data is approximated by an isotropic Gaussian. We illustrate the working flowchart in https://anonymous.4open.science/r/AutoUAD-FDFC/AutoAD_flowchart.pdf for your better understanding. In NPD, we use the **unlabeled** $\mathcal{X} _{val}$ to represent a "normal" set as it is indeed a subset of the normal data $\mathcal{X}$.
> We then generate an extra dataset $\mathcal{X} _{gen} \sim \mathcal{N}(\boldsymbol{\mu} _{trn}, \text{diag}(\sigma _{trn}^2))$. **The intuition behind this is that we hope $\mathcal{X} _{gen}$ can contain samples very close to real anomalies or that are not similar to the "normal" data**. A good UAD model should be able to distinguish the difference between $\mathcal{X} _{val}$ and $\mathcal{X} _{gen}$.
>
> **Q15:** As justified in Q14's response, we do not pose any assumption on the training data. Real-world datasets are more complex since their feature can be correlated non-linearly. Actually, we considered the correlation between coordinates in real-world data, i.e., in $\mathcal{X} _{trn}$. For this reason, we select isotropic Gaussian, which has higher entropy to generate diverse samples that are not similar to $\mathcal{X} _{trn}$. It is also justified in Fig. 3, 4, and 7. It shows $\mathcal{X} _{gen}$ is not similar to $\mathcal{X} _{trn}$ and $\mathcal{X} _{gen}$ often contains samples close to the real anomalies. In Theorem 3, we theoretically show the divergence between $\mathcal{X} _{gen}$ and $\mathcal{X} _{trn}$.
>
> **Q16:** The outline of $\mathcal{X} _{gen}$ (not $\mathcal{X} _{trn}$) is generally a hypersphere because $\mathcal{X} _{gen}$ is generated from an isotropic Gaussian.  In contrast, the outline of $\mathcal{X} _{trn}$ can be very different from a hypersphere because the variance across different coordinates varies. We use Fig. 4 to show that $\mathcal{X} _{trn}$ is roughly enclosed by the hypersphere determined by $\mathcal{X} _{gen}$, though "normal" (blue) point has a strong correlation across different coordinates.

---

> ### Author Response · Authors · 2024-11-19
> **Rebuttal: Part V**
>
> **Q17:** We feel sorry for the misleading. In Figure 6, the upper subfigures (first row) are the values of AUC on the test set. The lower subfigures (second row) are the values of metrics (NPD, EM/MV, EAG, RTM) evaluated on the training set. It is seen that the NPD is positively proportional to the AUC. We add the clarification and y-axis explicitly.
>
> **Q18 and W1: overly optimistic:** No. Please see our newly added flowchart of AutoUAD for better understanding in https://anonymous.4open.science/r/AutoUAD-FDFC/AutoAD_flowchart.pdf, where the datasets used for training and testing are separated. The data used for model selection is the **unlabeled** training data $\mathcal{X}$. Even in NPD, the $\mathcal{X}_{val}$ is a **unlabeled** subset. Hence, we argue our results are not overly optimistic.
>
> **Q19:** In our UAD setting, also known as one-class-classification, the training data are considered normal (may contain a few unlabeled anomalies). Therefore, for any dataset, we only use the "normal" data to perform training. The test data contains both normal samples and anomalous samples. See our implementation details in Appendix F. As for the anomaly ratio bound in Assumption 2, we assume no more than 20\% of the "normal" data behave like an anomaly. This is often caused by mislabeling and noise in the "normal" data. We claim assumption 2 is very mild. So, benchmark datasets with higher anomaly ratios do not violate Assumption 2 because, during the training phase, we never know the anomaly ratio in the testing phase.
>
> **F1, 2, 3, 4, 5, 9:** Thanks for your nice suggestions. We have revised the related content in the main text.
> For instance, we change the normal distribution in Definition 1 into "distribution of normal data". We change the standard normal distribution in Figure 3 into "standard Gaussian distribution"
>
>
> **F6:** Thanks for your suggestions. If $\text{median}(\boldsymbol{s})=0$, the denominator can be replaced by the smallest non-zero value in $\boldsymbol{s}$. If all values are 0, we set $RTM=0$. In the experiment, we barely observed the case median(s) close to 0. Please see the statistical information in the table below. The median is larger than 0 in most cases. In all experiments, we set $\epsilon=1e-9$. Compared with the $\text{median}(\boldsymbol{s})$, $\epsilon$ is always sufficiently small. In the 500 BO searches, no occurrences ($0\%$) that the $\text{median}(\boldsymbol{s})$ falls into the range $[-1e-6,1e-6]$.
>
> ### Table: Statistic information about training anomaly score **$\boldsymbol{s}$** for 4 UAD methods on the Satellite dataset over 500 BO searches.
>
> | **UAD Model** | **Avg. Median($\boldsymbol{s}$)** | **Min($\boldsymbol{s}$)** | **Max($\boldsymbol{s}$)** |
> |---------------|-----------------------------------|---------------------------|---------------------------|
> | AE            | $1.628 \pm 0.246$                | 0.5698                    | 9.155                    |
> | DeepSVDD      | $0.7376 \pm 0.213$               | 0.0087                    | 10.03                    |
> | OCSVM         | $0.4013 \pm 6.26$                | -94.99                    | 554.8                    |
> | DPAD          | $0.0026 \pm 0.0002$              | 0                         | 0.1223                   |
>
> **F10:** Thanks for your careful reading. NPD is correct, we have revised the text and figures.
>
> **We are looking forward to your feedback and please do not hesitate to let us know if there are any concerns or questions still not properly addressed. We are always here and eager to response to any of your questions. Thank you.**

---

> > ### Comment · Reviewer_FcaY · 2024-11-19
> >
> > Thank you for your extensive response.
> >
> > With the additional theoretical results and the explanation of the experiments, my main concerns have been resolved. Therefore, I change my rating accordingly.
> >
> > To avoid potential confusion, I would suggest to clearly state that the hyperparameter tuning and model selection is done with the training data only (and that the validation data is part of the training data), while the test data is only used to evaluate the final model.

---

> > > ### Author Response · Authors · 2024-11-19
> > >
> > > We sincerely appreciate your response to our rebuttal. Your support is the greatest encouragement to us. We just updated the manuscript, in which your suggested statement is added to the end of the first paragraph of Section 4.

---

### Official Review · Reviewer_DpZ3 · 2024-10-30

**Soundness:** 2
**Presentation:** 2
**Contribution:** 2
**Rating:** 6
**Confidence:** 4

**Summary:**

This paper aims at improving hyperparameter optimization of unsupervised anomaly detection methods. To that end, it first proposes three different metrics that can be used to estimate the performance of UAD methods at testing. The metrics are used within a Bayesian optimization framework to estimate the hyperparameters of different UAD methods.

**Strengths:**

- The work addresses a relevant problem that requires further attention from the literature
- A large set of datasets are used for evaluationLarge set of datasets considered for evaluation.

**Weaknesses:**

- The paper lacks clarity. As pointed in the questions section, there are multiple sentences that seem incomplete, which makes difficult to understand the concept being explained.
- There are some of the statements and claims that are contradictory across the paper. For example, the paper claims to be unsupervised and criticizes previous works that claim to be unsupervised but still require some form of supervision: this work aims at fixing this. Nonetheless, Definition 2 and remark 3.2 introduce labels for normal and abnormal points ($\mathcal{Y}$) and in the explanation of Definition 4 there seems to be the notion of supervision as there is the idea of true anomalies.
- Other claims lack a justification (see question 3 regarding assumption 2).
- The assumption of an isotropic Gaussian distribution for the generated dataset may be over-simple. Real-world datasets may have more complex dynamics.

**Questions:**

**Questions:**:
1. In Eq 4, if the median (median(s))  is close to zero, the selected value of $\epsilon$ may induce some instability of V_rtm. How do you deal with those cases?
2. In definition 1, what is $\mathcal{D}'_1$?
3. Assumption 2 lacks justification. What leads to conclude that 20% of the points on any given dataset are similar to true anomalies?
4. What is $\xi$ in Eq 5?
5. "In many practical scenarios, such as medical diagnosis and mechanical fault detection, noisy or wrongly collected data constitute a small portion" -> Do you have evidence that backs this claim?
6. If Bayesian optimization, according to the claims, works on supervised setting, how can it be used in this setup that is AutoUAD?

**Comments:**
- The formulation of definition 1 has some flaws. If one says that N points were drawn from a distribution it means that they were effectively drawn. One cannot then say that N is unknown. Perhaps what is really meant is that points in $\mathcal{X}$ come from two different distributions, but it is not possible to determine which points come from which (D_0 or D_1). Please reformulate to the latter clearly stating where points come without the need of mentioning the quantities (N). Also, explain what is the difference between $\mathcal{D}_1$ and $\mathcal{D}'_1$.
- Assumption 1 seems to be missing something : "... will assign the majority of data points with low anomaly scores"  to what? same for the minority. Please complete/
- What is the point of Figure 2? it does not seem to match what is written in the text. What makes the difference between, for instance, fig 2 a and b?
- The observations from assumption 1 and Figure 2 are somehow trivial. This is the typical assumption in most anomaly detection setups.
- There is some text missing in definition 4 ("there exists and the variance.." -> there exists what?) that makes text incomprehensible. Please revise the sentence.

**Minor comments:**
- "Histograms of training anomaly scores of models with low high testing AUCs" -> with low and high?

**Details Of Ethics Concerns:**

None.

---

> ### Author Response · Authors · 2024-11-18
> **Rebuttal: Part I**
>
> Dear Reviewer DpZ3,
>
> We sincerely thank your valuable comments. It seems that there are a few misunderstandings (e.g. Weakness 2 and Weakness 4). Our detailed responses to your comments are as follows.
>
> **Weakness 1:** Thank you so much for pointing out these issues of writing. We have fixed them in the revision. The revised details are listed below:
>
> - **Question 2:** $\mathcal{D}'_1$ is the distribution of true anomalies in the test data, while $\mathcal{D} _1$ is the distribution of possible anomalies (unlabeled) in the training data. We use both $\mathcal{D} _1$ and $\mathcal{D}'_1$ since the distribution of anomaly in the test data is not necessarily identical to the distribution of possible anomaly in the training data, which is a standard assumption in unsupervised anomaly detection. Moreover, the training data may not contain any anomalies.
>
> - **Question 4:** We notice this is a typo and it makes the reading of the whole section difficult. We revise Definition 4 by changing $s_k$ into $\xi$, making AG a function related to $\xi$ (a value from the distribution of $s$) and $p(s)$. Furthermore, we emphasize that $p(s)$ is an unknown and theoretical distribution of the anomaly score $\boldsymbol{s}$ to reduce the ambiguity.
>
> - **Comment 2:** We apologize for the misleading. This is a grammar error. The correct sentence should be "A good UAD model will assign low anomaly scores to the majority of data points, and assign relatively high anomaly scores to the minority of data points."
>
> - **Minor Comments:** Thanks for your careful reading. This is a typo. It should be "Histograms of training anomaly scores of models with low **and** high testing AUCs"
>
> **Weakness 2:** "Definition 2 and remark 3.2 introduce labels for normal and abnormal points... "
>
> **Response:** The notation of $\tilde{\mathcal{X}}$ and $\tilde{\mathcal{Y}}$ represents the **testing dataset**.
> * Our experiments and indeed all UAD research require a labeled test set to evaluate the effectiveness of final UAD models. In our work, we use the test dataset to evaluate the performance of the hyper-parameter-optimized UAD model. In the training stage and hyperparameter selection stage, we never use labeled data.
> * Moreover, we introduce $\tilde{\mathcal{X}}$ and $\tilde{\mathcal{Y}}$ in Definition 2 and formula (3) because we were presenting the goal of AutoUAD and the surrogate function $\mathcal{V}\left(\mathcal{M}_i, \mathcal{X}\right)$. The goal of AutoUAD is to perform hyperparameter optimization and model selection on (unlabeled) training data to ensure high accuracy on the unseen testing data.
>
> To give a comprehensive understanding of which part we use the labeled data, we provide a flowchart of AutoUAD at https://anonymous.4open.science/r/AutoUAD-FDFC/AutoAD_flowchart.pdf, where no ground-truth label is required during the training.
>
> **Weakness 2:** "Definition 4 there seems to be the notion of supervision ..."
>
> **Response:** Sorry for the misleading. Here, we would like to express that in an ideal case, selecting the threshold value depends on a prior anomaly ratio. Here, however, we do not have such prior information. As the following sentence justified, "due to ... and the lack of supervision, ...", threshold selection becomes an additional hyper-parameter which makes AG and even EAG not work well in the experiment. Therefore, no supervision is involved.
>
> **Weakness 3 and Question 3:** Thanks for your suggestions. As many Unsupervised Outlier/Anomaly detection literature assumed, the majority of the training data is normal. For the specific anomaly ratio in the dataset, scholars often make assumptions about the anomaly ratio [1, 2]. One can also refer to [3] to estimate the anomaly ratio in the dataset. Hence, we claim the assumption is proper. Nevertheless, such an assumption can also be regarded as a hyper-parameter in EAG metric. So, we further propose NPD metric which does not rely on Assumptions 1 and 2.
>
> [1] Nicolas Goix. How to evaluate the quality of unsupervised anomaly detection algorithms? ICML Workshop 2016.
>
> [2] Qiu et al. Latent outlier exposure for anomaly detection with contaminated data. ICML 2022.
>
> [3] Li et al. Deep anomaly detection under labeling budget constraints. ICML 2023.
>
> **Weakness 4:** Thanks for the comment. The isotropic Gaussian is not an assumption for the real-world dataset. In contrast, we intend to use isotropic Gaussian to form a region (hypersphere) to enclose the training data. In the hypersphere, the regions without training data are the regions where unseen anomalies may fall. We use the isotropic Gaussian to guide the model to learn a tight decision boundary. Real-world datasets are more complex. Our idea is also justified in Figs. 3, 4, and 7. It shows $\mathcal{X} _{gen}$ is not similar to $\mathcal{X} _{trn}$ and often contains samples close to the real anomalies.
> We added one more theorem (Theorem 4 in Appendix B) that provides a theoretical guarantee (error bound) for our method.

---

> ### Author Response · Authors · 2024-11-18
> **Rebuttal: Part II**
>
> **Question 1:** This is really an insightful question.
> We set $\epsilon=1e-9$ in the experiments and we barely observed the case median(s) close to 0. The following table provides some statistical information, where the median of $\mathbf{s}$ is much larger than our $\epsilon$. In other words, compared with the $\text{median}(\boldsymbol{s})$, $\epsilon$ is always sufficiently small. In the 500 BO searches, no occurrences ($0\%$) that the $\text{median}(\boldsymbol{s})$ falls into the range $[-1e-6,1e-6]$. Therefore, the bias or instability given by $\epsilon$ is tiny.
>
> The small $\text{median}(\boldsymbol{s})$ may be a shortcoming of our $\mathcal{V}_ {\text{RTM}}$ but the other two metrics ($\mathcal{V}_ {\text{EAG}}$ and $\mathcal{V}_ {\text{NPD}}$) we proposed do not have this shortcoming. More importantly, as shown by the experiments (e.g. Table 1), $\mathcal{V}_ {\text{NPD}}$ is much more effective than $\mathcal{V}_ {\text{RTM}}$.
>
> **Table:** Statistic information about training anomaly score **$\boldsymbol{s}$** for 4 UAD methods on the Satellite dataset over 500 BO searches.
>
> | **UAD Model** | **Avg. Median($\boldsymbol{s}$)** | **Min($\boldsymbol{s}$)** | **Max($\boldsymbol{s}$)** |
> |---------------|-----------------------------------|---------------------------|---------------------------|
> | AE            | $1.628 \pm 0.246$                | 0.5698                    | 9.155                    |
> | DeepSVDD      | $0.7376 \pm 0.213$               | 0.0087                    | 10.03                    |
> | OCSVM         | $0.4013 \pm 6.26$                | -94.99                    | 554.8                    |
> | DPAD          | $0.0026 \pm 0.0002$              | 0                         | 0.1223                   |
>
>
> **Question 5:** Anomaly detection, also known as novelty detection or rare event detection, means the anomaly is rare. It is common in many real applications, such as security (intruders), geology (earthquake), food control (foreign objects), economics (bankruptcy), and neuroscience (an unexperienced stimulus) [1]. In addition, consider a cloud computing system requiring high system availability ($\ge 99.9$\%) [2]. Then, the collected system log may only contain $0.1$\% system failure data.
>
> These anomaly events themselves are rare. If they are unexpectedly considered as "normal" data, it is still a very small portion. We hope the evidence can support our claim.
>
> [1] Ander et al. Analyzing rare event, anomaly, novelty and outlier detection terms under the supervised classification framework. Artificial Intelligence Review 53 (2020): 3575-3594.
>
> [2] Kai Hwang. Cloud computing for machine learning and cognitive applications. Mit Press, 2017, Chapter 1, 1.4.
>
> **Question 6:** Although Bayesian optimization is usually used for tuning the hyperparameters in supervised learning, it can be used for unsupervised learning **when we derive an effective surrogate function or metric to evaluate the model performance using the unlabeled data only**. For instance, [3] used Bayesian optimization to search hyperparameters for spectral clustering algorithms, which are unsupervised.
>
> In our work, as described in the text around formula (3), we want to construct some $\mathcal{V}\left(\mathcal{M}_i, \mathcal{X}\right)$ to evaluate the model performance using the unlabeled training data only and we hope that $\mathcal{V}\left(\mathcal{M}_i, \mathcal{X}\right)$ is a good proxy for the expected testing error (never known). Therefore, we provided three examples of $\mathcal{V}\left(\mathcal{M}_i, \mathcal{X}\right)$ including RTM, EAG, and NPD, which are calculated using the unlabeled training data only. We then use Bayesian optimization to maximize the three metrics with respect to the hyperparameters of UAD methods.
>
> To make the general idea of applying Bayesian optimization to UAD more intuitive and clearer, we provide a working flowchart in https://anonymous.4open.science/r/AutoUAD-FDFC/AutoAD_flowchart.pdf.
>
> [3] Fan et al. A simple approach to automated spectral clustering. NeurIPS 2022.

---

> ### Author Response · Authors · 2024-11-18
> **Rebuttal: Part III**
>
> **Comment 1: Definition 1:** Thanks for your comment. What we want to express is exactly the same as you suggested. The misleading may be caused by that there are three numbers $N,N_0,N_1$ in the definition. $N$ is actually known since it is the total number of samples in our training dataset $\mathcal{X}$. In contrast, $N_0$ and $N_1$ are unknown, though $N_0 + N_1 = N$. We think it is sufficient to express "it is not possible to determine which points come from which ($D_0$ or $D_1$)". Please feel free to let us know if this is still not clear enough.
>
> **Comment 1: difference between $\mathcal{D}_1$ and $\mathcal{D}'_1$:**
> $\mathcal{D} _1$ is the distribution of the anomalies in the training data if $N_1\neq 0$. $\mathcal{D}'_1$ is the distribution of anomalies in the (unseen) test data. The distribution of anomalies in the test data is not necessarily identical to the distribution of possible anomalies in the training data. That's why we introduce $\mathcal{D}'_1$ in addition to $\mathcal{D}_1$.
>
>
> **Comment 3:** Fig. 2(a) is an OCSVM with a low test AUC score. It does not satisfy Assumption 1 because the majority of training samples have high anomaly scores. Fig. 2(b) is an OCSVM with a high test AUC score, where the majority of training samples have relatively low anomaly scores. It satisfies the Assumption 1. By comparing Fig. 2(c) and Fig. 2(d), although both of them seem to satisfy Assumption 1, the model with higher test AUC assigns higher anomaly scores to a small percentage of data described in line 235 (new version). So, models with high testing AUC should be evaluated with a higher metric value like RTM and EAG. We believe the figures are consistent with the main text.
>
> The reason for the differences between them is the use of **different hyper-parameters** of the model trained on the same training data. That's why we consider the hyper-parameter optimization is challenging.
>
> **Comment 4:** Thanks for the comment. Since it is common in UAD research, we adopt such an assumption here to propose the RTM metric. However, such a metric requires setting additional hyper-parameter, e.g. $\tau$, making the performance empirical. Hence, we further propose NPD metric that does not require an additional hyper-parameter.
>
> **Comment 5** We apologize for the typo. The correct sentence should be "..., there exists $\xi$ such that $P(s \ge \xi)$ is small, and ..." please see our revised Eq. (5).
>
> We hope our response can be helpful in addressing your concerns. Please do not hesitate to let us know if any of your concerns haven't been properly addressed or you have further questions.
>
> Sincerely,
>
> Authors

---

> ### Comment · Reviewer_DpZ3 · 2024-11-24
>
> Many thanks for the effort to prepare the detailed rebuttal. I have updated the score, reflecting some doubts I still have on the usage of a Gaussian distribution. As the authors claim, real datasets are much more complex and the paper fails to explain why this would still work.
>
> Second, if 20% of anomalies is a hyperparameter, how sensitive the method is to it?

---

> ### Author Response · Authors · 2024-11-25
>
> We are very grateful for your feedback.
>
> Regarding the Gaussian distribution, we'd like to provide more clarification.
>
> * In our metric NPD, the Gaussian distribution is not used to approximate real datasets and is not used as an assumption for real data. NPD is real-distribution agnostic, no matter how complex the real data distribution is.
>
> * Our goal of using Gaussian distribution is to form a large hypersphere to enclose most of the training data, which is feasible owing to the definition $\mathcal{N}(\boldsymbol{\mu}_ {t r n}, \operatorname{diag}(\boldsymbol{\sigma}_{t r n}^2))$. As shown in Figure 4, we visualized partial features of four real datasets. These real datasets (blue points) are very different from Gaussian and most of the samples are enclosed by the hypersphere.
>
> * In the hypersphere, the regions not filled by the training data (blue points) are potential regions of anomaly. Our metric NPD measures the anomaly difference between the potential anomalous regions and the normal regions (where the blue points lie in), and hence guides the model to learn a compact decision boundary for anomaly detection.
>
> * The effectiveness of NPD is also theoretically supported by Theorem 2, Theorem 3, and Theorem 4.
>
>
> Regarding the "at most  $20\%$" assumption, we added the following experiments.
>
> Due to time constraints, we selected five datasets, Breastw, Fault, Glass, Pima, and SpamBase, to conduct two sets of experiments to examine EAG's sensitivity to different anomaly ratios in the assumption.
>
> * First, we vary the choice of $s_{thr}$ as it is decided by the anomaly ratio $\hat{r}$ in the assumption, where $s_{thr} = G^{-1}(1-\hat{r})$. Note that in this case, the training data are not contaimined. The results are shown below. We see that the OCSVM is not very sensitive to $\hat{r}$.
>
> **Table:** Test AUC and F1 scores of EAG on OCSVM and DPAD across 5 datasets with different choices of $r$ (corresponding to different possible maximum anomaly ratios)
> \begin{matrix}
> \hline \hline
> \textbf{Model} & \hat{r}        & \textbf{1\\%}          & \textbf{5\\%}          & \textbf{10\\%}           & \textbf{15\\%}          & \textbf{20\\%}           & \textbf{25\\%}           & \textbf{30\\%}            \\\\ \hline
> {OCSVM} & AUC        &  68.39 \pm 19.6        &  68.37 \pm 21.5        &  70.61 \pm 19.8        &  69.75 \pm 18.4        &  72.19 \pm 19.2        &  72.54 \pm 19.7        &  81.52 \pm 10.9        \\\\
>                        & F1         &  60.41 \pm 31.4        &  60.81 \pm 31.8        &  61.67 \pm 32.0        &  58.66 \pm 31.6        &  57.68 \pm 31.0        &  56.94 \pm 30.9        &  64.79 \pm 25.7        \\\\ \hline
> {DPAD}  & AUC        &  53.62 \pm 6.1         &  55.95 \pm 12.0        &  61.03 \pm 21.2        &  66.22 \pm 22.1        &  54.77 \pm 7.2         &  69.17 \pm 18.3        &  54.99 \pm 7.4         \\\\
>                       & F1         &  44.97 \pm 19.0        &  46.72 \pm 13.8        &  53.86 \pm 30.8        &  54.55 \pm 22.6        &  47.88 \pm 21.2        &  60.87 \pm 26.6        &  45.49 \pm 26.1        \\\\ \hline\hline
> \end{matrix}
>
>
> * Second, we construct a contaminated training dataset by adding true anomalies (keep unlabeled during the training). We can see that both OCSVM and DPAD perform better when the real anomaly ratio is less than the hyperparameter $20\\%$ in our assumption and they perform best when the real anomaly ratio is the same as the hyperparameter $20\\%$. This further demonstrates the effectiveness of our assumption.
>
> **Table:** Test AUC and F1 scores of EAG on OCSVM and DPAD across 5 datasets with different contamination ratios.
> \begin{matrix}
> \hline\hline
> \textbf{Model} & \textbf{Contamination Ratio}        & \textbf{1\\%}          & \textbf{5\\%}          & \textbf{10\\%}           & \textbf{15\\%}          & \textbf{20\\%}           & \textbf{25\\%}          & \textbf{30\\%}           \\\\ \hline
> OCSVM & AUC        &   67.62 \pm 20.4         &   61.43 \pm 20.2         &   63.04 \pm 21.5         &   67.42 \pm 17.7         &   68.78 \pm 18.3         &   60.68 \pm 25.6         &   59.77 \pm 23.4         \\\\
>                        & F1         &   60.54 \pm 31.6         &   56.19 \pm 29.9         &   57.17 \pm 30.6         &   56.48 \pm 35.2         &   56.07 \pm 35.5         &   48.75 \pm 34.4         &   53.99 \pm 35.0         \\\\ \hline
> DPAD  & AUC        &   55.49 \pm 10.9         &   58.16 \pm 16.0         &   52.02 \pm 24.7         &   52.86 \pm 4.7          &   60.79 \pm 20.9         &   48.36 \pm 13.6         &   53.74 \pm 26.6         \\\\
>                        & F1         &   46.77 \pm 14.0         &   54.34 \pm 21.1         &   49.24 \pm 29.4         &   47.07 \pm 20.1         &   51.86 \pm 34.0         &   44.39 \pm 25.2         &   51.60 \pm 34.4         \\\\ \hline\hline
> \end{matrix}
> Thank you again and we hope this response can address your remaining concerns. We are looking forward to your further feedback.

---

> > ### Author Response · Authors · 2024-12-02
> >
> > Dear Reviewer,
> >
> > Since the author-reviewer discussion period is coming to an end, we'd like to know if our additional response addressed your doubt and question or not. Thank you for your time.
> >
> > Regards,
> >
> > Authors

---

### Official Review · Reviewer_7zTy · 2024-10-30

**Soundness:** 3
**Presentation:** 3
**Contribution:** 3
**Rating:** 6
**Confidence:** 4

**Summary:**

Unsupervised anomaly detection (UAD) faces challenges in model selection and hyper-parameter tuning due to the lack of labeled anomalies and dataset variability. This work introduces three evaluation metrics—relative-top-median, expected-anomaly-gap, and normalized pseudo discrepancy (NPD)—to estimate model performance without labels. Using Bayesian optimization, these metrics streamline UAD tuning, showing effectiveness across 38 datasets.

**Strengths:**

- The paper is well-written.
- The focus on hyper-parameter optimization in UAD is important and well-motivated.
- The proposed method is novel.
- Extensive experiments effectively demonstrate the method’s effectiveness.

**Weaknesses:**

- The details of AutoUAD are complex and difficult to understand.

**Questions:**

Please see Weaknesses

---

> ### Author Response · Authors · 2024-11-18
> **Rebuttal**
>
> Dear Reviewer 7zTy:
>
> Thank you so much for recognizing the novelty and effectiveness of our method.
> Regarding the weakness you mentioned, we here explain the whole pipeline of AutoUAD for your better understanding. Particularly, we draw an intuitive figure to show the flow chart of AutoUAD at https://anonymous.4open.science/r/AutoUAD-FDFC/AutoAD_flowchart.pdf.
>
> **Motivation:** In UAD, since the training data are unlabeled, optimizing the hyper-parameters of any UAD method and comparing between different UAD methods are both challenging. We need to establish some metrics to evaluate the quality of UAD methods by utilizing the unlabeled training data only, **without accessing any labeled data**.
>
> **Contributions:**
> - In this paper, we propose two internal evaluation metrics, relative-top-median (RTM) and expected-anomaly-gap (EAG), and one semi-internal evaluation metric, normalized pseudo discrepancy (NPD), to measure the quality of a UAD model based on the **unlabeled** training data.
>
> - We implement automated UAD using Bayesian optimization, for which the objective is maximizing one of RTM, EAG, and NPD, with respect to the hyperparameters of UAD methods.  Therefore, our AutoUAD automatically and efficiently selects the possibly best hyper-parameters for UAD methods.
>
> - We provide theoretical guarantees for our NPD metric to ensure feasibility and reliability. In this revision, we provide one more theorem (Theorem 4 in Appendix B) that states that maximizing our NPD metric can reduce the upper bound of the false positive rate and false negative rate.
>
> - Extensive empirical experiments on 38 benchmark datasets (detailed in Appendix D) show the proposed NPD metric consistently outperforms existing model selection heuristics and works well on complex state-of-art UAD algorithms.
>
> **AutoUAD Details:** We add a figure of a working flowchart for your easier understanding of AutoUAD in https://anonymous.4open.science/r/AutoUAD-FDFC/AutoAD_flowchart.pdf.
> In the training phase, candidate UAD models $\mathbb{M}$ contain multiple UAD algorithms $\{\mathcal{M} _1, \mathcal{M} _2, ..., \mathcal{M} _C \}$, and each algorithm $\mathcal{M} _i$ has multiple candidate hyper-parameters:
> $\{\Theta _i^{(1)}, \Theta _i^{(2)}, ... , \Theta _i^{(j)}, ...\}, \Theta _i^{(j)} \in \prod^{H_i} _{k=1} \mathcal{S} _k^{(i)}.$
> For each $\mathcal{M}_i$, the Bayesian optimizer will select hyper-parameters to train the UAD model. After training ends, the model is evaluated by our (semi-)internal metrics (RTM, EAG, or NPD) using the training set where no ground-truth label is required. The evaluation output $\mathcal{V}(\mathcal{M}_i, \mathcal{X})$ provides feedback to the Bayesian optimizer to select new hyper-parameters for the next round. During the whole training process, no ground-truth label is required.
>
> For internal metrics, RTM and EAG, we obtain $\boldsymbol{s}$ from the trained UAD model $\mathcal{M}_i(\Theta_i^{(j)})$ inferred on the training dataset $\mathcal{X}$. Then, RTM and EAG can be calculated using Eq. (4) and Eq. (6), respectively.
>
> For the proposed semi-internal metric NPD, the training dataset is randomly split into $\mathcal{X} _{trn}$  and $\mathcal{X} _{val}$  before the training phase.
> An extra dataset $\mathcal{X} _{gen}$ is generated from an isotropic Gaussian distribution:
> $
> \mathcal{N}(\boldsymbol{\mu} _{trn}, \text{diag}(\boldsymbol{\sigma}^2 _{trn})),
> $
> where $\boldsymbol{\mu} _{trn}$ and $\boldsymbol{\sigma}^2 _{trn}$ are the mean and variance vectors of $\mathcal{X} _{trn}$.
> The dataset $\mathcal{X} _{trn}$ is used to train the UAD model. After training, anomaly scores $\boldsymbol{s} _{val}$ and $\boldsymbol{s} _{gen}$ are computed from $\mathcal{X} _{val}$ and $\mathcal{X} _{gen}$, respectively. Then, NPD is calculated by taking $\boldsymbol{s} _{val}$ and $\boldsymbol{s} _{gen}$ as input using Eq. (7).
> Note that $\mathcal{X} _{val}$ is still an **unlabeled** subset of $\mathcal{X}$, and $\mathcal{X} _{gen}$ is just random noise. These datasets, $\mathcal{X} _{val}$ and $\mathcal{X} _{gen}$, are only used for semi-internal evaluation and are not involved in model training.
>
> The intuition behind NPD is that we hope $\mathcal{X} _{gen}$ contains samples that are not similar to the normal data $\mathcal{X}$. The model should be able to distinguish the difference between $\mathcal{X} _{val}$ and $\mathcal{X} _{gen}$.
> We argue that $\mathcal{X} _{gen}$ can contain samples close to real anomalies, so that NPD can highlight the significance of evaluating a good UAD model. This claim is justified in Theorem 3.
>
> In the testing stage, the selected model ($\mathcal{M}_i(\Theta_i^*)$ or $\mathcal{M}^*(\Theta_i^*)$) is evaluated by $\mathcal{E}$ (AUC/F1 metric) using the testing set with the ground-truth label to show the effectiveness of our methods.
>
> We hope the clarification and the newly added flowchart are helpful for your understanding. We are looking forward to your further feedback.
>
> Sincerely,
>
> Authors

---

> > ### Author Response · Authors · 2024-11-24
> >
> > Dear Reviewer 7zTy,
> >
> > Did our explanation for AutoUAD and the flowchart  https://anonymous.4open.science/r/AutoUAD-FDFC/AutoAD_flowchart.pdf address your concern? Please do not hesitate to let us know if there is still anything unclear or if you have further questions.
> >
> > Sincerely,
> >
> > Authors

---

> > > ### Author Response · Authors · 2024-12-01
> > >
> > > Dear Reviewer 7zTy,
> > >
> > > While the author-reviewer discussion period is going to end, we haven't received any feedback from you on our responses. It would be highly appreciated if you provide feedback to our explanation and revision regarding our work.
> > >
> > > Sincerely,
> > >
> > > Authors

---

### Meta-Review · Area_Chair_AFTz · 2024-12-19

**Metareview:**

Based on the reviews, I recommend accepting the paper. All three reviewers, who are domain experts, have suggested acceptance and expressed high confidence in their evaluations. Notably, one reviewer has provided an exceptionally detailed review, offering 30 constructive suggestions and technical comments. The reviewers highlight several strengths of the work, including its high-quality empirical evaluation and its significance to the ICLR community.

**Additional Comments On Reviewer Discussion:**

The main points raised by the reviewers focused on clarity, assumptions, methodology, and experimental results.:
- **Reviewer 7zTy** initially found the AutoUAD pipeline complex and difficult to understand. The authors responded by clarifying the pipeline, which addressed the concern and led the reviewer to increase the rating.

- **Reviewer DpZ3** pointed out issues with clarity, contradictions, and the assumption of an isotropic Gaussian distribution. The authors provided detailed explanations and additional results, particularly addressing concerns about the Gaussian assumption. This satisfied the reviewer, who revised the rating.

- **Reviewer FcaY** questioned the strong assumptions and the overly optimistic experimental results. The authors responded with comprehensive clarifications, including theoretical justifications and additional results, which resolved the concerns and led the reviewer to change the rating to acceptance.

In weighing these points, I found the authors' responses to key concerns satisfactory. While clarity and assumptions were initially questioned, the authors addressed these through revisions and additional results.

---

### Decision · Program_Chairs · 2025-01-22

Accept (Poster)